# LD$^2$: Scalable Heterophilous Graph Neural Network with Decoupled Embeddings

**Ningyi Liao**
Nanyang Technological University
`liao0090@e.ntu.edu.sg`

**Siqiang Luo**[*]
Nanyang Technological University
`siqiang.luo@ntu.edu.sg`

**Xiang Li**
East China Normal University
`xiangli@dase.ecnu.edu.cn`

**Jieming Shi**
Hong Kong Polytechnic University
`jieming.shi@polyu.edu.hk`

## Abstract

Heterophilous Graph Neural Network (GNN) is a family of GNNs that specializes in learning graphs under heterophily, where connected nodes tend to have different labels. Most existing heterophilous models incorporate iterative non-local computations to capture node relationships. However, these approaches have limited application to large-scale graphs due to their high computational costs and challenges in adopting minibatch schemes. In this work, we study the scalability issues of heterophilous GNN and propose a scalable model, LD$^2$, which simplifies the learning process by decoupling graph propagation and generating expressive embeddings prior to training. Theoretical analysis demonstrates that LD$^2$ achieves optimal time complexity in training, as well as a memory footprint that remains independent of the graph scale. We conduct extensive experiments to showcase that our model is capable of lightweight minibatch training on large-scale heterophilous graphs, with up to $15\times$ speed improvement and efficient memory utilization, while maintaining comparable or better performance than the baselines. Our code is available at: `https://github.com/gdmnl/LD2`.

## 1 Introduction

Graph Neural Networks (GNNs) combine graph management techniques and neural networks to learn from graph-structured data, and have shown remarkable performance in diverse graph processing tasks, including node classification [1, 2], link prediction [3, 4], and community detection [5, 6]. Common GNN models rely on the principle of *homophily*, which assumes that connected nodes tend to be similar to each other in terms of classes [7]. This inductive bias introduces additional information from the graph structure and improves model performance in appropriate tasks [8].

However, this assumption does not always hold in practice. A broad range of real-world graphs are *heterophilous*, where class labels of neighboring nodes usually differ from the ego node [9]. In such cases, the aggregation mechanism employed by conventional GNNs, which only passes messages from a node to its neighbors, may mix the information from non-homophilous nodes and cause them to be less discriminative. Consequently, the locality-based design is considered less advantageous or even potentially harmful in these applications [10, 11]. Various solutions have been proposed to address the heterophily problem, giving rise to a class of specialized GNNs known as heterophilous GNNs. Common strategies to address heterophily include discovering multi-hop or global graph relations [12, 13, 14, 15, 16, 17], and retrieving expressive node information through enhanced network architectures [18, 19, 20, 21, 22, 23].

---

[*]Corresponding author.

37th Conference on Neural Information Processing Systems (NeurIPS 2023).

Table 1: Time and memory complexity of homophilous and non-homophilous models with respect to precomputation, training, and inference stages. "Batch" refers to the minibatch availability of the original model design. Training and inference memory indicate the GPU usage for storing and updating representation and weight matrices, while precomputation is mostly on RAM. Training and inference time complexity represent the forward-passing computational operations on respective node sets.

| Model | Batch | Pre. Mem. | Training Mem. | Inference Mem. | Pre. Time | Training Time | Inference Time |
|---|---|---|---|---|---|---|---|
| MLP | Y | – | $O(Ln_bF + LF^2)$ | $O(Ln_bF + LF^2)$ | – | $O(ILnF^2)$ | $O(LnF^2)$ |
| GCN [1] | N | – | $O(LnF + LF^2)$ | $O(LnF + LF^2)$ | – | $O(ILmF + ILnF^2)$ | $O(LmF + LnF^2)$ |
| GSAINT [27] | Y | – | $O(L_PLn_bF + LF^2)$ | $O(LnF + LF^2)$ | – | $O(IL_PLnF + ILnF^2)$ | $O(LmF + LnF^2)$ |
| APPNP [28] | Y | $O(m)$ | $O(Ln_bF + LF^2 + nn_b)$ | $O(Ln_bF + LF^2 + nn_b)$ | $O(m)$ | $O(IL_PmF + ILnF^2)$ | $O(L_PmF + LnF^2)$ |
| PPRGo [29] | Y | $O(n/\delta)$ | $O(Ln_bF + LF^2)$ | $O(Ln_bF + LF^2)$ | $O(m/\delta)$ | $O(InF^2 + ILnF^2)$ | $O(nF^2 + LnF^2)$ |
| SGC [30] | Y | $O(nF)$ | $O(Ln_bF + LF^2)$ | $O(Ln_bF + LF^2)$ | $O(L_PmF)$ | $O(ILnF^2)$ | $O(LnF^2)$ |
| GPRGNN [17] | N | $O(m)$ | $O(LnF + LF^2 + m)$ | $O(LnF + LF^2 + m)$ | $O(m)$ | $O(IL_PmF + ILnF^2)$ | $O(L_PmF + LnF^2)$ |
| GCNJK [21] | N | – | $O(L_CnF + L_CF^2)$ | $O(L_CnF + L_CF^2)$ | – | $O(ILmF + ILnF^2)$ | $O(LmF + LnF^2)$ |
| MixHop [12] | N | – | $O(CLnF + CLF^2)$ | $O(CLnF + CLF^2)$ | – | $O(IL_PLmF + ILnF^2)$ | $O(L_PLmF + LnF^2)$ |
| LINKX [26] | Y | – | $O(L_Cn_bF + L_CF^2 + nF)$ | $O(L_Cn_bF + L_CF^2 + nF)$ | – | $O(ImF + ILnF^2)$ | $O(mF + LnF^2)$ |
| **LD² (ours)** | Y | $O(CnF)$ | $O(L_Cn_bF + L_CF^2)$ | $O(L_Cn_bF + L_CF^2)$ | $O(L_PmF)$ | $O(ILnF^2)$ | $O(LnF^2)$ |

**Existing heterophilous GNNs are not scalable enough.** Scalability has become a prominent concern in GNN studies. The ever-increasing sizes of graph data nowadays can easily exceed the memory limit of devices such as GPUs, rendering these models impractical for large-scale tasks [24, 25]. We observe that this issue is particularly critical in the context of heterophilous GNNs, due to an inherent conflict that most current models have not taken into account: heterophily-oriented designs usually rely on non-local information calculated by certain types of whole-graph operations. As the graph structure is involved, the time and memory overhead escalates substantially with the graph size. A recent investigation [26] reveals that all the evaluated full-graph GNNs run out of 24GB GPU memory when applied to the million-scale graph wiki (1.77M nodes, 243M edges), while minibatch strategies often result in performance loss. It is thus crucial to develop GNNs scalable to large graphs while retaining the capability for heterophily.

**LD²: a scalable solution via decoupling.** In this work, we examine the scalability problem and propose LD², a scalable GNN model for heterophilous graphs with Low-Dimensional embeddings and Long-Distance aggregation. The model highlights simplicity by decoupling graph dependency from iterative computations and solely learning from a set of precomputed embeddings. Derived from node attributes and graph topology, these novel embeddings are capable of aggregating node relations of varying objectives and distances in the graph into low-dimensional features. To facilitate the decoupled scheme, we specifically propose an algorithm to efficiently estimate all embeddings before training, which enjoys time complexity only linear to the graph scale and a guaranteed precision bound. After the precomputation, a simple but powerful multi-channel neural network is subsequently employed to learn from the extracted node features. Theoretical and empirical results showcase that the combination of embeddings effectively retrieves representations among heterophilous nodes. On the efficiency aspect, LD² is advantageous in its scalable design, including the straightforward minibatch scheme, optimal training and inference times, and the superior memory utilization.

**Our contribution.** (1) We propose LD² as a scalable GNN under heterophily, which removes the reliance on iterative train-time full-graph computations. The model realizes theoretically optimized training, highlighting time complexity that is only linear to the number of nodes $O(n)$ and memory overhead independent of the graph scale. To the best of our knowledge, LD² is the first model achieving such optimization in the context of heterophilous GNNs. (2) We design an array of feature and topology embeddings by applying multi-hop discriminative propagation, encoding expressive node representation within a compact size. An end-to-end precomputation algorithm is proposed for efficient embedding calculation with a linear complexity. (3) We conduct comprehensive experiments to evaluate the efficacy and efficiency of LD². On large-scale datasets, our model demonstrates $3$–$15\times$ faster minibatch training and inference, and up to $5\times$ smaller memory footprint than state of the art, with comparable or better accuracy. Particularly, it completes training on the wiki dataset within 1 minute and inference in 0.1 seconds, demanding only 5GB of GPU memory.

## 2 Preliminary and Related Work

**Graph Notation and Heterophily.** In an undirected graph $G = (V, E)$ with node set $V$ and edge set $E$, the number of nodes, number of edges, and average degree are denoted by $n = |V|$, $m = |E|$, and $d = m/n$, respectively. The neighborhood of an ego node $u \in V$ is the set $\mathcal{N}(u) = \{v|(u,v) \in E\}$, and its degree $d(u) = |\mathcal{N}(u)|$. The diagonal degree matrix is $\boldsymbol{D} = \mathrm{diag}(d(u_1), d(u_2), \cdots, d(u_n))$.

The graph connectivity is represented by the adjacency matrix $\boldsymbol{A} \in \mathbb{R}^{n \times n}$. We adopt the general graph normalization scheme [31, 32] with coefficients $a, b \in [0, 1]$ and $\bar{\boldsymbol{A}} = \boldsymbol{D}^{-a} \boldsymbol{A} \boldsymbol{D}^{-b}$. The normalized adjacency matrix with self-loop edges is also frequently used, which is denoted as $\tilde{\boldsymbol{A}} = (\boldsymbol{I} + \boldsymbol{D})^{-a} (\boldsymbol{I} + \boldsymbol{A})(\boldsymbol{I} + \boldsymbol{D})^{-b}$, and the corresponding graph Laplacian matrix is $\tilde{\boldsymbol{L}} = \boldsymbol{I} - \tilde{\boldsymbol{A}}$. Each node $u \in V$ is represented by an $F$-dimensional attribute vector $\boldsymbol{X}(u)$, which composes the attribute matrix $\boldsymbol{X} \in \mathbb{R}^{n \times F}$. The graph heterophily is measured by the node homophily score [14], which is the average proportion of the neighbors with the same class of each node.

**Vanilla and Sampling-based GNNs.** We summarize our analysis of the time and memory complexity of related GNN models in Table 1. In general, a GNN recurrently computes the node representation $\boldsymbol{H}^{(l)}$ in its $l$-th layer. For the vanilla GCN [1], the model input is the attribute matrix $\boldsymbol{H}^{(0)} = \boldsymbol{X}$, and the layer representation is updated by $\boldsymbol{H}^{(l+1)} = \sigma(\tilde{\boldsymbol{A}} \boldsymbol{H}^{(l)} \boldsymbol{W}^{(l)})$, $l = 0, 1, \cdots, L - 1$, where $\boldsymbol{W}^{(l)}$ is the trainable weight matrix each layer, $\sigma(\cdot)$ is the activation function, and $L$ is the number of layers. For simplicity we assume the feature size $F$ to be constant in all layers. Previous research [33, 31] points out that the operation of dominating expense in both GCN training and inference is the *graph propagation* $\tilde{\boldsymbol{A}} \cdot \boldsymbol{H}^{(l)}$, which can be regarded as repetitive sparse-dense matrix multiplications, resulting in a total complexity $O(LmF)$. The overhead for *feature transformation* by applying $\boldsymbol{W}^{(l)}$ is $O(LnF^2)$. These two procedures are iteratively performed for $I$ epochs throughout training. In terms of memory usage, GCN typically requires $O(LnF + LF^2)$ space to store layer-wise node representations and weight matrices, respectively. For large-scale cases where $n \gg F$, the overhead of dense node representations $O(LnF)$ becomes the primary term [24].

The above analysis indicates that the scalability bottleneck of GCN lies in the time complexity of graph propagation as well as the memory overhead of full-graph representation. There is a large scope of GNNs attempting to address the issue by sampling techniques, which simplify the propagation by replacing the entire graph with subgraphs in minibatches [2, 27, 33]. For example, the widely-used GSAINT [27] incorporates $L_P$-hop random walk sampling, reducing the in-memory representation to $O(L_P L n_b F)$, where $n_b$ is the batch size. However, it is not applicable to the inference stage.

**Full-graph Heterophilous GNNs.** In the context of GNNs under heterophily, a large number of models augment the *spatial* graph convolution to a full-graph scheme, relying upon the complete graph topology to compute inter-node relationships. For instance, H₂GCN [13] and MixHop [12] incorporate 2-hop propagation $\tilde{\boldsymbol{A}}^2 \boldsymbol{H}^{(l)}$, while GeomGCN [14] and GloGNN [22] exploit hierarchical computation on non-local connections. These high-order calculations are shown to be effective in retrieving information beyond immediate neighbors, but come at the price of more complex propagation operations. Another common practice for non-homophilous design is altering transformation to learn from multiple features, i.e. channels. MixHop [12] mixes its multi-hop representation as $\boldsymbol{H}^{(l+1)} = \sigma(\boldsymbol{H}^{(l)} \boldsymbol{W}_0^{(l)} \| \tilde{\boldsymbol{A}} \boldsymbol{H}^{(l)} \boldsymbol{W}_1^{(l)} \| \tilde{\boldsymbol{A}}^2 \boldsymbol{H}^{(l)} \boldsymbol{W}_2^{(l)})$, where $(\cdot \| \cdot)$ denotes matrix concatenation. GGCN [19] and ACM [20] select frequency-based filters expressed by $\tilde{\boldsymbol{A}} \boldsymbol{H} \boldsymbol{W}_l, (\boldsymbol{I} - \tilde{\boldsymbol{A}}) \boldsymbol{H} \boldsymbol{W}_h, \boldsymbol{I} \boldsymbol{H} \boldsymbol{W}_i$, while GCNJK [21] records individual layer representations as channels. Denote the number of channels as $C$ and $L_C = L + C$, employing multi-channel learning increases the memory budget for node representations and weight matrices by a factor related to $C$ or $L_C$.

Alternatively, a line of approaches choose to apply learnable *spectral* filters for flexibility on non-homophilous data, deriving specific propagation in the spectral domain for various graphs. GPRGNN [17] generalizes the multi-hop propagation as $\boldsymbol{T} = \sum_{l=0}^{L_P} \theta_l \tilde{\boldsymbol{A}}^l$, which inevitably involves the entire adjacency matrix for acquiring weight factors $\theta_l$. BernNet [34] and ChebNetII [35] respectively utilize order-$L_P$ Bernstein and Chebyshev polynomials with respect to $\tilde{\boldsymbol{L}}$ to approximate their spectral filters. These models focus relatively more on the expressiveness of graph propagation rather than the overall GNN architectural design, usually demanding additional cost for determining spectral filters and updating the learnable basis.

In spite of their advantageous capabilities, above full-graph designs hardly address the scalability bottleneck in GNN propagation, which can be observed from the $O(n)$ or $O(m)$ terms in their complexity. Recent studies also discover that heterophilous GNNs are naturally unsuitable for enforcing scalable training via sampling-based minibatching, since their distant or full-graph information is heavily overlooked in batches built on locality [11]. Evaluations show that simply fitting these models to learn from induced subgraph samples causes performance degradation [26]. Therefore, we believe these full-graph models targeting effectiveness are orthogonal to our study on scalability.

**Decoupled GNNs.** As the scalability of GNN is closely tied to the graph propagation, a promising approach to simplify the process is decoupling it from iterative training and efficiently computing it in advance. The representative two-stage model SGC [30] encodes graph information with $\tilde{X}$ into an embedding matrix $P = \tilde{A}^{L_P} \cdot X$, which is then input to a Multi-Layer Perceptron $H^{(L)} = \text{MLP}(P)$. [36, 32, 25] further generalize the embedding aggregation. Such decoupled GNN is considered optimal for training efficiency, as its time complexity $O(ILnF)$ is identical to the simple MLP [31].

However, applying the decoupling technique to heterophilous GNNs is non-trivial due to the full-graph relationships. To our knowledge, LINKX [26] is the only model conceptually similar to this scheme, removing graph-related propagation during training iterations and enabling solely node-wise minibatching. It exploits a simple architecture $H^{(L)} = \text{MLP}(XW_X \| AW_A)$, where the matrix $A$ is used as an input feature in learning. The major drawback of this design is that it suffers from the $O(nF)$ term in model size and $O(mF)$ term in forward time, hindering its scalability to large graphs.

## 3 Method

In this section, we first present an overview of our $\text{LD}^2$ model, then respectively elaborate on the selection of the adjacency and feature embeddings. Lastly, an end-to-end scalable algorithm, namely $\text{A}^2\text{Prop}$, is proposed to efficiently and concurrently compute all the embeddings.

### 3.1 $\text{LD}^2$: A Decoupled Heterophilous GNN

In order to achieve superior time and memory scalability for heterophilous GNNs, we employ the concept of decoupling, which removes the dependency of graph adjacency propagation in training iterations. The main idea of our model is first generating *embeddings* from raw *features* including node attributes and adjacency in a precomputation stage. Then, these embeddings are taken as inputs to learn *representations* by a simple neural network model. We embrace the multi-channel architecture [37, 20] to enhance flexibility, where the input data is a list consisting of embedding matrices $[P_1, P_2, \cdots, P_C]$. Each embedding is separately processed and then merged in the network.

$\text{LD}^2$ utilizes diverse embeddings based on pure graph adjacency and node attributes, denoted as $P_A(A)$ and $P_X(X, A)$, respectively. Both types of embeddings can be produced by our precomputation $\text{A}^2\text{Prop}$ following Algorithm 1. The initial layer of the $\text{LD}^2$ network applies a separate linear transformation to each embedding input, and the results are concatenated to form the representation matrix. Lastly, an $L$-layer MLP is leveraged for the classification task. The high-level framework of $\text{LD}^2$ is depicted in Figure 1 and can be expressed as follow:

$$\text{Precompute}: P_A, P_X = \text{A}^2\text{Prop}(A, X); \quad \text{Transform}: H^{(L)} = \text{MLP}(P_A W_A \| P_X W_X). \quad (1)$$

**Training/Inference Complexity.** Our decoupled model design enables a simple on-demand mini-batch scheme in training and inference, that only $n_b$ rows corresponding to the batch nodes in the embedding matrices are loaded into GPU and processed by the network transformation. For $\text{LD}^2$ with $C$ channels, the GPU memory footprint is therefore bounded by $O(L_C n_b F + L_C F^2)$. It is worth noting that such complexity does not depend on the graph scale $n$ or $m$. Consequently, the training is freely configurable with an arbitrary GPU memory budget. Regarding computation operations, the time complexity of forward inference through the graph is $O(LnF^2)$, being just linear to $n$. As

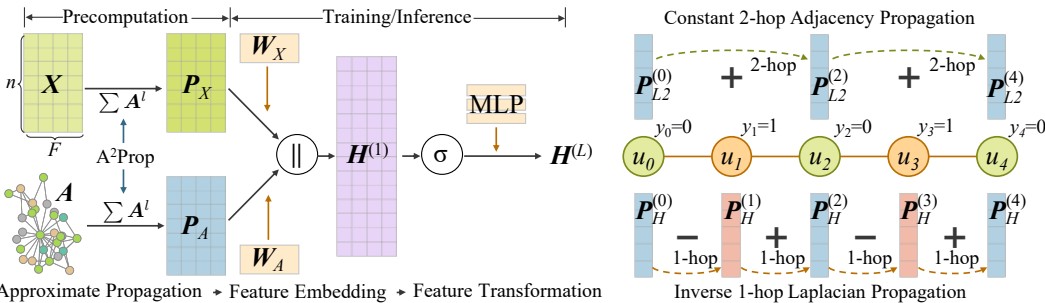

Figure 1: $\text{LD}^2$ framework: decoupled precomputation and training.  Figure 2: Propagations under heterophily.

the model complexity only contain essential operations of MLP-like transformation on nodes in the graph with no additional expense, this is the optimal scale with respect to the iterative training of GNN architectures.

## 3.2 Low-Dimensional Adjacency Embedding

Several studies reveal that, despite the feature information of nodes, the pure graph structure is equally or even more important in the context of heterophilous GNNs [9, 14, 26]. Particularly, the most informative aspects are often associated with 2-hop neighbors, i.e., "neighbors of neighbors" of ego nodes. [13] proves that even under heterophily, the 2-hop neighborhood is expected to be homophily-dominant. [38] also verifies that the 2-hop similarity is strongly relevant to GNN performance. We thence intend to explicitly model such topological information.

The 2-hop relation can be described by the 2-hop adjacency matrix $\boldsymbol{A}^2$. Note that as the sparse matrix $\boldsymbol{A}$ has $m$ entries, the number of entries in $\boldsymbol{A}^2$ is at the scale of $O(md)$, which indicates that directly applying 2-hop graph propagation in the training stage will demand even more expensive time and memory overhead to be scaled up. We instead propose an approximate scheme that seeks to prevent the 2-hop adjacency from explicit processing, and retrieves a low-dimensional but expressive embedding prior to training in the precomputation stage. In other words, we utilize the embedding to resemble 2-hop information which can be directly learned by the neural network transformation. Denote the $F$-dimensional embedding matrix as $\boldsymbol{P}_A \in \mathbb{R}^{n \times F}$. We aim to minimize its approximation error in Frobenius norm ($\| \cdot \|_F$):

$$\boldsymbol{P}_A = \underset{\boldsymbol{P} \in \mathbb{R}^{n \times F}}{\arg\min} \|\boldsymbol{A}^2 - \boldsymbol{P}\boldsymbol{P}^T\|_F^2. \tag{2}$$

The solution to Eq. (2) can be derived from the eigendecomposition of the symmetric matrix $\boldsymbol{A}^2$, that $\boldsymbol{P}_A^* = \boldsymbol{U}|\boldsymbol{\Lambda}|^{1/2}$, where $\boldsymbol{\Lambda} = \mathrm{diag}(\lambda_1, \cdots, \lambda_F)$ is the diagonal matrix with top-$F$ eigenvalues $\lambda_1 \geq \lambda_2 \geq \cdots \geq \lambda_F$, and $\boldsymbol{U} \in \mathbb{R}^{n \times F}$ is the matrix consisting of corresponding orthogonal eigenvectors. The eigenvalues are also called *frequencies* of the graph, and large eigenvalues of the adjacency matrix refer to low-frequency signals in the graph spectrum.

**Spectral Analysis.** Let $A_2(u, v)$ be the entry $(u, v)$ of matrix $\boldsymbol{A}^2$. Its diagonal degree matrix is $\boldsymbol{D}_2 = \mathrm{diag}(d_2(u_1), d_2(u_2), \cdots, d_2(u_n))$, where $d_2(u) = \sum_{v \in V} A_2(u, v)$. Denote $\boldsymbol{P}_A(u)$ as the $F$-dimensional adjacency embedding vector of node $u$. We show that the embedding matrix $\boldsymbol{P}_A^*$ defined by Eq. (2) is also the solution to the following optimization problem:

$$\boldsymbol{P}_A = \underset{\boldsymbol{P} \in \mathbb{R}^{n \times F}, \boldsymbol{P}^\top \boldsymbol{D}_2 \boldsymbol{P} = \boldsymbol{\Lambda}}{\arg\min} \sum_{u, v \in V} A_2(u, v)\|\boldsymbol{P}(u) - \boldsymbol{P}(v)\|^2. \tag{3}$$

This is because $\sum_{u, v} A_2(u, v)\|\boldsymbol{P}(u) - \boldsymbol{P}(v)\|^2 = 2\sum_u d_2(u)\|\boldsymbol{P}(u)\|^2 - 2\sum_{u, v} A_2(u, v)\boldsymbol{P}(u)\boldsymbol{P}(v) = 2\,\mathrm{tr}(\boldsymbol{P}^\top \boldsymbol{D}_2 \boldsymbol{P} - \boldsymbol{P}^\top \boldsymbol{A}^2 \boldsymbol{P})$. As $\boldsymbol{P}^\top \boldsymbol{D}_2 \boldsymbol{P}$ is fixed, finding the minimum of Eq. (3) is equivalent to optimizing $\max_{\boldsymbol{P}} \boldsymbol{P}^\top \boldsymbol{A}^2 \boldsymbol{P}$, of which the solution is exactly $\boldsymbol{P}_A^*$ according to the property of eigenvectors. Equation (3) implies that, 2-hop neighbors $(u, v), t \in \mathcal{N}(u), v \in \mathcal{N}(t)$ in the graph will share similar embeddings $\boldsymbol{P}_A(u)$ and $\boldsymbol{P}_A(v)$.

In fact, the low-dimensional embedding $\boldsymbol{P}_A^*$ can be interpreted as the adjacency spectral embedding of the 2-hop graph $\boldsymbol{A}^2$. Graph spectral embedding is a technique concerning the low-frequency spectrum of a graph, and is employed in tasks such as graph clustering [39]. As $\boldsymbol{P}_A$ corresponds to the dominant eigenvalues of $\boldsymbol{A}^2$, the embedding provides an approximate representation of the 2-hop neighborhoods based on the overall graph topology. Alternatively, if we regard the adjacency information solely as features input into the network like LINKX, $\boldsymbol{P}_A$ correlates to the uncentered principal components of matrix $\boldsymbol{A}$. Therefore, learning a linear transformation $\boldsymbol{P}_A \boldsymbol{W}_A$ with weight matrix $\boldsymbol{W}_A \in \mathbb{R}^{F \times F}$ in LD$^2$ is the same expressive as the rank-$F$ approximation of $\boldsymbol{A}\boldsymbol{W}_{A0}$ in LINKX, where $\boldsymbol{W}_{A0} \in \mathbb{R}^{n \times F}$, but with a less computational cost independent to the graph scale. Compared to other works attempting to generate graph embeddings based on graph geometric or similarity measures [40, 41, 14, 16, 19, 42], our approach offers the advantages of lower dimensionality and efficient calculation as demonstrated in Section 3.4.

## 3.3 Long-Distance Feature Embedding

Decoupling the node features through approximate propagation has been extensively studied in regular GNNs with various schemes [28, 30, 36, 29, 43, 32, 44]. Nonetheless, these approaches are

based on the homophily assumption and focus on local neighborhoods. In order to apply decoupled propagation to heterophilous graphs and exploit the multi-channel ability of our model, we formulate the general form of approximate propagation as the weighted sum of powers of a propagation matrix applied to the input feature, i.e., $\boldsymbol{P}_X = \sum_{l=1}^{L_P} \theta_l \boldsymbol{T}^l \boldsymbol{X}$. Examples of propagation matrix $\boldsymbol{T}$ include $\tilde{\boldsymbol{A}}$ and $\tilde{\boldsymbol{L}}$, which respectively correspond to aggregative and discriminative operations.

LD$^2$ utilizes the following channels jointly as input embeddings: (1) *inverse* summation of 1-hop improved Laplacian propagations $\boldsymbol{P}_{X,H} = \frac{1}{L_{P,H}} \sum_{l=1}^{L_{P,H}} (\boldsymbol{I} + \tilde{\boldsymbol{L}})^l \boldsymbol{X}$, ($\theta_l = 1$, $\boldsymbol{T} = \boldsymbol{I} + \tilde{\boldsymbol{L}}$); (2) *constant* summation of 2-hop adjacency propagations $\boldsymbol{P}_{X,L2} = \frac{1}{L_{P,L2}} \sum_{l=1}^{L_{P,L2}} \bar{\boldsymbol{A}}^{2l} \boldsymbol{X}$, ($\theta_l = 1$, $\boldsymbol{T} = \bar{\boldsymbol{A}}^2$); (3) raw node attributes $\boldsymbol{P}_{X,0} = \boldsymbol{X}$.

Intuitively, the first two channels perform distinct topology-based propagations on node feature $\boldsymbol{X}$, and employ inverse or constant summation to aggregate multi-hop information, in contrast to the local *decaying* summation ($\theta_l \to 0$ when $l \to \infty$) commonly adopted in homophilous GNNs. Hence, such summations are suitable for retrieving long-range information under heterophily. The raw matrix $\boldsymbol{X}$ is also directly used as one input channel to depict node identity, which is a ubiquitous practice known as the skip connection or all-pass filter in heterophilous GNNs [12, 45, 20, 22].

Illustrated in Figure 2, the inverse embedding $\boldsymbol{P}_{X,H}$ is based on the intuition that, as neighbors tend to be different from the ego node, their features are also dissimilar. Hence in propagation, the embedding of the ego node should contain the previous embedding of itself, as well as the inverse of adjacent embeddings, which is exactly the interpretation of propagating node features by graph Laplacian matrix $\tilde{\boldsymbol{L}} = \boldsymbol{I} - \tilde{\boldsymbol{A}}$, while an additional identity matrix is applied to balance the embedding distribution. The second embedding $\boldsymbol{P}_{X,L2}$ performs a 2-hop propagation through the graph and aggregates the results of multi-scale neighbors. It echoes the earlier statement on the importance of 2-hop adjacency from the feature aspect. Note that for $\boldsymbol{P}_{X,L2}$, the employed adjacency matrix is $\bar{\boldsymbol{A}}$ which escapes self-loops, since it is shown to be relatively favorable for capturing non-local homophily in multi-hop propagations compared with $\tilde{\boldsymbol{A}}$ [13, 23].

**Spectral Analysis.** Assume that $\|\boldsymbol{X}(u)\| = \|\boldsymbol{P}(u)\| = 1$. We first examine the following regularization problem optimizing the embedding $\boldsymbol{P}$ based on input $\boldsymbol{X}$ for homophilous graphs [43]:

$$\boldsymbol{P}_{X,L} = \underset{\|\boldsymbol{P}(u)\|=1, \forall u \in V}{\arg\min} \sum_{u,v \in V} A(u,v) \|\boldsymbol{P}(u)/d^a(u) - \boldsymbol{P}(v)/d^b(v)\|^2 + \|\boldsymbol{P} - \boldsymbol{X}\|_F^2. \quad (4)$$

Differentiating the objective function with respect to $\boldsymbol{P}$ leads to $(\boldsymbol{I} - \tilde{\boldsymbol{A}})\boldsymbol{P} - \boldsymbol{X} = 0$. Therefore the closed-form solution is $\boldsymbol{P}_{X,L}^* = (\boldsymbol{I} - \tilde{\boldsymbol{A}})^{-1} \boldsymbol{X} = \sum_{l=0}^{\infty} \tilde{\boldsymbol{A}}^l \boldsymbol{X}$. In practical implementation, a limited $L_{P,L}$-hop summation is used instead due to the over-smoothing issue that the infinite form converges to identical node-wise embeddings. This Markov diffusion kernel $\boldsymbol{P}_{X,L} = \frac{1}{L_{P,L}} \sum_{l=0}^{L_{P,L}} \tilde{\boldsymbol{A}}^l \boldsymbol{X}$ is investigated in S$^2$GC [43] as an approach for balancing locality and multi-hop propagation, functioning as a low-pass filter to the signal $\boldsymbol{X}$ but also preserves high frequency. Its interpretation can be observed from Eq. (4), that it simultaneously minimizes the embedding difference of neighboring nodes as well as the approximation closeness to the input feature $\boldsymbol{X}$.

To obtain the channel $\boldsymbol{P}_{X,L2}$ used in LD$^2$, we introduce the low-frequency regularization preferably to 2-hop adjacency in the graph, as 1-hop neighbors exhibit heterophily. Therefore, replacing $\tilde{A}(u,v)$ in Eq. (4) with $\bar{A}_2(u,v)$ yields our constant 2-hop embedding $\boldsymbol{P}_{X,L2}$. It shares similar spectral properties with S$^2$GC for acting as a low-pass filter in 2-hop neighborhoods, while maintaining certain long-distance knowledge thanks to the multi-scale aggregation. The other channel in feature embedding, i.e. the Laplacian propagation, can be derived as $\boldsymbol{P}_{X,H} = (\boldsymbol{I} + \tilde{\boldsymbol{L}})(\boldsymbol{I} - \tilde{\boldsymbol{A}})^{-1} \boldsymbol{X} = (\boldsymbol{I} + \tilde{\boldsymbol{L}}) \boldsymbol{P}_{X,L}$. Based on the above analysis, the embedding $\boldsymbol{P}_{X,L}$ contains multi-hop neighborhood information, while $(\boldsymbol{I} + \tilde{\boldsymbol{L}})$ can be seen as the improved Laplacian operator extracting the high-frequency components. The embedding $\boldsymbol{P}_{X,H}$ thus serves as a high-pass filter focusing on discriminative structures in a non-local manner. In terms of spatial domain interpretation, such high-frequency information corresponds to the fine-grained embedding differences between two nodes [18, 46]. It is noticeable that these three channels $\boldsymbol{P}_{X,L2}, \boldsymbol{P}_{X,H}, \boldsymbol{P}_{X,0}$ respectively represent low-pass, high-pass, and all-pass propagations through the graph while addressing heterophily. Combining them as inputs to the neural network benefits the model performance with expressive information at various distances including identity, local, and global perspectives.

## 3.4 Approximate Adjacency Propagation Precomputation

Conventionally, calculating the graph propagation $\tilde{A} \cdot P$ for an arbitrary feature matrix $P$ is conducted by the sparse-dense matrix multiplication. However, such an approach does not recognize the property of the adjacency matrix $\tilde{A}$, that it can be represented by the adjacency list of nodes, and non-zero values in its data are solely determined by node degrees. Furthermore, since the propagation result is subsequently processed by the neural network, it is not necessary to be precise as the model is robust to handle noisy data [46, 47]. We first define the precision bound for approximate embedding:

**Definition 3.1** (**Approximate Vector Embedding**). *Given a relative error bound $0 < \epsilon < 1$, a norm threshold $\delta > 0$, and a failure probability $0 < \phi < 1$, the estimation $\hat{P}(u)$ for an arbitrary embedding vector $P(u)$ should satisfy that, for each $u \in V$ with $\|P(u)\| > \delta$, $\|P(u) - \hat{P}(u)\| \le \epsilon \cdot \|P(u)\|$ with probability at least $1 - \phi$.*

Graph power iteration algorithm is the variant of power iteration particularly applied for calculating powers of adjacency matrix $A$ [48]. In essence, the algorithm can be derived by maintaining a *residue* $R^{(l)}(u)$ that holds the current $l$-hop propagation results for each node, and iteratively updating the next-hop residues of neighboring nodes $R^{(l+1)}(v), v \in \mathcal{N}(u)$ for all nodes $u$. For each iteration, the *reserve* $\hat{P}^{(l)}$ is also added up and converges to an underestimation of $P$.

We propose Algorithm 1 for our specific scenario, namely Approximate Adjacency Propagation (A$^2$Prop). Based on power iteration, our algorithm is greatly generalized to accommodate normalized adjacency, feature vectors for nodes, and a limited number of hops. We show that the algorithmic output can be bounded by Definition 3.1. For $L_P$ iterations, denote the acceptable error per entry for push as $\delta_P$, the matrix-wise absolute error is $\|P - \hat{P}\|_{1,1} \le \sum_{f=1}^{L_P} \sum_{f=1}^{F} \sum_{u \in V} d(u)\delta_P = L_P mF\delta_P$. By setting $\delta_P = \epsilon\delta/L_P m$, the estimation $\hat{P}$ satisfies Definition 3.1.

**Approximate Feature Embedding.** The feature embedding formed as $P_X = \sum_{l=0}^{L_P} \theta_l T^l X$ can be computed by iteratively applying graph power iterations to the initial residue $R^{(0)} = X$. The implicit propagation behavior is described by matrix $T$. For example, for Laplacian propagation $T = I + \tilde{L}$ to node $u$, the embeddings from the previous iteration are aggregated as $R^{(l+1)}(u) = 2R^{(l)}(u) - \sum_{v \in \mathcal{N}(u)} R^{(l)}(v)/d^a(u)d^b(v) = \sum_{v \in \mathcal{N}(u) \cup \{u\}} \frac{\alpha_L(u,v)}{d^a(u)d^b(v)} \cdot R^{(l)}(v)$. Here $\alpha_T(u,v)$ is a propagation factor for unifying the aggregation by $T$, that $\alpha_L(u,u) = 2d^{a+b}(u), \alpha_L(u,v) = -1, v \in \mathcal{N}(u)$. For propagation $\tilde{A}$ and $\bar{A}$, the factor is $\alpha_A(u,v) = 1$ and $\alpha_A(u,u) = 1, 0$, respectively.

In each iteration $l$, the reserve is updated after propagation according to the coefficient $\theta_l$ to sum up corresponding embeddings. Intuitively, one multiplication of $\bar{A}^2$ is equivalent to two iterations of $\bar{A}$ propagation. Hence for $P_{X,L2}$ there is $\theta_l = l \bmod 2 = 0, 1, 0, 1, \cdots$ under the summation scheme

---

**Algorithm 1** A$^2$Prop: Approximate Adjacency Propagation

**Input:** graph $G$, feature matrix $X$, max hop $L_P$, normalization factor $a, b$, propagation factor $\alpha_T$, summation factor $\theta_l$, push threshold $\delta_P$
**Output:** adjacency embedding $P_A$, feature embedding $P_X$

1   $R_A^{(0)} \leftarrow N(0,1)$, $R_X^{(0)} \leftarrow X$
2   **for** $l$ from 0 to $L_P - 1$ **do**
3     $R_A^{(l+1)} \leftarrow 0$, $R_X^{(l+1)} \leftarrow 0$
4     **for all** $u \in V$ such that $\|R^{(l)}(u)\| > \delta_P$ **do**
5       **for all** $v \in \mathcal{N}(u) \cup \{u\}$ **do**
6         $R_A^{(l+1)}(v) \leftarrow R_A^{(l+1)}(v) + \alpha_A(u,v) \cdot R_A^{(l)}(u)$
7         $R_X^{(l+1)}(v) \leftarrow R_X^{(l+1)}(v) + \frac{\alpha_T(u,v)}{d^a(v)d^b(u)} \cdot R_X^{(l)}(u)$
8     **if** $l \bmod 2 = 1$ and $l < L_P - 1$ **then**
9       $P_A \leftarrow \texttt{orthonormalize}(R_A^{(l)})$
10    $P_X \leftarrow P_X + \theta_l \cdot R_X^{(l)}$
11    empty $R_A^{(l)}, R_X^{(l)}$
12  $P_A \leftarrow P_A \cdot |(R_A^{(L_P)})^\top \cdot P_A|^{1/2}$
13  $P_X \leftarrow P_X + \theta_{L_P} \cdot R_X^{(L_P)}$
14  **return** $P_A, P_X$

(a) genius

(b) pokec

Figure 3: Effect of embedding channels and propagation hops on accuracy.

in Algorithm 1. Since all embeddings we consider are constant, that is, $\theta_l \in \{0, 1\}$, the reserve can be simply increased without the rescaling terms in more general cases such as [32].

**Approximate Adjacency Embedding.** The adjacency embedding is represented by leading eigenvectors $P_A = U|\Lambda|^{1/2}$. This eigendecomposition of $A^2$ can be solved by the truncated power iteration [49]: Initialize the $n \times F$ residue by i.i.d. Gaussian noise $R^{(0)} = N(0, 1)$. For each iteration $l$, firstly multiply the residue by $A^2$ as $R^{(l+1)} = A^2 R^{(l)}$; then, perform column-wise normalization to the residue $\texttt{orthonormalize}(R^{(l+1)})$ so that its columns are orthogonal to each other and of L2 norm 1. After convergence, the matrix satisfies $A^2 R^{(L_P)} = R^{(L_P)} \Lambda$ within the error bound, which leads to the estimated output $\hat{U} = R^{(L_P)}, \hat{P}_A = \hat{U}|\hat{\Lambda}|^{1/2}$.

Similarly, the 2-hop power iteration of $P_A$ can be merged with those for $P_X$ with a shared maximal iteration $L_P$, and orthonormalization is conducted every two $A$ iterations. When the algorithm converges with error bound $\delta$, the number of iteration follows $L_P = O(\log(F/\delta)/(1 - |\lambda_{F+1}/\lambda_F|))$. By selecting proper values for $F$ and $\delta$, the algorithm produces satisfying results within $L_P$ iterations.

**Precomputation Complexity.** Since A²Prop serves as a general approximation for various adjacency-based propagations, the computation of all feature channels can be performed simultaneously in a single run. The memory overhead of the algorithm is mainly the residue and reserve matrices for $C$ embedding channels, which is $O(CnF)$ in total. Note that A²Prop precomputation is performed in the main memory, and benefits from a less-constrained budget compared to GPU memory.

For each A²Prop iteration, neighboring connections are accessed for at most $m$ times. The time complexity of Algorithm 1 can thus be bounded by $O(L_P mF)$. In addition, its loops over nodes and features can be parallelized and vectorized to reduce execution time. Moreover, the graph power iteration design is also amendable for further enhancements, such as reduction to sub-linear complexity [50, 25], better memory utilization [51, 52], and precision-efficiency trade-offs [53, 54]. We leave these potential improvements on A²Prop for future work.

## 4 Experimental Evaluation

We implement the LD² model and evaluate its performance from the perspectives of both efficacy and scalability. In this section we highlight key empirical results compared to minibatch GNNs on large-scale heterophilous graphs, while parameter settings, further experiments, and subsequent discussions can be found in the Appendix.

### 4.1 Experiment Setting

**Datasets.** We mainly perform experiments on million-scale and above heterophilous datasets [26, 55] for the transductive node classification task, with the largest available graph wiki ($m = 243M$) included. Evaluations on more homophilous and heterophilous graphs can be found in Appendices D and E. We leverage settings as per [26] such as the random train/test splits and the induced subgraph testing for GSAINT-sampling models.

**Baselines.** We focus on GNN models applicable to *minibatch* training in our evaluation regarding scalability, and hence most *full-batch* networks mentioned in Section 2 are excluded in the main experiments, while more comprehensive results for both minibatch and full-batch models are in Appendices D and E. Conventional baselines in the main experiments include MLP which only processes node attributes without considering graph topology, as well as PPRGo [29] and SGC [30] representing decoupled schemes for traditional graph propagation. For GNNs under non-homophily, we investigate GCNJK-GS [21] and MixHop-GS [12], where GSAINT random walk sampling [27] is utilized to empower the original backbone models for minibatching. LINKX is the decoupled heterophilous GNN proposed by [26]. Simple i.i.d. node batching is adopted for decoupled networks. Explorations on the model settings are displayed in Appendix C.

**Evaluation Metrics.** We uniformly use classification accuracy on the test set to measure network effectiveness. Note that since the datasets are updated and the minibatch scheme is employed, results may be different from their original works. In order to evaluate scalability performance, we conduct repeated experiments and record the network training/inference time and peak memory footprint as efficiency metrics. For precomputed methods, we consider the learning process combining both

Table 2: Average test accuracy (%) of minibatch LD$^2$ and baselines on heterophilous datasets. "> 12h" means the model requires more than 12h clock time to produce proper results. Respective results of the first and second best performances on each dataset are marked in **bold** and underlined fonts.

| Dataset | genius | tolokers | arxiv-year | penn94 | twitch-gamers | pokec | snap-patents | wiki |
|---|---|---|---|---|---|---|---|---|
| Nodes $n$ | 421,858 | 11,758 | 169,343 | 41,536 | 168,114 | 1,632,803 | **2,738,035** | 1,770,981 |
| Edges $m$ | 922,864 | 1,038,000 | 1,157,799 | 1,362,220 | 6,797,557 | 22,301,964 | 13,967,949 | **242,507,069** |
| $F$ / $N_c$ | 12 / 2 | 10 / 2 | 128 / 5 | 4,814 / 2 | 7 / 2 | 65 / 2 | 269 / 5 | 600 / 5 |
| MLP | 82.47 ±0.06 | 73.38 ±0.25 | 37.23 ±0.31 | 74.41 ±0.48 | 61.26 ±0.19 | 61.81 ±0.07 | 23.03 ±1.48 | 35.64 ±0.10 |
| PPRGo | 79.81 ±0.00 | 78.16 ±0.00 | 39.35 ±0.12 | 58.75 ±0.31 | 47.19 ±2.26 | 50.61 ±0.04 | (>12h) | (>12h) |
| SGC | 79.85 ±0.01 | 71.16 ±0.06 | 43.40 ±0.16 | 68.31 ±0.27 | 57.05 ±0.21 | 56.58 ±0.06 | 37.70 ±0.06 | 28.12 ±0.08 |
| GCNJK-GS | 80.65 ±0.07 | 74.41 ±0.73 | 48.26 ±0.64 | 65.91 ±0.16 | 59.91 ±0.42 | 59.38 ±0.21 | 33.64 ±0.05 | 42.95 ±0.39 |
| MixHop-GS | 80.63 ±0.04 | 77.47 ±0.40 | 49.26 ±0.16 | 75.00 ±0.37 | 61.80 ±0.00 | 64.02 ±0.02 | 34.73 ±0.15 | 45.52 ±0.11 |
| LINKX | 82.51 ±0.10 | 77.74 ±0.13 | **50.44** ±0.30 | **78.63** ±0.25 | 64.15 ±0.18 | 68.64 ±0.65 | 52.69 ±0.05 | 50.59 ±0.12 |
| **LD$^2$ (ours)** | **85.31** ±0.06 | **79.76** ±0.26 | 50.29 ±0.11 | 75.52 ±0.10 | **64.33** ±0.19 | **74.93** ±0.10 | **58.58** ±0.34 | **52.91** ±0.16 |

Table 3: Time and memory overhead of LD$^2$ and baselines on large-scale datasets. "Learn", "Infer", and "Mem." respectively refer to minibatch learning and inference time (s) and peak GPU memory (GB). Precomputation time is appended when applicable. "> 12h" means the model requires more than 12h clock time to produce proper results. Respective results of the first and second best performances among heterophilous models per metric are marked in **bold** and underlined fonts.

| Dataset | twitch-gamers | | | pokec | | | snap-patents | | | wiki | | |
|---|---|---|---|---|---|---|---|---|---|---|---|---|
| | Learn | Infer | Mem. | Learn | Infer | Mem. | Learn | Infer | Mem. | Learn | Infer | Mem. |
| MLP | 6.36 | 0.02 | 0.61 | 47.86 | 0.11 | 13.77 | 27.39 | 0.28 | 9.33 | 133.55 | 0.62 | 18.15 |
| PPRGo | 10.46+15.88 | 0.41 | 9.64 | 121.95+56.11 | 2.69 | 3.82 | (>12h) | | | (>12h) | | |
| SGC | 0.09+0.74 | 0.01 | 0.28 | 1.05+8.08 | 0.01 | 0.28 | 4.94+23.54 | 0.01 | 0.42 | 12.66+7.98 | 0.01 | 0.52 |
| GCNJK-GS | 71.48 | 0.02* | 7.33 | 27.33 | 0.09* | 9.03 | 19.02 | 0.23* | 9.21 | 95.52 | 0.69* | 16.36 |
| MixHop-GS | 52.12 | 0.01* | 1.49 | 71.35 | 0.03* | 12.91 | 45.24 | 0.16* | 19.58 | 84.22 | 0.23* | 16.28 |
| LINKX | 10.99 | 0.19 | 2.35 | 28.77 | 0.33 | 9.03 | 39.80 | 0.22 | 21.53 | 180.71 | 1.14 | 14.53 |
| **LD$^2$ (ours)** | 0.85+**1.96** | **0.01** | **1.44** | 17.95+**6.18** | **0.01** | **3.82** | 31.32+**6.96** | **0.02** | **3.96** | 28.12+**6.50** | **0.01** | **4.47** |

* Inference time of GSAINT sampling is less precise since they are conducted on induced subgraphs smaller than the raw graph.

precomputation and training. Evaluations are conducted on a machine with 192GB RAM, two 28-core Intel Xeon CPUs (2.2GHz), and an NVIDIA RTX A5000 GPU (24GB memory).

## 4.2 Performance Comparison

The main evaluations of LD$^2$ and baselines on 8 large heterophilous datasets are presented in Tables 2 and 3 for effectiveness and efficiency metrics, respectively. As an overview, our model demonstrates its scalability in completing training and inference with fast running speed and efficient memory utilization, especially on large graphs. At the same time, it achieves comparable or superior prediction accuracy against the state-of-the-art minibatch heterophilous GNNs on most datasets.

**Time Efficiency.** We first highlight the scalability performance of our LD$^2$ model. Specifically, compared to heterophilous benchmarks on the four largest graphs with million-scale data, LD$^2$ speeds up the minibatch training process by 3–15 times, with an acceptable precomputation cost. Its inference time is also consistently below 0.1 seconds. The outstanding efficiency of LD$^2$ is mainly attributed to the simple model architecture that removes graph-scale operations while ensuring rapid convergence. In contrast, the execution speeds of MixHop and LINKX are highly susceptible to node and edge sizes, given their design dependency on the entire input graph. The extensive parameter space also causes them to converge slower, necessitating relatively longer training times. PPRGo shows limited scalability due to the costly post-transformation propagation. The superiority of LD$^2$ efficiency even holds when compared to simple methods such as MLP and SGC, indicating that the model is favorable for incorporating extra heterophilous information with no significant additional overhead. The empirical results affirm that LD$^2$ exhibits optimized training and inference complexity at the same level as simple models.

**Memory Footprint.** LD$^2$ remarkably reduces run-time GPU memory consumption. As the primary overhead only comprises the model parameters and batch representations, it enables flexible configuration of the model size and batch size to facilitate powerful training. Even for the largest graph

wiki with $n = 1.77M$ and $F = 600$, the footprint remains below 5GB under our hyperparameter settings. Other heterophilous GNNs, though adopting the minibatch scheme, experience high memory requirements and even occasionally encounter out-of-memory errors during experiments, as their space-intensive graph propagations are executed on the GPU. Consequently, when the graph scales up, they can only be applied with highly constrained model capacities to conserve space, potentially resulting in compromised performance.

**Test Accuracy.** With regard to efficacy, $LD^2$ achieves top testing accuracy on 6 out of 8 heterophilous graphs and comparable performance on the remaining ones. It also consistently outperforms the sampling-based GCNJK and MixHop, as well as conventional GNNs. Particularly, by extracting embeddings from not only node features but pure graph topology as well, $LD^2$ obtains significant improvements over feature-based networks on datasets such as genius, snap-patents, and wiki, demonstrating the importance of pure graph information in heterophilous learning. We deduce that the relatively suboptimal accuracy on penn94 may be correlated with the difficulty of fitting one-hot encoding features into informative embeddings, as explored in Appendix G. Consistent with the previous studies [26], regular GNN baselines suffer from performance loss on most heterophilous graphs, while MLP achieves comparably high accuracy when node attributes are discriminative enough. For non-homophilous models GCNJK and MixHop, the minibatch scheme hinders them from reaching higher results because of the neglect of their full-graph relationships.

### 4.3 Effect of Parameters

To gain deeper insights into the multi-channel embeddings of $LD^2$, in Figure 3 we explore the effect of embeddings channels and propagation hops which are critical to our model design, while more discussions on other parameters and factors are displayed in Appendix F.

**Embedding Channels.** Lines in Figure 3 represent the results of learning on separate inputs on two representative datasets genius and pokec. It can be observed that different graphs imply varying patterns when embedding channels and propagation hops are changed. For the genius dataset where raw node attributes already achieve an accuracy above 82%, applying the other two feature embeddings further improves the result. While the adjacency embedding alone shows secondary performance, integrating it with other channels proves beneficial. In comparison, on pokec, it is the inverse embedding $\boldsymbol{P}_{X,H}$ that becomes the key contributor. The empirical evaluation supports our design that by adopting multi-channel and heterophily-oriented embeddings, $LD^2$ benefits from learning both topology and feature for a more comprehensive understanding of the graph data.

**Propagation Hops.** As elaborated in Section 3.4, propagation hops $L_P$ determines the number of iterations in Algorithm 1. Particularly, for the approximate adjacency embedding $\boldsymbol{P}_A$, it also affects the convergence of decomposition. As shown by the brown dashdotted lines in Figure 3, the accuracy typically becomes stable when $L_P > 8$, indicating the utility of the low-dimensional approximation in producing effective topology embedding within limited iterations. For the multi-channel scheme in general, as the graph scale increases, employing more propagation hops becomes advantageous in capturing distant information. Aligned with our analysis, above observation validates that $LD^2$ is powerful in capturing implicit information of various frequencies and scales that is important in the presence of heterophily.

## 5 Conclusion

In this work, we propose $LD^2$, a scalable GNN design for heterophilous graphs, that leverages long-distance propagation to capture non-local relationships among nodes, and incorporates low-dimensional yet expressive embeddings for effective learning. The model decouples full-graph dependency from the iterative training, and adopts an efficient precomputation algorithm for approximating multi-channel embeddings. Theoretical and empirical evidence demonstrates its optimized training characteristics, including time efficiency with a complexity linear to $O(n)$, and GPU memory independence from the graph size $n$ and $m$. As a noteworthy result, $LD^2$ successfully applies to million-scale datasets under heterophily, with learning times as short as 1 minute and GPU memory expense below 5GB. We also recognize the current limitations of our work including potential accelerations for precomputation and adaptability to diverse feature patterns. Detailed limitations and broader impacts are addressed in the Appendix.

## Acknowledgments and Disclosure of Funding

This research is supported by Singapore MOE AcRF Tier-2 funding (MOE-T2EP20122-0003), NTU startup grant (020948-00001), and the Joint NTU-WeBank Research Centre on FinTech. Xiang Li is supported by National Natural Science Foundation of China No. 62202172 and Shanghai Science and Technology Committee General Program No. 22ZR1419900. Jieming Shi is supported by Hong Kong RGC ECS No. 25201221 and National Natural Science Foundation of China No. 62202404. We also thank Yuyuan Song for contributing to the experiments in this paper.

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
