# A   Detailed Theoretical Analysis

## A.1   Graph Spectrum

Consider an undirected graph $G = (V, E)$ whose adjacency matrix is symmetric. Following notations in Section 2, we denote the eigendecomposition of the normalized graph adjacency and Laplacian matrices respectively as $\tilde{A} = UMU^\top$ and $\tilde{L} = VNV^\top$, where $M = \mathrm{diag}(\mu_1, \cdots, \mu_n)$, $|\mu_1| \geq |\mu_2| \geq \cdots \geq |\mu_n|$, $N = \mathrm{diag}(\nu_1, \cdots, \nu_n)$, $0 = \nu_1 < \nu_2 \leq \cdots \leq \nu_n$, and $U, V$ are the matrices of corresponding eigenvectors. We also immediately have $\tilde{A}^2 = UM^2U^\top = U\Lambda U^\top$, $\lambda_f = |\mu_f|^2$.

Intuitively, since $\tilde{L} = I - \tilde{A}$, the leading eigenvalues $\mu_1, \mu_2, \cdots$ of $\tilde{A}$ correspond to the smallest of those $\nu_1, \nu_2, \cdots$ of $\tilde{L}$. These eigenvalues are known as the low-frequency spectrum of the graph that correlates to graph connectivity. Specially, $\nu_2 > 0$ if and only if the graph is connected, which is our case. Similarly, small values of $\mu_f$ and large values of $\nu_i$ represent the high frequency part of the graph. Graph spectrum is a graph invariant despite the status of node labels.

We plot the spectrum computed by A$^2$Prop of 7 heterophilous graphs and 2 homophilous graphs in Figure 4, which shows the leading $k$ eigenvalues $\mu_1, \mu_2, \cdots, \mu_k$. The figure indicates the importance of high-frequency information in graphs under heterophily. For appropriately large heterophilous graphs penn94, arxiv-year, genius, and twitch-gamers, the spectrum converges slowly to 0. In other words, even high-frequency parts with large $f$ still exhibit relatively large eigenvalues compared to the dominant $\mu_1$. In contrast, the homophilous protein and reddit are significant in low frequency of a few eigenvalues. Hence, during the iterative propagation, the topology information carried by these high-frequency components can be preserved to enhance performance.

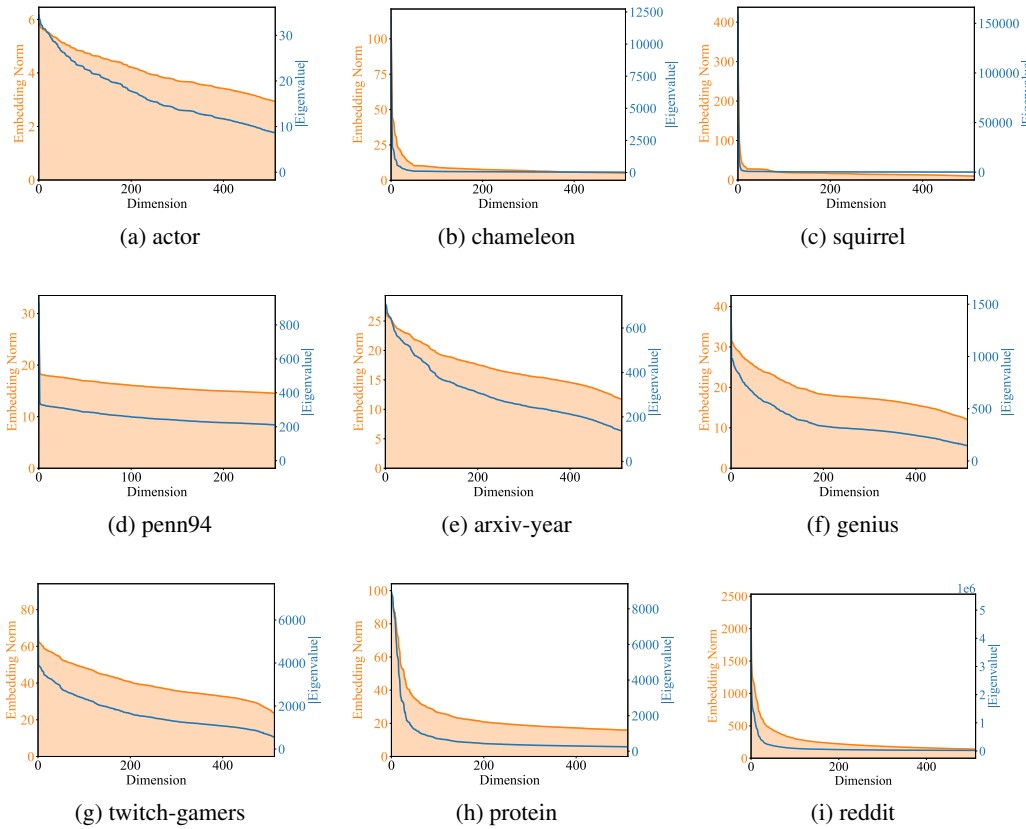

Figure 4: Spectrum and adjacency embedding distribution of heterophilous and homophilous graphs. Blue lines are the magnitude of leading adjacency eigenvalues. Yellow areas are the norm value distributions of adjacency embeddings.

## A.2 Graph Signal Filter

Given an arbitrary signal, i.e. a feature matrix $\boldsymbol{X}$, the graph Fourier transform is $\hat{\boldsymbol{X}} = \boldsymbol{V}^\top \boldsymbol{X}$, and the inverse transform is $\boldsymbol{X} = \boldsymbol{V}\hat{\boldsymbol{X}}$. A convolutional kernel $f$ is an operator applied to conduct the transform. The graph convolution is defined for feature vector $x$ as:

$$f * x = \boldsymbol{V}\langle \boldsymbol{V}^\top f, \boldsymbol{V}^\top x\rangle = \boldsymbol{V}g(\boldsymbol{N})\boldsymbol{V}^\top x,$$

where $g(\boldsymbol{N})$ denotes the corresponding filter in spectral domain.

To parameterize the filter, it is often expressed as an $L_p$-degree polynomial on the input eigenvalue $\nu$:

$$g(\nu) = \sum_{l=0}^{L_P} \theta_l \nu^l.$$

And the spectral filter is:

$$g(\tilde{\boldsymbol{L}}) = \sum_{l=0}^{L_P} \theta_l \boldsymbol{V} \nu^l \boldsymbol{V}^\top = \sum_{l=0}^{L_P} \theta_l \tilde{\boldsymbol{L}}^l. \tag{5}$$

Hence, a spectral filtering process to signal $\boldsymbol{X}$ is equivalent to multiplying the polynomial filter $g(\tilde{\boldsymbol{L}})\boldsymbol{X}$, and vice versa. Note that since $\tilde{\boldsymbol{L}} = \boldsymbol{I} - \tilde{\boldsymbol{A}}$, the filter can be equivalently expressed by $g(\tilde{\boldsymbol{A}})$ with respect to $\tilde{\boldsymbol{A}}$. We use the term filter in both cases interchangeably when there is no confusion.

Common GNN operations can thus be interpreted as an approach of graph signal processing. For example, the vanilla GCN [1] propagates by $\tilde{\boldsymbol{A}}$, which indicates $\theta_0 = 1, \theta_1 = -1, \theta_l = 0, l > 2$ and $g(\boldsymbol{N}) = \boldsymbol{I} - \boldsymbol{N}$. It strengthens the low frequency value $\nu_0$ and suppresses $\nu_1$. In comparison, FAGCN [18] defines its high-frequency filter as $\epsilon \boldsymbol{I} - \tilde{\boldsymbol{A}}$, or equivalently, $\theta_0 = \epsilon - 1, \theta_1 = 1, \theta_l = 0, l > 2$. It is shown by the paper that the combination of low-pass, high-pass, and all-pass filters is more expressive under heterophily.

## A.3 Iterative and Decoupled GNN

For general GNN architecture with iterative propagation, each of its layer applies the signal filtering as:

$$\boldsymbol{H}^{(l+1)} = \sigma(g(\tilde{\boldsymbol{L}})\boldsymbol{H}^{(l)}\boldsymbol{W}^{(l)}), \ \boldsymbol{H}^{(0)} = \boldsymbol{X}.$$

To simplify the process, assume the transformation is linear. Then all the weight transformations can be compressed together as $\boldsymbol{W}$, and

$$\boldsymbol{H}^{(L)} = g^L(\tilde{\boldsymbol{L}})\boldsymbol{X}\boldsymbol{W}.$$

The filter $g^L(\tilde{\boldsymbol{L}})$ can be similarly expressed by a polynomial with proper $L_P$ and $\theta_l$. Thus the decoupled GNN is simply replacing the weight multiplication with MLP transformation:

$$\boldsymbol{H}^{(L)} = \mathrm{MLP}(\boldsymbol{P}), \ \boldsymbol{P} = g(\tilde{\boldsymbol{L}})\boldsymbol{X}.$$

Despite its simplicity, the formulation of decoupled GNN completely preserves the spectral filter used in iterative GNNs. Hence their expressiveness in processing graph signals with polynomial filters is the same, and the difference only lies in the neural network transformation. In fact, [56] proves that such design is sufficient to node discrimination tasks under mild conditions.

## A.4 Interpretation of LD$^2$ Filters

The spectral interpretation of the adjacency and feature channels utilized in Sections 3.2 and 3.3 is clearer under Eq. (5). For the inverse Laplacian filter $\sum_{l=1}^{L_{P,H}} \tilde{\boldsymbol{L}}^l$, there is the finite geometric series $\theta_0 = 0, \theta_1 = \cdots = \theta_{L_P} = 1$. Hence, it is able to preserve up to $L_P$ high-frequency components, while stopping the low frequency by $\theta_0 = 0$.

For the constant 2-hop adjacency filter $\sum_{l=1}^{L_{P,L2}} \bar{\boldsymbol{A}}^{2l}$, it corresponds to a low-pass filter $\sum_{l=1}^{L_{P,L2}}(\boldsymbol{I} - \bar{\boldsymbol{L}}_2)^l$ on the 2-hop neighbor graph, whose Laplacian is $\bar{\boldsymbol{L}}_2$. When the 2-hop neighborhood is homophily-dominant, $\boldsymbol{P}_{X,L2}$ is useful for aggregating neighborhood-based information.

Lastly, the adjacency embedding is $\boldsymbol{P}_A = \boldsymbol{U}|\boldsymbol{\Lambda}|^{1/2} = \boldsymbol{U}|\boldsymbol{M}|$, which is invariant under graph filter $\boldsymbol{P}_A = (\boldsymbol{A}\boldsymbol{M}^{-1})\boldsymbol{P}_A$. It hence explicitly contains information of the graph spectrum. As we show in Figure 4, its relative column norms are related with the corresponding eigenvalues.

# B Detailed Explanation of Differences from Existing Models

## B.1 Iterative GNNs with High-Frequency Propagation

We regard the message-passing GNN architectures performing iterative graph propagation and feature transformation procedures as the family of iterative GNNs. We recognize there are existing works utilizing the concept of high-pass filters or negative edges, which is similar to the inverse Laplacian propagation used in our $LD^2$:

- **FAGCN** [18] introduces the high-frequency filter $\epsilon \boldsymbol{I} - \bar{\boldsymbol{A}}$. In each layer, it respectively applies low- and high-frequency filters to the layer representation and aggregates by attention mechanism.

- **GGCN** [19] proposes the process of assigning signs to edges based on inter- and intra-node similarity. In practice, they utilize cosine similarity between node feature vectors. Its aggregation is performed on the representations corresponding to the positive edges, the negative edges, and the raw representation of previous layer, with weights of each channel controlled by a learnable scalar factor.

- **ACM** [20] explores the channel mixing mechanism, similarly applying multiple channels to learn the layer representation. For each layer, low-frequency, high-frequency, and identity channels are respectively applied to the current representation before a learnable node-wise aggregation: $\tilde{\boldsymbol{A}} \boldsymbol{H} \boldsymbol{W}_l, (\boldsymbol{I} - \tilde{\boldsymbol{A}}) \boldsymbol{H} \boldsymbol{W}_h, \boldsymbol{I} \boldsymbol{H} \boldsymbol{W}_i$

We compare $LD^2$ in the aspects of underline{selection of filers} and underline{aggregation scheme}. The filter combination $\frac{1}{L_{P,L2}} \sum_{l=1}^{L_{P,L2}} \bar{\boldsymbol{A}}^{2l}$, $\frac{1}{L_{P,H}} \sum_{l=1}^{L_{P,H}} \tilde{\boldsymbol{L}}^l$, and $\boldsymbol{I}$ is different to all these works. More importantly, all above models utilize learnable aggregation, i.e. attention, matrix-wise, or vector-wise weighted summation, for adding up these channels. However, this strategy potentially mixes the opposite information from homophilous and heterophilous nodes and leads to performance degradation. It also brings additional overhead in learning. In Appendix D we show that the overhead of GGCN is too high to be employed on graphs larger that penn94. Instead, in $LD^2$ we use a simple concatenation, and the relationship among channels is learned by the MLP.

## B.2 Iterative GNNs with Multi-Hop Propagation

We also compare the constant 2-hop adjacency propagation with other multi-scale designs:

- **H$_2$GCN** [13] examines the homophily-dominant property for 2-hop neighbors, and simultaneously performs propagation on both 1-hop and 2-hop adjacency matrices $\tilde{\boldsymbol{A}}$ and $\tilde{\boldsymbol{A}}_2$. Specifically, the 2-hop matrix $\tilde{\boldsymbol{A}}_2$ is the adjacency matrix of the induced subgraph consisting of only strict 2-hop neighbors $\tilde{\mathcal{N}}_2(u) = \{v | t \in \mathcal{N}(u), v \in \mathcal{N}(t), v \notin \mathcal{N}(u)\}$. The two representations are usually aggregated by a jumping knowledge layer.

- **MixHop** [12] concatenates identity, 1-hop, and 2-hop propagations in each of its layer $\boldsymbol{H}^{(l+1)} = \sigma(\boldsymbol{H}^{(l)} \boldsymbol{W}_0^{(l)} \| \tilde{\boldsymbol{A}} \boldsymbol{H}^{(l)} \boldsymbol{W}_1^{(l)} \| \tilde{\boldsymbol{A}}^2 \boldsymbol{H}^{(l)} \boldsymbol{W}_2^{(l)})$. Such aggregation results in expanding width of representations over multiple layers.

- **FSGNN** [23] explores convolutions $\bar{\boldsymbol{A}}^l$ and $\tilde{\boldsymbol{A}}^l$ for $l = 1, 2, 3$ on their effectiveness for homophilous and heterophilous graphs. It demands up to $O(L_P n F)$ memory for keeping all the embeddings.

- **GloGNN** [22] considers global information during message-passing, which is equivalent to a propagation of layer representations of nodes from different hops.

We note that the full-graph multi-scale propagation usually results in underline{difficulties for minibatching}. The iterative additional propagation also escalates the underline{issue in scalability}. As the number of entries in $\boldsymbol{A}^2$ is at the scale of $O(md)$, for each propagation the expense is increased by $O(d)$ times. Evaluations of full-batch MixHop and GloGNN in Appendix D validates our analysis that their model size and memory overhead scales greatly when the graph size increases.

In $LD^2$, we address these two drawbacks by adopting a low-dimensional embedding that integrates with raw features, which forms the channel $\boldsymbol{P}_{X,L2}$ and is straightforward for batching.

## B.3 Post-Propagation Decoupled GNNs

For GNNs decoupling the graph structure from iterative propagation, we further divide them into two categories according to their decoupling scheme. Models in the post-propagation category apply an embedding matrix that contains graph information only after the MLP feature transformation. Its general framework is $\boldsymbol{H}^{(L)} = \sigma(\boldsymbol{P}(\boldsymbol{A}) \cdot \text{MLP}(\boldsymbol{X}))$.

- **APPNP** [28] is among the first models proposing the decoupling design in GNN studies. It introduces the personalized PageRank (PPR) [40] matrix $\boldsymbol{P} = \sum_{l=0}^{L_P} \alpha(1-\alpha)^l \tilde{\boldsymbol{A}}^l$ to replace the iterative propagation. The decaying aggregation is shown to be only effective in locality-based homophilous settings.
- **PPRGo** [29] enhances the APPNP structure by a top-$k$ PPR matrix and a precomputation phase. It hence enjoys better adaptability to minibatch training.
- **GPRGNN** [17] is the post-propagation decoupled model for graphs of non-homophily. A learnable Generalized PPR matrix is calculated every forward passing to fit different weights for varying hops: $\boldsymbol{P}_\theta = \sum_{l=0}^{L_P} \theta_l \tilde{\boldsymbol{A}}^l$.
- **BernNet** [34] learns the graph filter by an order-$L_P$ Bernstein polynomial with non-negative coefficients constraints, denoted as $\boldsymbol{P}_\theta = \sum_{l=0}^{L_P} \theta_l T_{l,L_P}(\tilde{\boldsymbol{L}})$, where $T_{l,L_P}(\cdot)$ is the Bernstein polynomial.
- **ChebNetII** [35] similarly applies the Chebyshev polynomial approximation with decoupled transformation, that $\boldsymbol{P}_\theta = \sum_{l=0}^{L_P} \sum_{j=0}^{L_P} \theta_j T_l(x_j) T_l(\tilde{\boldsymbol{L}})$.

Although simplifying the training process, post-propagation designs still suffer from batching and scalability issues as they require graph propagation in every training epochs. Spectral models including GPRGNN, BernNet, and ChebNetII face the scalability bottleneck of storing the intermediate feature matrices of all orders in their trivial batching implementation, since they need to learn the coefficient from these feature matrices. While APPNP and PPRGo are more suitable for minibatch training, their efficiency is constrained due to the iterative propagation.

## B.4 Pre-Propagation Decoupled GNNs

In contrast to post-propagation decoupling, pre-propagation models conduct graph propagation in advance, mostly interacting with the node feature matrix. Then they perform transformation on the fixed embeddings: $\boldsymbol{H}^{(L)} = \text{MLP}(\boldsymbol{P}(\boldsymbol{X}, \boldsymbol{A}))$.

- **SGC** [30] is the simple decoupled propagation with a fix-hop propagation $\boldsymbol{P} = \tilde{\boldsymbol{A}}^{L_P} \cdot \boldsymbol{X}$.
- **S²GC** [43] performs constant summation on the powers of graph adjacency to obtain a low-frequency embedding $\boldsymbol{P} = \frac{1}{L_{P,L}} \sum_{l=0}^{L_{P,L}} \tilde{\boldsymbol{A}}^l \cdot \boldsymbol{X}$.
- **GDC** [36] formulates the generalized form $\boldsymbol{P}_\theta = \sum_{l=0}^{L_P} \theta_l \tilde{\boldsymbol{A}}^l \cdot \boldsymbol{X}$. It however mostly focuses on the Heat Kernel scheme where $\theta_l = e^{-t} \cdot \frac{t^l}{l!}$.
- **AGP** [32] proposes generalization in two aspects. First, it extends graph normalization $\boldsymbol{D}^{-1/2} \boldsymbol{A} \boldsymbol{D}^{-1/2}$ to arbitrary $\boldsymbol{D}^{-a} \boldsymbol{A} \boldsymbol{D}^{-b}$ with $a, b \in [0, 1]$. Second, it efficiently computes propagation with general coefficients $\theta_l$. In the paper, it explores the SGC, APPNP, and GDC-HeatKernel schemes.
- **LINKX** [26] directly utilizes adjacency and node feature matrices as inputs without precomputation.

The pre-propagation decoupling is known to be the most scalable design with respect to GNN training, thanks to the simple transformation scheme. We hence usually refer to this kind of architecture when using the term decoupled GNNs. As the graph structure is decoupled, the model scalability in training is only related with the dimension of inputs and layer width, and the depth of network. An extreme case, however, is LINKX that incorporates the adjacency matrix as input. Hence, its complexity is still at the scale of $O(m)$ and $O(n)$ of time and memory, respectively.

Despite the simple architecture, the embedding scheme for pre-propagation models needs to be carefully designed, otherwise they may suffer from accuracy degradation due to information loss

in the approximate propagation. Experiments in Appendix D show that most of these homophilous models cannot achieve high accuracy on heterophilous datasets. Our model, LD$^2$, is thence the first attempt that proposes novel embeddings that specifically suitable for heterophilous graphs, while maintaining the low-dimension and long-distance advantages of the decoupling design.

## C   Experiment Settings

### C.1   Heterophily Measurement

For multiclass classification task on graph $G = (V, E)$, a node $u \in V$ is labeled by $y(u) \in \{0, 1, \cdots, N_c - 1\}$, where $N_c$ is the number of classes. We measure the graph heterophily by node homophily score [14], which is the average proportion of the 1-hop neighbors with the same class of each node:

$$\mathcal{H}_{n,1} = \frac{1}{|V|} \sum_{u \in V} \frac{|\{v \in \mathcal{N}(u) : y(v) = y(u)\}|}{|\mathcal{N}(u)|}. \tag{6}$$

Generally, $\mathcal{H}_{n,1} \in [0, 1]$. A homophily score closer to 0 indicates higher heterophily, and vice versa.

As we are particularly interested in 2-hop neighbors, we also calculate the 2-hop node homophily score $\mathcal{H}_{n,2}$, which is achieved by substituting the 1-hop neighbor set in Eq. (6) with the strict 2-hop set $\bar{\mathcal{N}}_2(u) = \{v | t \in \mathcal{N}(u), v \in \mathcal{N}(t), v \notin \mathcal{N}(u)\}$.

### C.2   Datasets Statistics

We extensively evaluate 22 node classification datasets. Among them, actor, chameleon, and squirrel are *heterophilous* datasets used by [14]; roman-empire, minesweeper, amazon-ratings, and tolokers are from [55]; penn94, arxiv-year, genius, twitch-gamers, pokec, snap-patents, and wiki are *heterophilous* datasets proposed in [26]; cora, pubmed, ogbn-arxiv, protein, yelp, reddit, amazon, and ogbn-papers are popular large-scale *homophilous* datasets. Sources of the datasets are respectively cited in Table 4. Several issues revealed by [55] on the heterophilous graphs are addressed before conducting experiments. Isolated nodes in the graph are removed.

Directed edges are only considered in arxiv-year and snap-patents as per [26], while all other graphs are transformed to undirected ones. We explore the effect of directed edges in Appendix F.5.

Protein and yelp are multilabel classification tasks where more than one target classes exist for each node, while other datasets are multiclass tasks. For main experiments, we apply random 50/25/25

Table 4: Dataset statistics. $\mathcal{H}_{n,1}$ and $\mathcal{H}_{n,2}$ are 1-hop and 2-hop homophilous scores, respectively.

| Description | Dataset | Nodes $n$ | Edges $m$ | $d$ | $F$ | $N_c$ | Notes | $\mathcal{H}_{n,1}$ | $\mathcal{H}_{n,2}$ |
|---|---|---|---|---|---|---|---|---|---|
| Heterophilous Small-scale | actor [14] | $7,600$ | $34,259$ | $4.508$ | $932$ | $5$ | – | $0.216$ | $0.216$ |
| | chameleon [14] | $2,277$ | $65,019$ | $28.555$ | $2,325$ | $5$ | – | $0.247$ | $0.253$ |
| | squirrel [14] | $5,201$ | $401,907$ | $77.275$ | $2,089$ | $5$ | – | $0.217$ | $0.214$ |
| | roman-empire [55] | $22,662$ | $65,854$ | $2.906$ | $300$ | $18$ | – | $0.046$ | $0.084$ |
| | minesweeper [55] | $10,000$ | $78,804$ | $7.880$ | $7$ | $2$ | – | $0.683$ | $0.680$ |
| | amazon-ratings [55] | $24,492$ | $186,100$ | $7.598$ | $300$ | $5$ | – | $0.376$ | $0.366$ |
| Heterophilous Large-scale | tolokers [55] | $11,758$ | $1,038,000$ | $88.280$ | $10$ | $2$ | – | $0.634$ | $0.651$ |
| | penn94 [26] | $41,536$ | $1,362,220$ | $33.796$ | $4,814$ | $2$ | – | $0.504$ | $0.478$ |
| | arxiv-year [26] | $169,343$ | $1,157,799$ | $7.837$ | $128$ | $5$ | directed | $0.289$ | $0.337$ |
| | genius [26] | $421,858$ | $922,864$ | $3.188$ | $12$ | $2$ | – | $0.368$ | $0.823$ |
| | twitch-gamers [26] | $168,114$ | $6,797,557$ | $41.434$ | $7$ | $2$ | – | $0.562$ | $0.531$ |
| | pokec [26] | $1,632,803$ | $22,301,964$ | $14.659$ | $65$ | $2$ | – | $0.454$ | $0.605$ |
| | snap-patents [26] | $2,738,035$ | $13,967,949$ | $6.101$ | $269$ | $5$ | directed | $0.220$ | $0.298$ |
| | wiki [26] | $1,770,981$ | $242,507,069$ | $137.934$ | $600$ | $5$ | – | $0.306$ | – |
| Homophilous | cora [57] | $2,485$ | $12,623$ | $5.080$ | $1,433$ | $7$ | – | $0.814$ | $0.720$ |
| | pubmed [57] | $19,717$ | $88,648$ | $4.496$ | $500$ | $3$ | – | $0.792$ | $0.742$ |
| | protein [2] | $56,944$ | $818,716$ | $14.378$ | $50$ | $121$ | multilabel | – | – |
| | ogbn-arxiv [58] | $169,343$ | $2,484,941$ | $14.674$ | $128$ | $40$ | – | $0.635$ | $0.489$ |
| | yelp [27] | $716,847$ | $6,977,410$ | $9.733$ | $300$ | $100$ | multilabel | – | – |
| | reddit [2] | $232,965$ | $114,615,892$ | $491.988$ | $602$ | $41$ | – | $0.445$ | – |
| | amazon [33] | $2,400,608$ | $123,718,024$ | $51.536$ | $100$ | $47$ | – | $0.833$ | – |
| | ogbn-papers [58] | $111,059,956$ | $1,615,685,872$ | $14.548$ | $128$ | $172$ | – | $0.964$ | – |

train/validate/test splits for all graphs, and perform transductive predictions on them, with the only exception ogbn-papers. We also mark protein and yelp in inductive settings as protein-ind and yelp-ind, respectively. We uniformly use the micro F1 score to evaluate efficacy, which is equivalent to accuracy for multiclass predictions.

We list the statistics of these datasets in Table 4, including the 1-hop and 2-hop node homophily scores for non-multilabel datasets. The empirical results support our analysis in Section 3 that regardless of heterophily, 2-hop neighbors in the graph tend to exhibit higher homophily.

### C.3 Baseline Models

Here we list all 17 types of baseline models we evaluate, varying from common and heterophilous GNNs with full-batch and mini-batch training schemes. Since we mostly focus on scalability and evaluate on large graphs, models such as $H_2$GCN [13], GeomGCN [14], FAGCN [18], and WRGCN [16] are not included, as they experience prohibited running time or out-of-memory error on most of the datasets.

- **I.I.D. Minibatch Heterophilous Models:** LINKX [26] is applicable to the simple i.i.d. node batching scheme by design.
- **Graph Sampling Minibatch Heterophilous Models:** GCNJK-GS [21] and MixHop-GS [12] are the variants of these models employed with GSAINT random walk sampling [27], which is the sampling scheme of best performance evaluated in [26].
- **I.I.D. Minibatch Homophilous Models:** MLP processes only node attributes without considering graph topology. PPRGo [29] is the post-propagation decoupled model as a faster alternative of APPNP. SGC [30] is the 2-hop varaint of the pre-propagation model. All decoupled models utilize simple sampling.
- **Full-batch Heterophilous Models:** LINKX [26], GCNJK [21], MixHop [12], GCNII [45], and GPRGNN [17] are based on the implementation of [26]. GGCN [19], FSGNN [23], GloGNN++ [22] and ACM [20] mostly follow their own public source codes.
- **Full-batch Homophilous Models:** The vanilla GCN [1], decoupled models APPNP [28] and SGC [30] are evaluated as representatives of homophilous GNNs with different scalabilities.

### C.4 Model and Training Hyperparameters

We particularly explore model hyperparameters including the number of layers $L$, i.e. model depth, and the number of hidden size, i.e. layer width, since these settings are mostly correlated with the efficacy and efficiency performance of models.

For minibatch training, we comprehensively tune the hyperparameters of batch size and learning rate among baselines to produce comparable performance. We exploit the validation set to select the training epoch with best validation accuracy, and use early stopping if the model training converges.

We select above hyperparameters based on the following principle: We first refer to their original papers and implementations and explore model depth and width, in order to achieve relatively optimal reproduced performance. Then we select the largest batch size applicable to the GPU while preventing out of memory error for efficiency consideration. Other hyperparameters including weight decays and learning rates are tuned accordingly. For other architectural and training settings, we mostly follow the implementation in [26] when applicable, in order to produce similar evaluation to the benchmark.

Table 5 shows the details of hyperparameters exploration ranges and eventual settings of 8 datasets and 7 minibatch models corresponding to main experiments in Tables 2 and 3. Explorations and settings of $LD^2$ hyperparameters are further discussed in Appendix F.

Table 5: Hyperparameter explorations and settings of respective datasets and minibatch models in main experiments. Hyperparameters are explored based on the combination of listed ranges, and underlined values are optimal settings used.

| Dataset | Model | learning rate | batch | layer | hidden | weight decay | dropout | other |
|---|---|---|---|---|---|---|---|---|
| tolokers | MLP | 0.05, 0.02, 0.01, 0.005, 0.001 | 8192 | 2, 3, 4 | 512 | 1.00E-03 | 0.5 | – |
| | PPRGo | 0.01, 0.005, 0.001 | 8192 | 2, 3 | 256 | 1.00E-04 | 0.1 | $\alpha$=0.5, $k$=128 |
| | SGC | 0.01, 0.005, 0.001 | 8192 | 2, 3 | 256 | 1.00E-04 | 0.5 | $\alpha$=0.5, $a$=0.5 |
| | GCNJK-GS | 0.05, 0.02, 0.01, 0.005, 0.001 | 8192 | 2, 3, 4 | 512 | 1.00E-03 | 0.5 | jk_type=cat |
| | MixHop-GS | 0.05, 0.02, 0.01, 0.005, 0.001 | 8192 | 1, 2, 3, 4 | 512 | 1.00E-03 | 0.5 | – |
| | LINKX | 0.05, 0.02, 0.01, 0.005, 0.001 | 8192 | 1, 2, 3, 4 | 512 | 1.00E-05 | 0.5 | – |
| | LD$^2$ | 0.01, 0.005, 0.002, 0.001 | 8192 | 1, 2, 3 | 512 | 1.00E-04 | 0.5 | $L_P$=10, $a$=0.5 |
| penn94 | MLP | 0.05, 0.02, 0.01, 0.005, 0.001 | 8192 | 2, 3, 4 | 512 | 1.00E-03 | 0.5 | – |
| | PPRGo | 0.01, 0.005, 0.001 | 2048 | 2, 3 | 256 | 1.00E-04 | 0.1 | $\alpha$=0.5, $k$=256 |
| | SGC | 0.01, 0.005, 0.001 | 81920 | 2, 3 | 256 | 1.00E-04 | 0.5 | $\alpha$=0.5, $a$=0.5 |
| | GCNJK-GS | 0.05, 0.02, 0.01, 0.005, 0.001 | 8192 | 2, 3, 4 | 512 | 1.00E-03 | 0.5 | jk_type=cat |
| | MixHop-GS | 0.05, 0.02, 0.01, 0.005, 0.001 | 8192 | 1, 2, 3, 4 | 512 | 1.00E-03 | 0.5 | – |
| | LINKX | 0.05, 0.02, 0.01, 0.005, 0.001 | 8192 | 1, 2, 3, 4 | 512 | 1.00E-03 | 0.5 | – |
| | LD$^2$ | 0.01, 0.005, 0.002, 0.001 | 20480 | 1, 2, 3 | 512 | 1.00E-04 | 0.5 | $L_P$=20, $a$=0.5 |
| arxiv-year | MLP | 0.05, 0.02, 0.01, 0.005, 0.001 | 8192 | 2, 3, 4 | 512 | 1.00E-03 | 0.5 | – |
| | PPRGo | 0.01, 0.005, 0.001 | 8192 | 2, 3 | 256 | 1.00E-04 | 0.1 | $\alpha$=0.5, $k$=128 |
| | SGC | 0.01, 0.005, 0.001 | 81920 | 2, 3 | 256 | 1.00E-04 | 0.5 | $\alpha$=0.5, $a$=0.5 |
| | GCNJK-GS | 0.05, 0.02, 0.01, 0.005, 0.001 | 8192 | 1, 2, 3, 4 | 512 | 1.00E-03 | 0.5 | jk_type=cat |
| | MixHop-GS | 0.05, 0.02, 0.01, 0.005, 0.001 | 8192 | 1, 2, 3, 4 | 512 | 1.00E-03 | 0.5 | – |
| | LINKX | 0.05, 0.02, 0.01, 0.005, 0.001 | 8192 | 1, 2, 3, 4 | 512 | 1.00E-03 | 0.5 | – |
| | LD$^2$ | 0.01, 0.005, 0.002, 0.001 | 81920 | 1, 2, 3 | 512 | 1.00E-04 | 0.5 | $L_P$=16, $a$=0.5 |
| genius | MLP | 0.05, 0.02, 0.01, 0.005, 0.001 | 8192 | 2, 3, 4 | 512 | 1.00E-03 | 0.5 | – |
| | PPRGo | 0.01, 0.005, 0.001 | 8192 | 2, 3 | 256 | 1.00E-04 | 0.1 | $\alpha$=0.5, $k$=128 |
| | SGC | 0.01, 0.005, 0.001 | 81920 | 2, 3 | 256 | 1.00E-04 | 0.5 | $\alpha$=0.5, $a$=0.5 |
| | GCNJK-GS | 0.05, 0.02, 0.01, 0.005, 0.001 | 8192 | 2, 3, 4 | 512 | 1.00E-03 | 0.5 | jk_type=cat |
| | MixHop-GS | 0.05, 0.02, 0.01, 0.005, 0.001 | 8192 | 1, 2, 3, 4 | 512 | 1.00E-03 | 0.5 | – |
| | LINKX | 0.05, 0.02, 0.01, 0.005, 0.001 | 8192 | 1, 2, 3, 4 | 512 | 1.00E-03 | 0.5 | – |
| | LD$^2$ | 0.01, 0.005, 0.002, 0.001 | 20480 | 1, 2, 3 | 512 | 1.00E-04 | 0.5 | $L_P$=20, $a$=0.5 |
| twitch-gamers | MLP | 0.05, 0.02, 0.01, 0.005, 0.001 | 8192 | 2, 3, 4 | 512 | 1.00E-03 | 0.5 | – |
| | PPRGo | 0.01, 0.005, 0.001 | 8192 | 2, 3 | 256 | 1.00E-04 | 0.1 | $\alpha$=0.5, $k$=128 |
| | SGC | 0.01, 0.005, 0.001 | 81920 | 2, 3 | 256 | 1.00E-04 | 0.5 | $\alpha$=0.5, $a$=0.5 |
| | GCNJK-GS | 0.05, 0.02, 0.01, 0.005, 0.001 | 8192 | 1, 2, 3, 4 | 512 | 1.00E-03 | 0.5 | jk_type=cat |
| | MixHop-GS | 0.05, 0.02, 0.01, 0.005, 0.001 | 8192 | 1, 2, 3, 4 | 512 | 1.00E-03 | 0.5 | – |
| | LINKX | 0.05, 0.02, 0.01, 0.005, 0.001 | 8192 | 1, 2, 3, 4 | 512 | 1.00E-03 | 0.5 | – |
| | LD$^2$ | 0.01, 0.005, 0.002, 0.001 | 20480 | 1, 2, 3 | 512 | 1.00E-04 | 0.5 | $L_P$=10, $a$=0.5 |
| pokec | MLP | 0.05, 0.02, 0.01, 0.005, 0.001 | 8192 | 2, 3, 4 | 512 | 1.00E-03 | 0.5 | – |
| | PPRGo | 0.01, 0.005, 0.001 | 8192 | 2, 3 | 256 | 1.00E-04 | 0.1 | $\alpha$=0.5, $k$=128 |
| | SGC | 0.01, 0.005, 0.001 | 81920 | 2, 3 | 256 | 1.00E-04 | 0.5 | $\alpha$=0.5, $a$=0.5 |
| | GCNJK-GS | 0.05, 0.02, 0.01, 0.005, 0.001 | 8192 | 1, 2, 3, 4 | 512 | 1.00E-03 | 0.5 | jk_type=cat |
| | MixHop-GS | 0.05, 0.02, 0.01, 0.005, 0.001 | 8192 | 1, 2, 3, 4 | 512 | 1.00E-03 | 0.5 | – |
| | LINKX | 0.05, 0.02, 0.01, 0.005, 0.001 | 8192 | 1, 2, 3, 4 | 256 | 1.00E-03 | 0.5 | – |
| | LD$^2$ | 0.01, 0.005, 0.002, 0.001 | 81920 | 1, 2, 3 | 512 | 1.00E-04 | 0.5 | $L_P$=20, $a$=0.5 |
| snap-patents | MLP | 0.05, 0.02, 0.01, 0.005, 0.001 | 8192 | 2, 3, 4 | 512 | 1.00E-03 | 0.5 | – |
| | PPRGo | 0.01, 0.005, 0.001 | 1024 | 2, 3 | 256 | 1.00E-04 | 0.1 | $\alpha$=0.5, $k$=64 |
| | SGC | 0.01, 0.005, 0.001 | 81920 | 2, 3 | 256 | 1.00E-04 | 0.5 | $\alpha$=0.5, $a$=0.5 |
| | GCNJK-GS | 0.05, 0.02, 0.01, 0.005, 0.001 | 8192 | 1, 2, 3, 4 | 512 | 1.00E-03 | 0.5 | jk_type=cat |
| | MixHop-GS | 0.05, 0.02, 0.01, 0.005, 0.001 | 8192 | 1, 2, 3, 4 | 512 | 1.00E-03 | 0.5 | – |
| | LINKX | 0.05, 0.02, 0.01, 0.005, 0.001 | 8192 | 1, 2, 3, 4 | 256 | 1.00E-03 | 0.5 | – |
| | LD$^2$ | 0.01, 0.005, 0.002, 0.001 | 81920 | 1, 2, 3 | 256 | 1.00E-04 | 0.5 | $L_P$=20, $a$=0.5 |
| wiki | MLP | 0.05, 0.02, 0.01, 0.005, 0.001 | 8192 | 2, 3, 4 | 512 | 1.00E-03 | 0.5 | – |
| | PPRGo | 0.01, 0.005, 0.001 | 1024 | 2, 3 | 256 | 1.00E-04 | 0.1 | $\alpha$=0.5, $k$=64 |
| | SGC | 0.01, 0.005, 0.001 | 81920 | 2, 3 | 256 | 1.00E-04 | 0.5 | $\alpha$=0.5, $a$=0.5 |
| | GCNJK-GS | 0.05, 0.02, 0.01, 0.005, 0.001 | 8192 | 1, 2, 3, 4 | 512 | 1.00E-03 | 0.5 | jk_type=cat |
| | MixHop-GS | 0.05, 0.02, 0.01, 0.005, 0.001 | 8192 | 1, 2, 3, 4 | 512 | 1.00E-03 | 0.5 | – |
| | LINKX | 0.05, 0.02, 0.01, 0.005, 0.001 | 8192 | 1, 2, 3, 4 | 128 | 1.00E-03 | 0.5 | – |
| | LD$^2$ | 0.01, 0.005, 0.002, 0.001 | 81920 | 1, 2, 3 | 512 | 1.00E-04 | 0.5 | $L_P$=10, $a$=0.5 |

Table 6: Average test accuracy (%) of minibatch $LD^2$ and baselines on 6 small-scale heterophilous datasets. "rank" is the average ranking among all 14 heterophilous datasets.

| Dataset | actor | chameleon | squirrel | roman-empire | minesweeper | amazon-ratings | rank |
|---|---|---|---|---|---|---|---|
| MLP | 36.05 | 43.86 | 33.16 | 65.89 | 49.26 | 44.14 | 4.9 |
| PPRGo | 21.51 | 49.48 | 33.95 | 72.84 | 80.00 | 48.10 | 4.9 |
| SGC | 26.14 | 66.96 | 59.39 | 64.37 | 79.89 | 46.49 | 4.8 |
| GCNJK-GS | 29.00 | 41.58 | 27.63 | 53.49 | 58.03 | 40.58 | 5.1 |
| MixHop-GS | 33.24 | 75.00 | 33.24 | 63.47 | 60.07 | 46.79 | 3.8 |
| LINKX | 30.58 | 67.37 | 60.26 | 49.74 | 50.96 | 51.97 | 2.6 |
| $LD^2$ | 33.04 | 69.76 | 66.87 | 77.30 | 77.11 | 52.05 | 2.0 |

Table 7: Average test accuracy (%) of full-batch $LD^2$ and baselines on 10 heterophilous datasets. "OOM" means the model occurs out of memory error with applicable hyperparameters. "rank" is the average ranking among these 10 datasets.

| Dataset | actor | chameleon | squirrel | penn94 | arxiv-year | genius | twitch-gamers | pokec | snap-patents | wiki | rank |
|---|---|---|---|---|---|---|---|---|---|---|---|
| APPNP | 32.11 | 48.95 | 33.97 | 72.85 | 38.31 | 83.59 | 60.07 | 61.23 | 31.11 | (OOM) | 9.1 |
| SGC | 34.42 | 44.74 | 29.59 | 72.92 | 34.85 | 80.00 | 59.56 | 61.27 | 29.89 | (OOM) | 9.9 |
| GCN | 25.11 | 43.33 | 29.21 | 72.95 | 46.24 | 80.07 | 61.01 | 67.31 | 46.55 | (OOM) | 9.1 |
| GCNII | 25.11 | 52.46 | 37.59 | 72.71 | 45.29 | 80.02 | 60.70 | 72.60 | (OOM) | (OOM) | 9.4 |
| GPR-GNN | 32.47 | 56.49 | 34.82 | 73.88 | 44.81 | 83.16 | 60.13 | 73.48 | 42.82 | (OOM) | 7.0 |
| GCNJK | 26.42 | 57.89 | 28.52 | 70.94 | 50.38 | 50.38 | 61.50 | 69.10 | 47.79 | (OOM) | 8.7 |
| MixHop | 31.26 | 60.35 | 47.35 | 75.00 | 54.31 | 84.33 | 64.30 | 78.05 | 54.24 | (OOM) | 4.2 |
| LINKX | 27.21 | 66.14 | 60.11 | 76.17 | 53.38 | 82.57 | 64.06 | 66.93 | 53.91 | (OOM) | 5.4 |
| GGCN | 35.21 | 50.35 | 32.82 | 73.44 | (OOM) | (OOM) | (OOM) | (OOM) | (OOM) | (OOM) | 10.5 |
| FSGNN | 33.96 | 71.17 | 60.34 | 74.28 | 42.74 | 82.61 | 60.97 | (OOM) | (OOM) | (OOM) | 7.3 |
| ACM | 34.79 | 63.68 | 50.96 | 74.10 | 43.22 | 80.64 | 58.65 | 55.05 | (OOM) | (OOM) | 8.1 |
| GloGNN++ | 34.16 | 69.12 | 24.21 | 79.43 | 53.70 | 82.67 | 64.14 | 70.82 | (OOM) | (OOM) | 6.0 |
| $LD^2$ | 33.35 | 61.87 | 51.22 | 73.24 | 49.53 | 80.45 | 63.97 | 72.66 | 54.52 | 49.19 | 3.9 |

# D  Extended Experiments under Heterophily

## D.1  Efficacy Results

To supplement the main results shown in Table 2, we conduct extensive experiments on a total number of 14 heterophilous datasets, covering 7 minibatch models and 13 full-batch models. We produce all the experiments by our own to ensure the dataset updates and comparable efficiency performance.

Results regarding prediction accuracy are shown in Tables 6 and 7 for minibatch and full-batch models, respectively. We also include a rank of each model calculated by the average ranking on these 10 datasets. Models without final results due to exceeding running time or memory are regarded as 0% accuracy on the specific dataset when computing the rankings.

Our observations and conclusions in Section 4 still hold, that the $LD^2$ model achieves comparable or better results on most of the datasets, with the best average ranking for both minibatch and full-batch training. It is also the only model that succeeds in full-batch training on graph wiki under the 24GB GPU memory constraint.

However, we also note that, on certain datasets such as chameleon, squirrel, and genius, the testing performance of $LD^2$ full-batch learning is lower than those in minibatch training. We deduce the reason may root in suboptimal convergence for the model, where full-batch training may mix the gradient of heterophilous nodes and fail to discover certain sets of minority node groups which are beneficial for model prediction. Instead, smaller batches of samples are able to guide the model to the better global optimum.

In summary, the evaluation emphasizes the need of minibatch training for scaling GNNs to large-scale graphs to prevent prohibited overhead. We believe $LD^2$ provides an effective and practical solution to this scenario.

Table 8: Efficiency evaluation of minibatch $LD^2$ and baselines on heterophilous datasets. "Pre.", "Train", and "Infer" respectively refer to precomputation, training, and inference time (s). "Avg" is the average training time per epoch (ms) with a total number "#" of epochs before convergence. "RAM" and "GPU" respectively refer to peak RAM and GPU memory (GB). "Param." and "Size" are the number of trainable model parameters (M) and estimated memory size (MB), respectively.

| Model | Dataset | Learn | Infer | # | Avg | RAM | GPU | Param. | Size | Dataset | Learn | Infer | # | Avg | RAM | GPU | Param. | Size |
|---|---|---|---|---|---|---|---|---|---|---|---|---|---|---|---|---|---|---|
| MLP | | 1.78 | 0.004 | 236 | 7.56 | 4.89 | 0.06 | 1.01 | 3.86 | | 1.38 | 0.004 | 224 | 6.18 | 4.88 | 0.06 | 1.72 | 6.58 |
| PPRGo | | 0.29+1.02 | 0.026 | 106 | 9.59 | 5.86 | 0.85 | 0.31 | 1.17 | | 0.41+3.23 | 0.024 | 234 | 13.81 | 6.90 | 3.17 | 0.66 | 2.53 |
| SGC | | 0.02+0.52 | 0.010 | 94 | 5.49 | 5.08 | 0.04 | 0.24 | 0.92 | | 0.03+0.57 | 0.001 | 154 | 3.72 | 5.08 | 0.04 | 0.60 | 2.28 |
| GCNJK-GS | actor | 9.07 | 0.013 | 115 | 78.85 | 4.96 | 0.50 | 1.28 | 4.89 | chameleon | 9.61 | 0.011 | 119 | 80.76 | 4.95 | 0.44 | 1.99 | 7.61 |
| MixHop-GS | | 29.06 | 0.010 | 200 | 145.30 | 5.13 | 0.71 | 3.82 | 14.61 | | 35.37 | 0.040 | 113 | 313.01 | 6.20 | 1.49 | 5.96 | 22.77 |
| LINKX | | 0.78 | 0.006 | 144 | 5.44 | 4.89 | 0.07 | 2.32 | 8.84 | | 1.10 | 0.005 | 192 | 5.74 | 4.88 | 0.04 | 1.31 | 5.00 |
| $LD^2$ | | 0.42+0.90 | 0.008 | 54 | 16.58 | 5.19 | 0.10 | 1.36 | 5.21 | | 1.51+0.79 | 0.002 | 87 | 9.12 | 5.16 | 0.13 | 2.75 | 10.48 |
| MLP | | 2.01 | 0.010 | 260 | 7.73 | 4.99 | 0.05 | 1.60 | 6.12 | | 3.47 | 0.040 | 133 | 26.09 | 5.86 | 0.64 | 2.99 | 11.44 |
| PPRGo | | 0.40+7.01 | 0.040 | 291 | 24.09 | 8.34 | 4.33 | 0.60 | 2.29 | | 2.56+24.70 | 0.470 | 147 | 168.41 | 56.53 | 18.73 | 1.23 | 4.70 |
| SGC | | 0.21+1.07 | 0.010 | 236 | 4.53 | 5.10 | 0.06 | 0.54 | 2.05 | | 0.75+2.99 | 0.010 | 93 | 32.15 | 5.87 | 0.22 | 1.23 | 4.71 |
| GCNJK-GS | squirrel | 35.52 | 0.000 | 241 | 147.39 | 5.10 | 4.89 | 1.34 | 5.11 | penn94 | 48.82 | 0.040 | 141 | 346.24 | 6.30 | 4.62 | 2.73 | 10.42 |
| MixHop-GS | | 29.06 | 0.010 | 200 | 145.30 | 5.13 | 0.71 | 7.96 | 30.42 | | 35.37 | 0.040 | 113 | 313.01 | 6.20 | 1.49 | 7.41 | 28.27 |
| LINKX | | 2.23 | 0.006 | 234 | 9.51 | 3.00 | 0.07 | 2.00 | 7.63 | | 2.67 | 0.040 | 119 | 22.50 | 6.17 | 0.71 | 5.98 | 22.82 |
| $LD^2$ | | 3.87+0.66 | 0.001 | 122 | 5.41 | 5.06 | 0.05 | 0.44 | 1.70 | | 27.19+1.11 | 0.010 | 92 | 12.11 | 8.57 | 1.07 | 11.71 | 44.67 |
| MLP | | 1.04 | 0.004 | 212 | 4.92 | 5.16 | 0.05 | 0.43 | 1.64 | | 0.73 | 0.002 | 188 | 3.90 | 5.13 | 0.03 | 0.27 | 1.04 |
| PPRGo | | 0.40+4.42 | 0.044 | 461 | 9.60 | 5.82 | 1.57 | 0.08 | 0.31 | | 0.53+0.31 | 0.009 | 52 | 5.90 | 5.20 | 1.77 | 0.00 | 0.01 |
| SGC | | 1.17 | 0.002 | 119 | 9.80 | 5.09 | 0.06 | 0.08 | 0.32 | | 0.31 | 0.001 | 65 | 4.80 | 5.05 | 0.04 | 0.00 | 0.01 |
| GCNJK-GS | roman-empire | 9.29 | 0.009 | 111 | 0.08 | 5.00 | 0.91 | 0.71 | 2.71 | minesweeper | 7.83 | 0.007 | 131 | 0.06 | 4.86 | 0.86 | 0.53 | 2.05 |
| MixHop-GS | | 25.02 | 0.011 | 229 | 0.43 | 4.97 | 1.95 | 2.91 | 11.14 | | 7.52 | 0.007 | 107 | 0.07 | 4.86 | 1.23 | 2.39 | 9.13 |
| LINKX | | 0.81 | 0.006 | 152 | 0.01 | 4.90 | 0.19 | 6.01 | 22.94 | | 0.71 | 0.004 | 120 | 0.01 | 4.86 | 0.07 | 2.69 | 10.28 |
| $LD^2$ | | 0.07+2.07 | 0.002 | 157 | 13.20 | 5.22 | 0.27 | 0.70 | 2.67 | | 0.02+0.69 | 0.002 | 68 | 10.20 | 5.06 | 0.25 | 1.35 | 5.17 |
| MLP | | 4.13 | 0.005 | 500 | 8.25 | 5.20 | 0.10 | 0.42 | 1.62 | | 2.51 | 0.011 | 221 | 11.34 | 5.23 | 0.05 | 0.27 | 1.04 |
| PPRGo | | 1.11+9.81 | 0.038 | 500 | 19.60 | 6.81 | 3.29 | 0.08 | 0.30 | | 0.91+0.34 | 0.050 | 52 | 6.50 | 5.24 | 1.43 | 0.00 | 0.01 |
| SGC | | 1.87 | 0.002 | 191 | 9.83 | 5.08 | 0.06 | 0.08 | 0.30 | | 0.65 | 0.001 | 121 | 5.47 | 5.05 | 0.03 | 0.00 | 0.02 |
| GCNJK-GS | amazon-ratings | 31.04 | 0.010 | 261 | 0.12 | 5.00 | 1.98 | 0.69 | 2.64 | tolokers | 39.54 | 0.008 | 115 | 0.34 | 5.05 | 20.13 | 0.54 | 2.05 |
| MixHop-GS | | 36.35 | 0.010 | 323 | 0.11 | 4.99 | 1.35 | 2.85 | 10.90 | | 13.64 | 0.007 | 118 | 0.12 | 5.03 | 1.31 | 2.39 | 9.15 |
| LINKX | | 2.34 | 0.013 | 253 | 0.01 | 4.91 | 0.19 | 6.48 | 24.72 | | 3.41 | 0.061 | 128 | 0.03 | 4.96 | 0.09 | 3.14 | 12.00 |
| $LD^2$ | | 0.32+1.35 | 0.003 | 100 | 13.46 | 5.22 | 0.47 | 2.01 | 7.67 | | 0.07+1.16 | 0.003 | 111 | 10.44 | 5.07 | 0.31 | 1.47 | 5.60 |
| MLP | | 5.74 | 0.011 | 420 | 8.45 | 5.11 | 0.61 | 0.60 | 2.29 | | 7.99 | 0.020 | 128 | 62.42 | 4.99 | 2.01 | 0.54 | 2.06 |
| PPRGo | | 9.12+59.67 | 0.290 | 294 | 202.96 | 13.43 | 4.26 | 0.03 | 0.13 | | 13.38+25.61 | 0.200 | 146 | 175.81 | 6.02 | 1.40 | 0.07 | 0.26 |
| SGC | | 1.82+6.90 | 0.010 | 305 | 22.60 | 5.15 | 0.04 | 0.03 | 0.13 | | 0.05+18.16 | 0.010 | 256 | 70.94 | 5.12 | 0.10 | 0.00 | 0.02 |
| GCNJK-GS | arxiv-year | 81.15 | 0.050 | 283 | 286.75 | 5.40 | 2.83 | 0.87 | 3.32 | genius | 13.69 | 0.030 | 224 | 61.12 | 5.14 | 13.04 | 0.80 | 3.07 |
| MixHop-GS | | 64.74 | 0.015 | 311 | 208.17 | 5.45 | 6.01 | 4.95 | 18.93 | | 13.53 | 0.010 | 163 | 83.01 | 5.16 | 8.96 | 2.40 | 9.16 |
| LINKX | | 5.42 | 0.090 | 261 | 20.80 | 5.36 | 2.34 | 87.30 | 333.01 | | 7.07 | 0.073 | 117 | 60.30 | 5.96 | 5.57 | 216.52 | 825.97 |
| $LD^2$ | | 2.83+5.60 | 0.010 | 74 | 76.01 | 7.98 | 0.56 | 1.71 | 6.52 | | 0.79+29.00 | 0.020 | 168 | 172.28 | 5.38 | 0.45 | 0.36 | 1.36 |
| MLP | | 6.36 | 0.020 | 383 | 16.61 | 5.08 | 0.61 | 0.53 | 2.05 | | 47.86 | 0.110 | 408 | 117.30 | 6.24 | 13.77 | 0.56 | 2.16 |
| PPRGo | | 10.46+15.88 | 0.410 | 145 | 109.52 | 6.03 | 9.64 | 0.00 | 0.01 | | 121.95+56.11 | 2.690 | 128 | 439.49 | 29.03 | 3.82 | 0.02 | 0.07 |
| SGC | | 0.09+0.74 | 0.010 | 84 | 8.81 | 5.01 | 0.28 | 0.00 | 0.01 | | 1.05+8.08 | 0.010 | 117 | 68.87 | 5.47 | 0.28 | 0.02 | 0.07 |
| GCNJK-GS | twitch-gamers | 71.48 | 0.022 | 471 | 151.77 | 5.36 | 7.33 | 0.53 | 2.05 | pokec | 27.33 | 0.090 | 451 | 60.60 | 6.82 | 9.03 | 0.56 | 2.16 |
| MixHop-GS | | 52.12 | 0.010 | 475 | 109.73 | 5.35 | 1.49 | 2.39 | 9.13 | | 71.35 | 0.010 | 469 | 152.13 | 6.81 | 12.91 | 4.84 | 18.50 |
| LINKX | | 5.39 | 0.098 | 113 | 47.70 | 4.38 | 1.23 | 43.17 | 164.69 | | 24.31 | 0.314 | 130 | 187.60 | 7.59 | 15.67 | 418.28 | 1595.61 |
| $LD^2$ | | 0.85+1.96 | 0.010 | 99 | 19.80 | 5.43 | 1.44 | 1.48 | 5.66 | | 17.95+6.18 | 0.010 | 108 | 57.05 | 10.67 | 3.82 | 1.50 | 5.72 |
| MLP | | 27.39 | 0.280 | 141 | 194.26 | 9.60 | 9.33 | 0.41 | 1.55 | | 133.55 | 0.620 | 375 | 356.13 | 16.75 | 18.15 | 0.84 | 3.21 |
| PPRGo | | (>12h) | | | | | | | | | (>12h) | | | | | | | |
| SGC | | 4.94+23.54 | 0.010 | 396 | 59.49 | 12.18 | 0.42 | 0.07 | 0.27 | | 12.66+7.98 | 0.010 | 200 | 39.83 | 16.25 | 0.52 | 0.16 | 0.60 |
| GCNJK-GS | snap-patents | 19.02 | 0.232 | 323 | 58.87 | 9.06 | 9.21 | 0.41 | 1.56 | wiki | 95.52 | 0.690 | 423 | 225.82 | 23.25 | 16.36 | 1.11 | 4.24 |
| MixHop-GS | | 45.23 | 0.159 | 387 | 116.89 | 9.09 | 19.58 | 2.80 | 10.72 | | 84.22 | 0.230 | 422 | 199.57 | 23.26 | 16.28 | 3.31 | 12.66 |
| LINKX | | 39.80 | 0.220 | 205 | 194.50 | 17.53 | 21.53 | 701.14 | 2674.63 | | 233.01 | 1.730 | 259 | 899.65 | 24.57 | 21.57 | 453.66 | 1730.57 |
| $LD^2$ | | 31.32+6.96 | 0.020 | 68 | 102.35 | 35.07 | 3.96 | 0.63 | 2.40 | | 28.12+6.50 | 0.010 | 90 | 71.96 | 35.45 | 4.47 | 2.61 | 9.97 |

Table 9: Efficiency evaluation of full-batch LD$^2$ and baselines on heterophilous datasets.

| Model | Dataset | Learn | Infer | # | Avg | RAM | GPU | Param. | Size | Dataset | Learn | Infer | # | Avg | RAM | GPU | Param. | Size |
|---|---|---|---|---|---|---|---|---|---|---|---|---|---|---|---|---|---|---|
| APPNP |  | 1.13 | 0.001 | 166 | 6.79 | 4.86 | 0.08 | 0.24 | 0.92 |  | 1.04 | 0.001 | 158 | 6.60 | 4.86 | 0.07 | 0.60 | 2.28 |
| SGC |  | 0.48 | 0.001 | 194 | 2.49 | 4.87 | 0.03 | 0.00 | 0.02 |  | 0.61 | 0.001 | 282 | 2.15 | 4.86 | 0.03 | 0.01 | 0.04 |
| GCN |  | 2.98 | 0.005 | 139 | 21.42 | 4.86 | 0.07 | 0.06 | 0.23 |  | 2.39 | 0.005 | 111 | 21.52 | 4.86 | 0.06 | 0.15 | 0.57 |
| GCNII |  | 5.37 | 0.006 | 122 | 44.02 | 4.86 | 0.21 | 0.13 | 0.49 |  | 17.99 | 0.005 | 446 | 40.34 | 4.86 | 0.12 | 0.22 | 0.83 |
| GPRGNN |  | 1.96 | 0.003 | 121 | 16.21 | 4.86 | 0.08 | 0.24 | 0.92 |  | 5.22 | 0.003 | 331 | 15.78 | 4.86 | 0.07 | 0.60 | 2.28 |
| GCNJK |  | 1.92 | 0.001 | 243 | 7.90 | 4.87 | 0.34 | 1.01 | 3.87 |  | 1.53 | 0.002 | 164 | 9.33 | 4.86 | 0.43 | 1.73 | 6.59 |
| MixHop | actor | 1.62 | 0.001 | 165 | 9.82 | 4.87 | 0.36 | 1.32 | 5.05 | chameleon | 2.84 | 0.001 | 289 | 9.82 | 4.87 | 0.18 | 2.39 | 9.14 |
| LINKX |  | 0.63 | 0.000 | 113 | 5.56 | 4.87 | 0.15 | 2.32 | 8.84 |  | 0.81 | 0.000 | 173 | 4.70 | 4.86 | 0.07 | 1.31 | 5.00 |
| GGCN |  | 21.71 | 0.025 | 112 | 193.80 | 5.12 | 0.42 | 0.48 | 1.83 |  | 50.82 | 0.053 | 121 | 420.00 | 5.12 | 0.50 | 1.19 | 4.55 |
| FSGNN |  | 0.16+2.55 | 0.002 | 178 | 14.30 | 5.14 | 0.51 | 0.54 | 2.06 |  | 0.13+6.72 | 0.003 | 500 | 13.43 | 5.12 | 0.43 | 1.34 | 5.12 |
| ACM |  | 1.84 | 0.001 | 114 | 16.20 | 5.13 | 0.15 | 0.72 | 2.75 |  | 3.44 | 0.001 | 231 | 14.90 | 5.12 | 0.11 | 1.79 | 6.83 |
| GloGNN++ |  | 10.77 | 0.002 | 501 | 21.50 | 5.15 | 0.12 | 0.55 | 4.20 |  | 7.45 | 0.003 | 272 | 27.30 | 5.12 | 0.07 | 0.30 | 2.28 |
| LD$^2$ |  | 0.42+0.28 | 0.001 | 60 | 4.73 | 5.16 | 0.22 | 1.36 | 5.21 |  | 1.51+0.41 | 0.001 | 98 | 4.33 | 5.12 | 0.17 | 2.75 | 10.48 |
| APPNP |  | 1.24 | 0.001 | 232 | 5.34 | 4.88 | 0.10 | 0.54 | 2.05 |  | 1.23 | 0.002 | 118 | 10.39 | 5.61 | 0.98 | 1.23 | 4.71 |
| SGC |  | 0.62 | 0.001 | 286 | 2.16 | 4.88 | 0.10 | 0.01 | 0.04 |  | 0.95 | 0.001 | 195 | 4.85 | 5.60 | 0.92 | 0.01 | 0.04 |
| GCN |  | 3.49 | 0.005 | 155 | 22.54 | 4.90 | 0.28 | 0.13 | 0.51 |  | 8.06 | 0.007 | 254 | 31.74 | 5.60 | 1.53 | 0.31 | 1.18 |
| GCNII |  | 13.72 | 0.005 | 348 | 39.44 | 4.88 | 0.21 | 0.20 | 0.78 |  | 16.13 | 0.006 | 384 | 42.00 | 5.60 | 2.41 | 0.38 | 1.44 |
| GPRGNN |  | 2.76 | 0.003 | 168 | 16.43 | 4.89 | 0.10 | 0.54 | 2.05 |  | 3.73 | 0.005 | 148 | 25.17 | 5.61 | 0.99 | 1.23 | 4.71 |
| GCNJK |  | 8.03 | 0.004 | 369 | 21.77 | 4.89 | 1.61 | 1.61 | 6.13 |  | 14.80 | 0.019 | 184 | 80.45 | 5.61 | 9.07 | 3.00 | 11.44 |
| MixHop | squirrel | 5.10 | 0.001 | 213 | 23.94 | 4.91 | 0.57 | 5.60 | 21.39 | penn94 | 5.31 | 0.001 | 342 | 15.52 | 5.62 | 2.43 | 4.30 | 16.40 |
| LINKX |  | 0.82 | 0.000 | 155 | 5.28 | 4.89 | 0.17 | 2.00 | 7.63 |  | 6.26 | 0.001 | 110 | 56.87 | 5.64 | 1.34 | 12.00 | 45.77 |
| GGCN |  | 379.63 | 0.444 | 119 | 3190.20 | 5.18 | 2.77 | 1.07 | 4.09 |  | 1431.42 | 1.540 | 111 | 12895.70 | 5.95 | 11.42 | 2.47 | 9.41 |
| FSGNN |  | 0.17+5.39 | 0.002 | 500 | 10.80 | 5.21 | 0.81 | 1.21 | 4.60 |  | 2.31+16.41 | 0.054 | 177 | 92.27 | 7.84 | 13.55 | 2.77 | 10.58 |
| ACM |  | 8.97 | 0.002 | 477 | 18.80 | 5.19 | 0.20 | 1.61 | 6.14 |  | 10.89 | 0.009 | 173 | 62.90 | 5.97 | 2.01 | 3.70 | 14.11 |
| GloGNN++ |  | 3.02 | 0.003 | 105 | 28.50 | 5.17 | 0.01 | 0.47 | 3.59 |  | 118.13 | 0.172 | 142 | 826.10 | 6.04 | 3.43 | 24.00 | 183.07 |
| LD$^2$ |  | 3.87+0.43 | 0.001 | 99 | 4.37 | 5.23 | 0.42 | 5.77 | 22.03 |  | 27.19+0.40 | 0.001 | 87 | 4.60 | 8.04 | 3.72 | 11.45 | 43.66 |
| APPNP |  | 5.15 | 0.001 | 500 | 10.30 | 4.94 | 0.74 | 0.03 | 0.13 |  | 3.32 | 0.002 | 250 | 13.29 | 4.89 | 1.78 | 0.00 | 0.02 |
| SGC |  | 0.59 | 0.001 | 210 | 2.80 | 4.94 | 0.27 | 0.00 | 0.00 |  | 0.31 | 0.001 | 128 | 2.45 | 4.90 | 0.21 | 0.00 | 0.00 |
| GCN |  | 15.74 | 0.006 | 500 | 31.47 | 4.94 | 1.23 | 0.01 | 0.03 |  | 7.00 | 0.007 | 215 | 32.56 | 4.89 | 1.07 | 0.00 | 0.00 |
| GCNII |  | 33.46 | 0.006 | 488 | 68.56 | 4.96 | 3.27 | 0.08 | 0.30 |  | 2.69 | 0.002 | 103 | 26.10 | 4.89 | 3.87 | 0.02 | 0.07 |
| GPRGNN |  | 11.19 | 0.004 | 500 | 22.38 | 4.94 | 0.82 | 0.03 | 0.13 |  | 10.14 | 0.006 | 324 | 31.29 | 4.89 | 1.78 | 0.00 | 0.02 |
| GCNJK |  | 39.59 | 0.019 | 500 | 79.17 | 4.94 | 8.49 | 0.60 | 2.30 |  | 39.59 | 0.019 | 500 | 79.17 | 4.94 | 8.49 | 0.60 | 2.30 |
| MixHop | arxiv-year | 8.18 | 0.001 | 499 | 16.39 | 4.94 | 6.28 | 0.16 | 0.60 | genius | 3.55 | 0.001 | 337 | 10.53 | 4.88 | 7.67 | 0.16 | 0.60 |
| LINKX |  | 9.21 | 0.001 | 122 | 75.52 | 5.10 | 2.01 | 43.52 | 166.01 |  | 12.16 | 0.001 | 103 | 118.07 | 5.29 | 4.58 | 108.13 | 412.49 |
| GGCN |  | (OOM) |  |  |  |  |  |  |  |  | (OOM) |  |  |  |  |  |  |  |
| FSGNN |  | 0.17+47.74 | 0.064 | 441 | 108.90 | 5.39 | 3.26 | 0.08 | 0.29 |  | 0.08+60.62 | 0.062 | 469 | 129.33 | 5.19 | 6.16 | 0.01 | 0.03 |
| ACM |  | 49.13 | 0.011 | 500 | 98.30 | 6.22 | 1.69 | 0.10 | 0.39 |  | 84.27 | 0.031 | 400 | 210.70 | 5.25 | 3.53 | 0.01 | 0.04 |
| GloGNN++ |  | 106.30 | 0.109 | 116 | 916.40 | 5.88 | 6.98 | 87.04 | 664.03 |  | 306.54 | 0.214 | 148 | 2071.20 | 6.79 | 14.67 | 216.26 | 1649.95 |
| LD$^2$ |  | 2.83+0.91 | 0.001 | 194 | 4.77 | 5.55 | 4.29 | 1.71 | 6.52 |  | 0.79+0.90 | 0.002 | 133 | 6.77 | 5.23 | 10.06 | 2.16 | 8.24 |
| APPNP |  | 4.76 | 0.001 | 500 | 9.53 | 4.97 | 1.03 | 0.00 | 0.01 |  | 19.91 | 0.009 | 500 | 39.82 | 5.74 | 7.44 | 0.02 | 0.07 |
| SGC |  | 0.65 | 0.001 | 126 | 5.12 | 4.98 | 0.99 | 0.00 | 0.00 |  | 7.81 | 0.004 | 500 | 15.62 | 5.74 | 3.74 | 0.00 | 0.00 |
| GCN |  | 7.95 | 0.003 | 500 | 15.90 | 4.98 | 3.01 | 0.00 | 0.00 |  | 27.55 | 0.013 | 500 | 55.09 | 5.74 | 8.20 | 0.00 | 0.01 |
| GCNII |  | 23.10 | 0.006 | 234 | 98.74 | 4.99 | 3.45 | 0.07 | 0.27 |  | 84.30 | 0.005 | 500 | 168.60 | 5.76 | 16.97 | 0.02 | 0.08 |
| GPRGNN |  | 11.28 | 0.004 | 500 | 22.57 | 4.97 | 1.03 | 0.00 | 0.01 |  | 42.23 | 0.020 | 500 | 84.46 | 5.74 | 7.44 | 0.02 | 0.07 |
| GCNJK |  | 24.59 | 0.011 | 500 | 49.18 | 5.01 | 10.64 | 0.02 | 0.07 |  | 28.40 | 0.013 | 500 | 56.80 | 5.74 | 10.86 | 0.00 | 0.01 |
| MixHop | twitch-gamers | 6.10 | 0.001 | 499 | 12.21 | 4.97 | 3.85 | 0.15 | 0.60 | pokec | 11.46 | 0.004 | 499 | 22.97 | 5.74 | 17.76 | 0.05 | 0.20 |
| LINKX |  | 30.27 | 0.002 | 108 | 280.25 | 5.13 | 2.71 | 43.17 | 164.69 |  | 140.68 | 0.009 | 122 | 1153.10 | 7.29 | 19.57 | 418.15 | 1595.10 |
| GGCN |  | (OOM) |  |  |  |  |  |  |  |  | (OOM) |  |  |  |  |  |  |  |
| FSGNN |  | 0.19+45.55 | 0.118 | 500 | 91.10 | 5.53 | 2.77 | 0.01 | 0.02 |  | (OOM) |  |  |  |  |  |  |  |
| ACM |  | 26.01 | 0.038 | 104 | 250.10 | 6.28 | 2.06 | 0.01 | 0.03 |  | 124.71 | 0.145 | 111 | 1123.50 | 7.98 | 16.47 | 0.05 | 0.20 |
| GloGNN++ |  | 156.76 | 0.215 | 114 | 1375.00 | 5.87 | 7.19 | 86.34 | 658.74 |  | 200.56 | 0.214 | 140 | 1422.40 | 7.56 | 18.98 | 209.02 | 1594.73 |
| LD$^2$ |  | 0.85+0.81 | 0.001 | 145 | 5.57 | 5.25 | 3.99 | 1.48 | 5.66 |  | 17.95+0.99 | 0.001 | 274 | 3.60 | 7.23 | 8.06 | 0.05 | 0.18 |
| APPNP |  | 33.84 | 0.016 | 500 | 67.68 | 7.99 | 14.19 | 0.07 | 0.27 |  | (OOM) |  |  |  |  |  |  |  |
| SGC |  | 2.75 | 0.004 | 145 | 18.95 | 7.97 | 5.16 | 0.00 | 0.01 |  | (OOM) |  |  |  |  |  |  |  |
| GCN |  | 25.66 | 0.012 | 500 | 51.32 | 8.00 | 8.94 | 0.01 | 0.03 |  | (OOM) |  |  |  |  |  |  |  |
| GCNII |  | (OOM) |  |  |  |  |  |  |  |  | (OOM) |  |  |  |  |  |  |  |
| GPRGNN |  | (OOM) |  |  |  |  |  |  |  |  | (OOM) |  |  |  |  |  |  |  |
| GCNJK |  | 26.41 | 0.012 | 500 | 52.81 | 7.97 | 12.60 | 0.01 | 0.04 |  | (OOM) |  |  |  |  |  |  |  |
| MixHop | snap-patents | 8.36 | 0.003 | 499 | 16.75 | 7.97 | 20.79 | 0.06 | 0.21 | wiki | (OOM) |  |  |  |  |  |  |  |
| LINKX |  | 51.31 | 0.006 | 156 | 328.93 | 8.52 | 11.41 | 175.26 | 668.56 |  | (OOM) |  |  |  |  |  |  |  |
| GGCN |  | (OOM) |  |  |  |  |  |  |  |  | (OOM) |  |  |  |  |  |  |  |
| FSGNN |  | (OOM) |  |  |  |  |  |  |  |  | (OOM) |  |  |  |  |  |  |  |
| ACM |  | (OOM) |  |  |  |  |  |  |  |  | (OOM) |  |  |  |  |  |  |  |
| GloGNN++ |  | (OOM) |  |  |  |  |  |  |  |  | (OOM) |  |  |  |  |  |  |  |
| LD$^2$ |  | 31.32+1.80 | 0.001 | 172 | 3.67 | 24.14 | 16.80 | 0.08 | 0.29 |  | 28.12+0.99 | 0.001 | 253 | 3.90 | 33.82 | 19.91 | 0.16 | 0.63 |

### D.2 Efficiency Results

We then present the corresponding results on efficiency metrics for the minibatch and full-batch experiments in Tables 8 and 9, respectively.

Beside the time and memory metrics used in Section 4, we also include a wide range of efficiency measurements, presenting the model performance from different perspectives. To study the training procedure, we further record the convergence epoch number and average training epoch time. We also estimate the size of learnable parameters as well as the overall model size including parameters and buffer. These metrics are useful in explaining the GPU memory footprint occupied by the model architecture.

Summarized from the results, we again highlight the efficiency and scalability of $LD^2$ including fast training and inference and low GPU memory usage. In particular, for full-batch training on wiki, the model manages to finish in 1 second and 20GB of GPU memory, which is the only model to achieve this without the out-of-memory error.

By comparing minibatch and full-batch training, it can be inferred that the time efficiency of $LD^2$ is even superior for full-batch settings, reaching up to $200\times$ acceleration than previous models (pokec, GloGNN++). Its inference speed is also fast, mostly within 1ms. However, its memory overhead relatively increase while the model size being constant. This is the natural consequence of full-batch execution that loads all node in the GPU device, hence demands more space for representation and loses the superiority of $O(n_b)$ in memory complexity.

For other baselines, we notice LINKX and GloGNN++, both utilizing full-graph feature as input, exhibit significantly large model sizes. On graphs such as genius and pokec, they even demand up to 2GB solely for storing model weight parameters. The expanding model size may account for a series of scalability issues on large graphs, including slower convergence and longer forward time. Other efficacy-oriented advanced models, such as ACM and GGCN, are also less efficient due to the complicated calculations in training epochs, while compact designs such as GCNJK and MixHop are more scalable, experiencing less OOM errors. For non-heterophilous models, the only architecture of comparable efficiency performance to $LD^2$ is SGC, demonstrating the benefits of pre-propagation decoupling. Post-propagation approaches APPNP and GPRGNN show limited scalability, which aligns our analysis in Appendix B.

## E  Extended Experiments under Homophily

### E.1  Modification for Homophily

As analyzed in Section 3, the channel embeddings proposed by our work is specifically designed for graphs under heterophily, hence filtering out homophilous information. To enhance expressiveness when migrating to homophilous graphs, LD2 can be modified with different channels to fit different graph patterns. For a preliminary attempt, we additionally consider the following channels as the model input when training $LD^2$ under homophily:

- The 1-hop adjacency spectral embedding: $\boldsymbol{P}_{A,1} = \arg\min_{\boldsymbol{P} \in \mathbb{R}^{n \times F}} \|\boldsymbol{A} - \boldsymbol{P}\boldsymbol{P}^T\|_F^2$;

- The constant 1-hop adjacency propagation with self-loops $\boldsymbol{P}_{X,L} = \frac{1}{L_{P,L}} \sum_{l=1}^{L_{P,L}} \tilde{\boldsymbol{A}}^l \boldsymbol{X}$;

- The constant 1-hop adjacency propagation without self-loops $\boldsymbol{P}_{X,L'} = \frac{1}{L_{P,L'}} \sum_{l=1}^{L_{P,L'}} \bar{\boldsymbol{A}}^l \boldsymbol{X}$;

- The raw node attributes $\boldsymbol{P}_{X,0} = \boldsymbol{X}$.

### E.2  Efficacy Results

We also include experiments on 8 large-scale homophilous datasets, as the evaluation of the generality of $LD^2$. Similarly, the accuracy of minibatch and full-batch models are in Tables 10 and 11, respectively. Note that the implementation of [26] easily exceeds GPU memory when performing multilabel predictions, causing a series of model triggering OOM errors on protein and yelp.

Results in Tables 10 and 11 show that by amending homophilous channels to gather low-frequency information, $LD^2$ is able to achieve comparable performance on all datasets. Its minibatch performance is generally better than both homophilous and non-homophilous competitors. For large-scale

Table 10: Average test accuracy (%) of minibatch $LD^2$ and baselines on 7 homophilous datasets. "OOM" means the model occurs out of memory error with applicable hyperparameters.

| Dataset | cora | pubmed | protein | ogbn-arxiv | yelp | reddit | amazon |
|---------|------|--------|---------|------------|------|--------|--------|
| MLP | 73.39 | 87.04 | 32.41 | 57.57 | 42.43 | 63.79 | 70.41 |
| PPRGo | 85.85 | 88.88 | 41.24 | 68.00 | 55.49 | 61.49 | 87.92 |
| SGC | 80.86 | 87.83 | 44.67 | 69.44 | 9.29 | 14.88 | 28.77 |
| GCNJK-GS | 77.73 | 81.93 | 40.63 | 60.03 | (OOM) | 22.94 | 84.75 |
| MixHop-GS | 77.09 | 85.45 | 39.85 | 60.38 | (OOM) | 67.11 | 85.96 |
| LINKX | 87.70 | 85.38 | 87.48 | 72.35 | (OOM) | 19.56 | 89.85 |
| $LD^2$ | 87.66 | 89.19 | 97.18 | 73.48 | 59.14 | 72.99 | 89.54 |

Table 11: Average test accuracy (%) of full-batch $LD^2$ and baselines on 5 homophilous datasets. "OOM" means the model occurs out of memory error with applicable hyperparameters.

| Dataset | protein | ogbn-arxiv | yelp | reddit | amazon |
|---------|---------|------------|------|--------|--------|
| APPNP | 43.57 | 66.95 | (OOM) | (OOM) | (OOM) |
| SGC | 27.10 | 54.02 | (OOM) | 54.70 | 52.29 |
| GCN | 37.14 | 71.56 | (OOM) | (OOM) | (OOM) |
| GCNII | 19.37 | 70.00 | (OOM) | 45.21 | (OOM) |
| GPRGNN | (OOM) | 72.98 | (OOM) | (OOM) | (OOM) |
| GCNJK | 52.95 | 73.38 | (OOM) | (OOM) | (OOM) |
| MixHop | 76.28 | 74.39 | (OOM) | 58.17 | 88.14 |
| LINKX | 99.78 | 72.37 | (OOM) | 13.55 | 89.33 |
| ACM | 53.69 | 67.08 | (OOM) | (OOM) | (OOM) |
| GloGNN++ | (OOM) | 71.08 | (OOM) | (OOM) | (OOM) |
| $LD^2$ | 75.47 | 71.52 | 57.43 | 70.05 | 85.22 |

Table 12: Evaluation of minibatch $LD^2$ and baselines on 3 homophilous datasets with fixed splits. "OOM" means the model occurs out of memory error with applicable hyperparameters.

| Dataset | protein-ind | | | | yelp-ind | | | | ogbn-papers | | | |
|---------|------|-------|-------|------|------|-------|--------|-------|------|-------|-------|-------|
| Model | Acc | Learn | Infer | RAM | Acc | Learn | Infer | RAM | Acc | Learn | Infer | RAM |
| GCN-GS | 89.15 | 97.77 | 2.888 | 8.94 | 65.03 | 799.81 | 30.732 | 55.22 | | (OOM) | | |
| PPRGo | 48.33 | 7.40+33.37 | 0.816 | 6.98 | 26.25 | 6.29+130.37 | 18.315 | 9.88 | | (OOM) | | |
| $LD^2$ | 90.87 | 0.48+3.86 | 0.270 | 5.20 | 61.63 | 3.21+2.90 | 1.908 | 8.64 | 50.37 | 123.15+52.60 | 4.690 | 105.32 |

full-batch training, $LD^2$ is the only models that does not exceed memory limit on all datasets, again demonstrating its scalability regardless of graph heterophily.

Specifically, we here explore the capability of $LD^2$ on handling inductive tasks. This can be implemented by conducting precomputation respectively on the training and inference graphs. After that, the feature transformation model can be easily trained on the precomputed embeddings of the training graph and then perform inductive inference based on the embeddings of the other graph. Results are shown in Table 12. As a brief summary, $LD^2$ achieves comparable accuracy with GCN-GS, while PPRGo fails to adapt such settings.

### E.3 Efficiency Results

We similarly explore the efficiency metrics of $LD^2$ and 16 baselines on the homophilous graphs, with results shown in Tables 13 and 14. Our main conclusion still hold, that $LD^2$ exhibit great scalability with respect to learning time and memory footprint compared to both homophilous and heterophilous models. Its metrics on yelp and amazon explain that the success of finishing learning on these large datasets is achieved by the compact and efficient architectural design.

Table 13: Efficiency evaluation of minibatch LD$^2$ and baselines on homophilous datasets.

| Model | Dataset | Learn | Infer | # | Avg | RAM | GPU | Param. | Size | Dataset | Learn | Infer | # | Avg | RAM | GPU | Param. | Size |
|---|---|---|---|---|---|---|---|---|---|---|---|---|---|---|---|---|---|---|
| MLP | | 0.68 | 0.003 | 138 | 4.90 | 5.13 | 0.04 | 1.00 | 3.83 | | 1.35 | 0.005 | 243 | 5.60 | 5.21 | 0.05 | 0.52 | 2.00 |
| PPRGo | | 0.41+1.24 | 0.014 | 136 | 9.10 | 6.42 | 2.12 | 0.37 | 1.41 | | 0.90+10.06 | 0.029 | 332 | 30.30 | 8.71 | 7.16 | 0.13 | 0.49 |
| SGC | | 0.27 | 0.001 | 64 | 4.30 | 5.07 | 0.04 | 0.37 | 1.41 | | 1.72 | 0.002 | 183 | 9.40 | 5.12 | 0.06 | 0.13 | 0.50 |
| GCNJK-GS | cora | 4.82 | 0.008 | 104 | 0.05 | 4.97 | 0.15 | 1.27 | 4.86 | pubmed | 10.77 | 0.011 | 113 | 0.10 | 5.05 | 0.98 | 0.79 | 3.02 |
| MixHop-GS | | 12.16 | 0.008 | 181 | 0.07 | 4.97 | 0.30 | 4.60 | 17.58 | | 40.31 | 0.013 | 336 | 0.12 | 5.09 | 0.97 | 3.15 | 12.04 |
| LINKX | | 0.82 | 0.004 | 178 | 0.00 | 4.86 | 0.04 | 1.14 | 4.34 | | 1.11 | 0.005 | 195 | 0.01 | 4.91 | 0.14 | 5.31 | 20.25 |
| LD$^2$ | | 0.07+0.94 | 0.001 | 127 | 7.40 | 5.12 | 0.09 | 1.44 | 5.48 | | 0.04+2.15 | 0.004 | 85 | 25.33 | 5.20 | 0.08 | 0.72 | 2.75 |
| MLP | | 1.33 | 1.845 | 102 | 13.06 | 5.10 | 13.09 | 0.62 | 2.36 | | 8.08 | 0.024 | 500 | 16.17 | 5.16 | 0.62 | 0.61 | 2.36 |
| PPRGo | | 6.10+10.70 | 0.799 | 500 | 21.40 | 5.68 | 1.32 | 0.04 | 0.17 | | 8.92+147.71 | 0.809 | 426 | 346.47 | 13.65 | 2.73 | 0.11 | 0.41 |
| SGC | | 0.08+1.69 | 0.000 | 312 | 5.43 | 5.16 | 0.35 | 0.02 | 0.07 | | 1.74+1.79 | 0.010 | 220 | 8.10 | 5.21 | 0.38 | 0.04 | 0.17 |
| GCNJK-GS | protein | 1.33 | 1.845 | 102 | 13.06 | 5.10 | 13.09 | 1.07 | 4.07 | ogbn-arxiv | 8.08 | 0.024 | 500 | 16.17 | 5.16 | 0.62 | 0.94 | 3.59 |
| MixHop-GS | | 1.33 | 1.845 | 102 | 13.06 | 5.10 | 13.09 | 3.05 | 11.65 | | 8.08 | 0.024 | 500 | 16.17 | 5.16 | 0.62 | 2.75 | 10.53 |
| LINKX | | 4.36 | 1.080 | 500 | 8.71 | 5.10 | 13.27 | 14.75 | 56.28 | | 4.37 | 0.067 | 204 | 21.43 | 5.18 | 1.19 | 43.53 | 166.04 |
| LD$^2$ | | 0.16+2.98 | 0.002 | 403 | 7.39 | 5.24 | 0.64 | 0.36 | 1.38 | | 9.07+1.81 | 0.002 | 147 | 12.31 | 5.84 | 3.91 | 1.74 | 6.64 |
| MLP | | 50.30 | 0.213 | 500 | 100.59 | 13.06 | 0.91 | 0.86 | 3.29 | | 90.22 | 0.258 | 500 | 180.43 | 13.76 | 6.20 | 0.60 | 2.32 |
| PPRGo | | 8.56+44.16 | 25.500 | 499 | 51.00 | 18.15 | 2.29 | 0.16 | 0.63 | | 74.04+2408.53 | 184.590 | 499 | 369.18 | 37.35 | 16.72 | 0.16 | 0.63 |
| SGC | | 2.62+1.02 | 0.000 | 118 | 8.67 | 5.83 | 0.52 | 0.17 | 0.63 | | 2.36+8.94 | 0.010 | 155 | 57.71 | 6.95 | 0.37 | 0.04 | 0.15 |
| GCNJK-GS | reddit | 50.30 | 0.213 | 500 | 100.59 | 13.06 | 0.91 | 0.24 | 0.92 | amazon | 90.22 | 0.258 | 500 | 180.43 | 13.76 | 6.20 | 0.94 | 3.59 |
| MixHop-GS | | 50.30 | 0.213 | 500 | 100.59 | 13.06 | 0.91 | 3.49 | 13.33 | | 90.22 | 0.258 | 500 | 180.43 | 13.76 | 6.20 | 2.75 | 10.50 |
| LINKX | | 123.18 | 0.949 | 299 | 411.98 | 13.05 | 2.53 | 59.94 | 228.63 | | 170.52 | 1.281 | 292 | 583.96 | 13.76 | 19.26 | 614.73 | 2344.99 |
| LD$^2$ | | 1.98+1.23 | 0.002 | 102 | 11.99 | 7.51 | 3.66 | 2.12 | 8.09 | | 45.01+8.31 | 0.004 | 121 | 68.70 | 9.14 | 4.54 | 0.87 | 3.32 |

Table 14: Efficiency evaluation of full-batch LD$^2$ and baselines on homophilous datasets.

| Model | Dataset | Learn | Infer | # | Avg | RAM | GPU | Param. | Size | Dataset | Learn | Infer | # | Avg | RAM | GPU | Param. | Size |
|---|---|---|---|---|---|---|---|---|---|---|---|---|---|---|---|---|---|---|
| APPNP | | 4.38 | 0.098 | 500 | 8.76 | 4.98 | 19.72 | 0.04 | 0.17 | | 4.76 | 0.001 | 500 | 9.51 | 4.96 | 1.78 | 0.04 | 0.17 |
| SGC | | 0.34 | 0.051 | 101 | 3.34 | 4.98 | 17.40 | 0.01 | 0.02 | | 1.65 | 0.001 | 500 | 3.30 | 4.97 | 0.53 | 0.01 | 0.02 |
| GCN | | 3.34 | 0.121 | 486 | 6.87 | 4.97 | 19.19 | 0.01 | 0.02 | | 18.21 | 0.009 | 478 | 38.09 | 4.97 | 2.12 | 0.01 | 0.04 |
| GCNII | | 2.52 | 0.206 | 101 | 24.96 | 4.97 | 17.96 | 0.02 | 0.09 | | 37.24 | 0.006 | 500 | 74.48 | 4.97 | 3.41 | 0.08 | 0.31 |
| GPRGNN | | (OOM) | | | | | | | | | 11.26 | 0.004 | 500 | 22.52 | 5.00 | 2.16 | 0.04 | 0.17 |
| GCNJK | protein | 16.33 | 2.175 | 212 | 77.03 | 4.97 | 14.51 | 0.74 | 2.83 | ogbn-arxiv | 36.22 | 0.033 | 265 | 136.68 | 4.96 | 15.13 | 0.65 | 2.51 |
| MixHop | | 5.51 | 0.120 | 499 | 11.05 | 4.98 | 19.08 | 0.96 | 3.66 | | 6.77 | 0.001 | 451 | 15.01 | 4.96 | 6.42 | 0.79 | 3.02 |
| LINKX | | 73.89 | 0.098 | 499 | 148.07 | 4.98 | 17.79 | 14.75 | 56.28 | | 18.94 | 0.001 | 149 | 127.11 | 5.12 | 2.44 | 43.53 | 166.04 |
| GGCN | | (OOM) | | | | | | | | | (OOM) | | | | | | | | |
| ACM | | 89.89 | 0.804 | 500 | 179.80 | 6.14 | 14.71 | 0.13 | 0.51 | | 76.28 | 0.019 | 500 | 152.60 | 6.22 | 1.94 | 0.13 | 0.50 |
| GloGNN++ | | (OOM) | | | | | | | | | 238.53 | 0.145 | 209 | 1141.30 | 5.88 | 7.54 | 87.05 | 664.16 |
| LD$^2$ | | 0.16+0.56 | 0.001 | 180 | 3.17 | 5.09 | 0.27 | 0.10 | 0.36 | | 9.07+0.38 | 0.001 | 114 | 3.30 | 5.43 | 0.92 | 0.11 | 0.40 |
| APPNP | | (OOM) | | | | | | | | | (OOM) | | | | | | | | |
| SGC | | 29.20 | 0.014 | 500 | 58.40 | 7.93 | 15.38 | 0.02 | 0.09 | | 22.31 | 0.018 | 304 | 73.40 | 8.21 | 17.58 | 0.00 | 0.02 |
| GCN | | (OOM) | | | | | | | | | (OOM) | | | | | | | | |
| GCNII | | 198.17 | 0.015 | 500 | 396.35 | 7.93 | 17.09 | 0.04 | 0.15 | | (OOM) | | | | | | | | |
| GPRGNN | | (OOM) | | | | | | | | | (OOM) | | | | | | | | |
| GCNJK | reddit | (OOM) | | | | | | | | amazon | (OOM) | | | | | | | | |
| MixHop | | 33.66 | 0.015 | 499 | 67.45 | 7.93 | 18.85 | 1.16 | 4.42 | | 38.37 | 0.018 | 499 | 76.89 | 8.21 | 20.98 | 0.03 | 0.12 |
| LINKX | | 518.21 | 0.040 | 150 | 3454.72 | 8.15 | 13.71 | 59.94 | 228.64 | | 506.61 | 0.043 | 101 | 5015.91 | 8.77 | 19.78 | 153.66 | 586.15 |
| GGCN | | (OOM) | | | | | | | | | (OOM) | | | | | | | | |
| ACM | | (OOM) | | | | | | | | | (OOM) | | | | | | | | |
| GloGNN++ | | (OOM) | | | | | | | | | (OOM) | | | | | | | | |
| LD$^2$ | | 1.98+0.39 | 0.001 | 118 | 3.27 | 6.79 | 2.13 | 0.14 | 0.54 | | 45.01+2.02 | 0.001 | 165 | 4.07 | 9.62 | 7.30 | 0.04 | 0.15 |

# F  Effect of Parameters

## F.1  Convergence Curve

To examine the effect of model and training settings, in Figure 5, we display the model convergence curve, i.e. validation accuracy versus training time on heterophilous datasets and minibatch models corresponding to Table 2. It can be obviously observed that LD$^2$ outperforms other baseline methods on most datasets, demonstrating more stable curve, faster convergence, and significantly shorter overall training time. It is worth noting that the convergence of some baselines is beyond the display scopes in Figure 5.

Among other baselines, on small graphs, LINKX is relatively fast compared to GCNJK and MixHop which generally take more time per epoch. However, its large parameter space results in unstable performance, and hence requires more epochs to converge. When the graph scales larger, the efficiency of LINKX degrades due to its full-graph dependency. For simple and non-heterophilous models, though the decoupling design benefits them for less epoch time, their accuracies are suboptimal, and hence experience more training epochs than LD$^2$. Particularly, the PPRGo model is so large that it overfits on validation sets of small graphs such as squirrel.

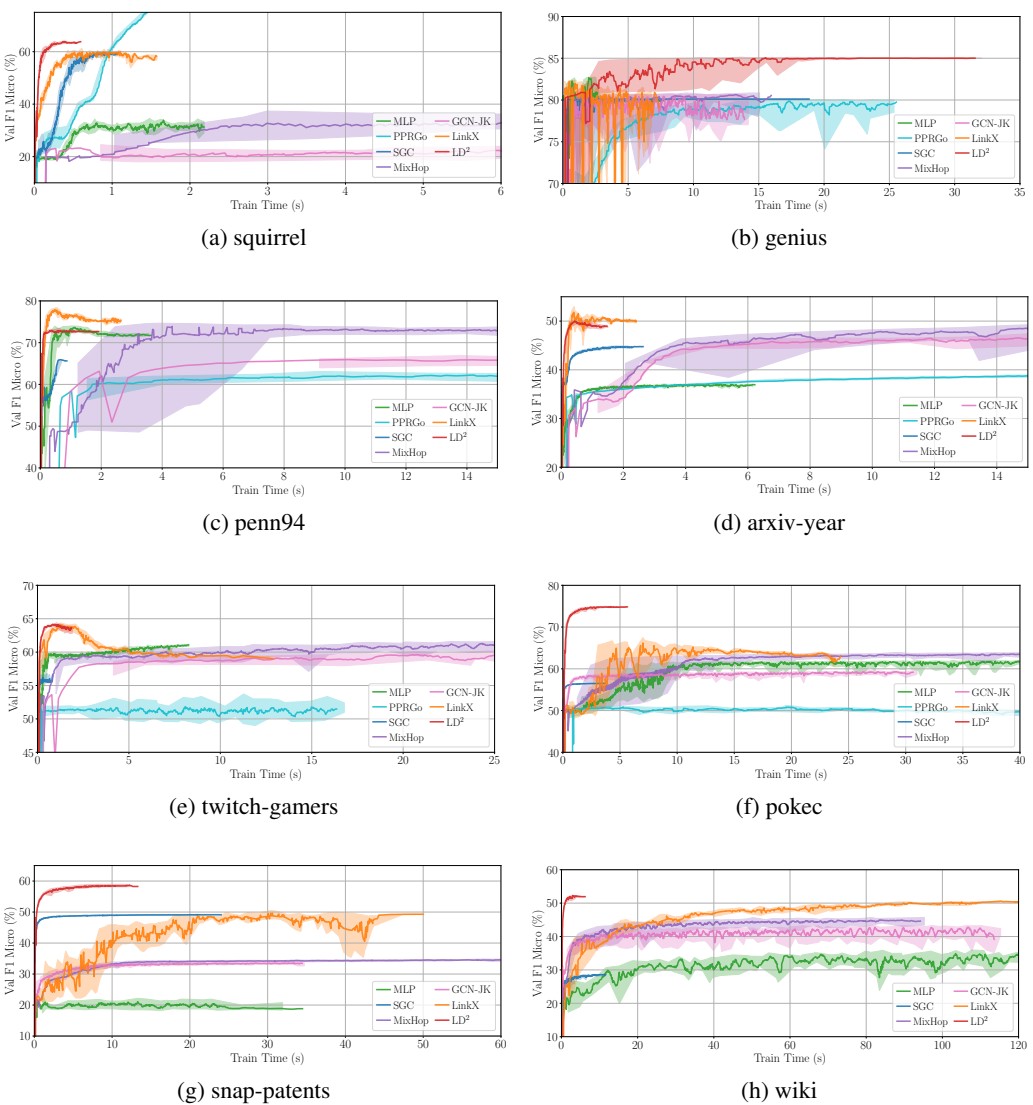

Figure 5: Validation accuracy convergence curves of minibatch LD$^2$ and baseline models on 8 heterophilous datasets. Curves only represents the process of the training phase. Shaded area is the result range of multiple runs.

## F.2   Embedding Schemes and Robustness

We conduct additional exploration on validating the effectiveness of utilizing $A^2$ adjacency spectral embedding (ASE) as the LD$^2$ embedding scheme. Experiments of other schemes are shown in Table 15, including the shortest path distance used as spatial encodings in Graphormer SPD [59] and node2vec embedding [60]. There are also rank-$F$ approximations in Eq. (2) by replacing $A^2$ with $A$ and $A^3$, which are respectively denoted as ASE($A$) and ASE($A^3$).

We compare the advantages of our proposed ASE($A^2$) with other structural embeddings such as SPD from three aspects:

- Regarding effectiveness, as we analyzed in Section 3.2, the ASE embedding is able to capture structural information especially on the homophilous components of the 2-hop graph, while SPD is more specific to encode distance information of directly connected nodes and node2vec represents local neighborhood, which are less suitable for heterophilous graphs.

- As for memory efficiency, ASE is a low-dimensional embedding of shape $n \times F$, where the feature dimension $F$ is generally much smaller than the graph scale. This implies better scalability compared to SPD embedding which is a dense $n \times n$ matrix. Node2vec requires additional $O(nd^2)$ space for storing the random walk results. As the average degree $d = m/n$ is at the scale of $O(\log n)$ to $O(n)$, node2vec is less scalable especially to large and dense graphs, which explains the OOM error on genius.

- ASE also benefits from better time efficiency in our model as described in Section 3.4, which can be computed along with feature embeddings with a complexity linear to edge size $m$, while node2vec exhibits the same $O(nd^2)$ complexity. The node2vec design is based on random walks, which is known to be less efficient for time efficiency and cache locality, which further degrades its scalability. Additionally, our ASE and other embeddings can be efficiently computed by an end-to-end algorithm as described in Section 3.4.

We also conduct experiments to evaluate the model performance under different levels of noise and incompleteness. The results are shown in Table 16. We mainly consider three types of noises, which are analyzed respectively as following:

- Push threshold: We vary the threshold $\delta_P$ in Algorithm 1 to control the precision of propagation. A larger $\delta_P$ implies less precise propagation ignoring small feature values, while $\delta_P = 10^{-5}$ is the original setting. It can be seen that by improving the precision, the final learning accuracy does not change significantly. It indicates that our setting of $\delta_P = 10^{-5}$ is sufficient for propagation and does not affect the learning performance.

- Edge removal: We randomly remove a percentage of edges to generate an incomplete variant of the graph. The LD$^2$ model is then applied to learn on the incomplete graph. The removal

Table 15: Performance of LD$^2$ with alternative adjacency embeddings on selected datasets. Particularly, $P_A = \text{ASE}(A^2)$ indicates the proposed LD$^2$ model with adjacency spectral embedding denoted in Eq. (2).

| $P_A$ | ASE($A^2$) | | | ASE($A$) | | | ASE($A^3$) | | | node2vec | | | SPD | | |
|---|---|---|---|---|---|---|---|---|---|---|---|---|---|---|---|
| Dataset | Acc | Pre. | RAM | Acc | Pre. | RAM | Acc | Pre. | RAM | Acc | Pre. | RAM | Acc | Pre. | RAM |
| squirrel | 66.87 | 3.87 | 0.64 | 60.95 | 2.47 | 0.47 | 62.26 | 3594.90 | 0.61 | 50.19 | 1620.58 | 4.58 | 54.50 | 323.30 | 0.99 |
| penn94 | 75.52 | 27.19 | 0.53 | 74.09 | 1.83 | 0.40 | 74.36 | 306.85 | 0.72 | 73.15 | 24654.00 | 22.82 | | (>12h) | |
| genius | 85.31 | 0.79 | 0.60 | 84.68 | 0.63 | 0.58 | 84.56 | 293.28 | 0.62 | | (OOM) | | | (>12h) | |

Table 16: Performance of LD$^2$ with different noisy data on selected datasets. Particularly, $\delta_P = 10^{-5}$ is the original LD$^2$ model result presented in main experiments.

| Noise | Push Threshold $\delta_P$ | | | Edge Removal | | | Attribute Noise | | |
|---|---|---|---|---|---|---|---|---|---|
| Dataset | $10^{-5}$ | $10^{-6}$ | $10^{-7}$ | 10% | 20% | 40% | $0.5\sigma$ | $1\sigma$ | $2\sigma$ |
| penn94 | 75.52 | 75.56 | 75.57 | 72.63 | 72.11 | 71.53 | 69.39 | 67.41 | 62.57 |
| genius | 85.31 | 85.29 | 85.16 | 84.98 | 84.77 | 84.30 | 81.29 | 81.22 | 81.18 |

causes a negative impact on the accuracy. However, as the node attributes $X$ are kept unchanged under the noise, the model is still able to achieve a reasonable performance.

- Attribute noise: We apply Gaussian noise with standard deviations proportional to the deviation of each feature dimension to the raw node attribute matrix $\boldsymbol{X}$ before precomputation. This is more aggressive as the noise level is much larger than the scale of propagation precision.Consequently, the model suffers a more significant accuracy reduction. However, as the noise level increases, the model's performance converges towards the performance achieved by only learning the adjacency embedding $\boldsymbol{P}_A$. This is because the adjacency information is unaffected under such kind of noise.

## F.3 Propagation Hyperparameters: Ablation Study

We here present our investigation in hyperparameters of the LD$^2$ design. According to Section 3, its tunable hyperparameters include the maximal propagation hop $L_P$ and the graph adjacency normalization coefficient $a, b$. The former hyperparameter affect the most distant scale of feature embeddings and the iteration convergence of adjacency embeddings in A$^2$Prop, while the latter is for $\bar{A}$ or $\tilde{L}$ propagation in feature embeddings.

As listed in Table 5, we explore the range of $L_P$ in $[2, 4, 6, \cdots, 20]$ for each each embeddings per dataset. To study the long-distance information retrieval ability of our model, we also evaluate the effect on different embedding combinations, such as $P_{X,L2}\|P_{X,H}, P_X = P_{X,0}\|P_{X,L2}\|P_{X,H}$, and $P_X\|P_A$. These partial combinations are similarly input into the MLP for training following Eq. (1). Comparison among different channel combinations is useful for studying the effect of each embedding channel.

Beside the representative results on genius and pokec shown in Figure 3, we display the effect of propagation hops $L_P$ on different embeddings on other heterophilous datasets in Figure 6. It can be inferred that our analysis in Section 4 generally holds, while different datasets present some different patterns. For example, the effect of the inverse embedding $P_{X,H}$ decreases when adding multiple hops in graphs such as genius, pokec, penn94, and arxiv-year. However, on the small heterophilous graph squirrel, it reaches maximum when $L_P = 10$, indicating that non-local inter-node relationships are beneficial in this case. We summarize our investigation that despite the pattern of specific channels or graphs, LD$^2$ is capable of gathering useful information from different embedding channels, and its performance generally increases when applying more hops, demonstrating the effectiveness of our long-distance design.

For graph adjacency normalization coefficients $a, b \in [0, 1]$ applied to features, we uniformly use $b = 1 - a$ and only tune the coefficient $a$ for the normalized feature embeddings $P_{X,L2}$ and $P_{X,H}$. We explore the choices respectively for each dataset.

Results of changing normalization are shown in Figure 7 on 9 heterophilous graphs. Different graphs have various favors of adapting the normalization, considering the varying implicit meanings of their features. In general, the accuracy gap is not significant for most varying parameter values. Note that the entire LD$^2$ model comprehensively extracts information from different inputs, we hence state that it is relatively not sensitive to the normalization parameter value.

## F.4 Precomputation Hyperparameters: Scalability Comparison

The maximal propagation hop $L_P$ and feature dimension $F$ also affects the A$^2$Prop precomputation algorithmic efficiency. To study the effect, we change $L_P$ in $[4, 8, 12, 16, 20]$ and the target dimension $F$ in $[F/5, 2F/5, 3F/5, 4F/5, F]$ to perform A$^2$Prop computation.

Results o 6 datasets with respect to precomputation time and RAM memory are displayed in Figure 8. As the graph and feature size scale up, the time and memory overhead of A$^2$Prop also increase, mostly linear to the scale of $n$ and $F$, which is in consistent to our complexity analysis. Figure 8 also implies that, when the graph scale is large, the dominant factor affecting the precomputation time is the number of propagation hops $L_P$, while the influence from $F$ is relatively minor. When varying $L_P$ and $F$, the RAM memory overhead of A$^2$Prop merely changes, indicating its efficient usage.

For precision related hyperparameters in A$^2$Prop, we commonly use relative error bound $\epsilon = 0.1$, norm threshold $\delta = 1 \times 10^{-5}$, and failure probability $\phi = 0.01$.

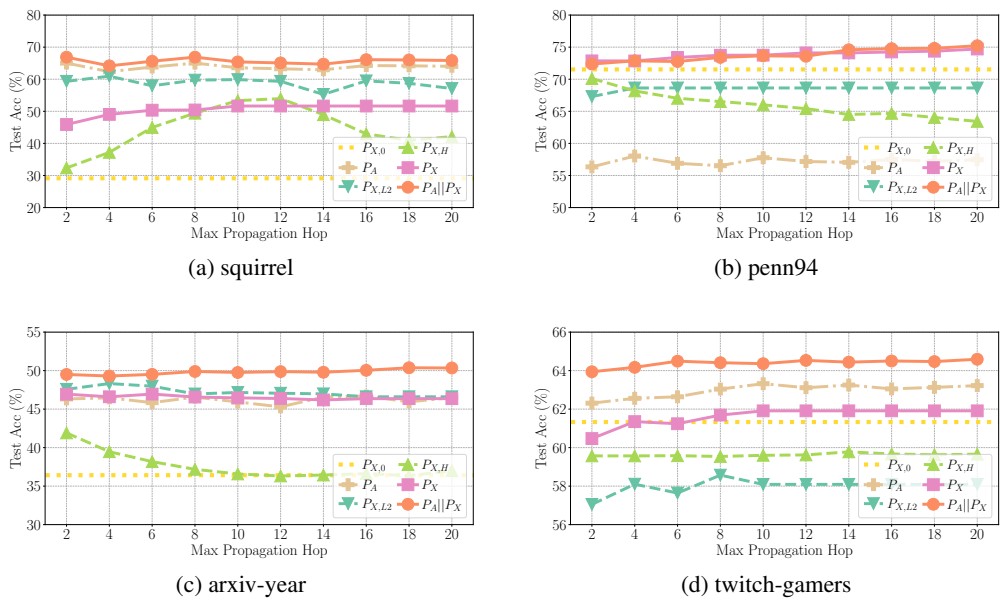

Figure 6: Effect of A$^2$Prop propagation hops on the effectiveness of different adjacency and feature embedding channels and their combinations on 4 heterophilous datasets.

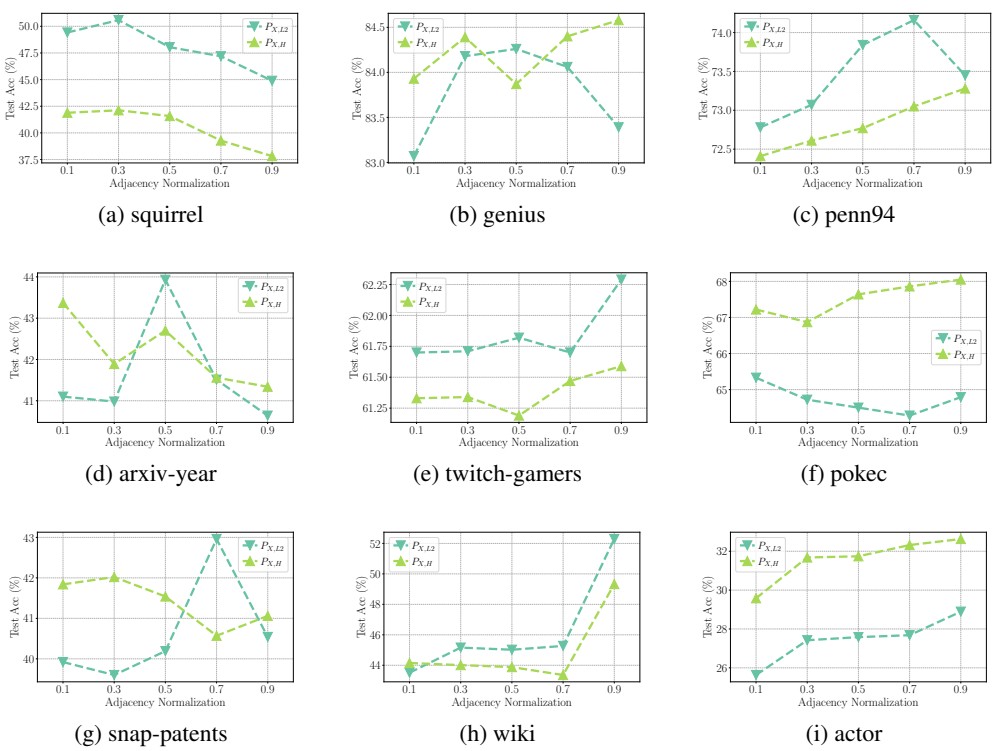

Figure 7: Effect of A$^2$Prop adjacency normalization on the effectiveness of different feature embedding channels on 9 heterophilous datasets.

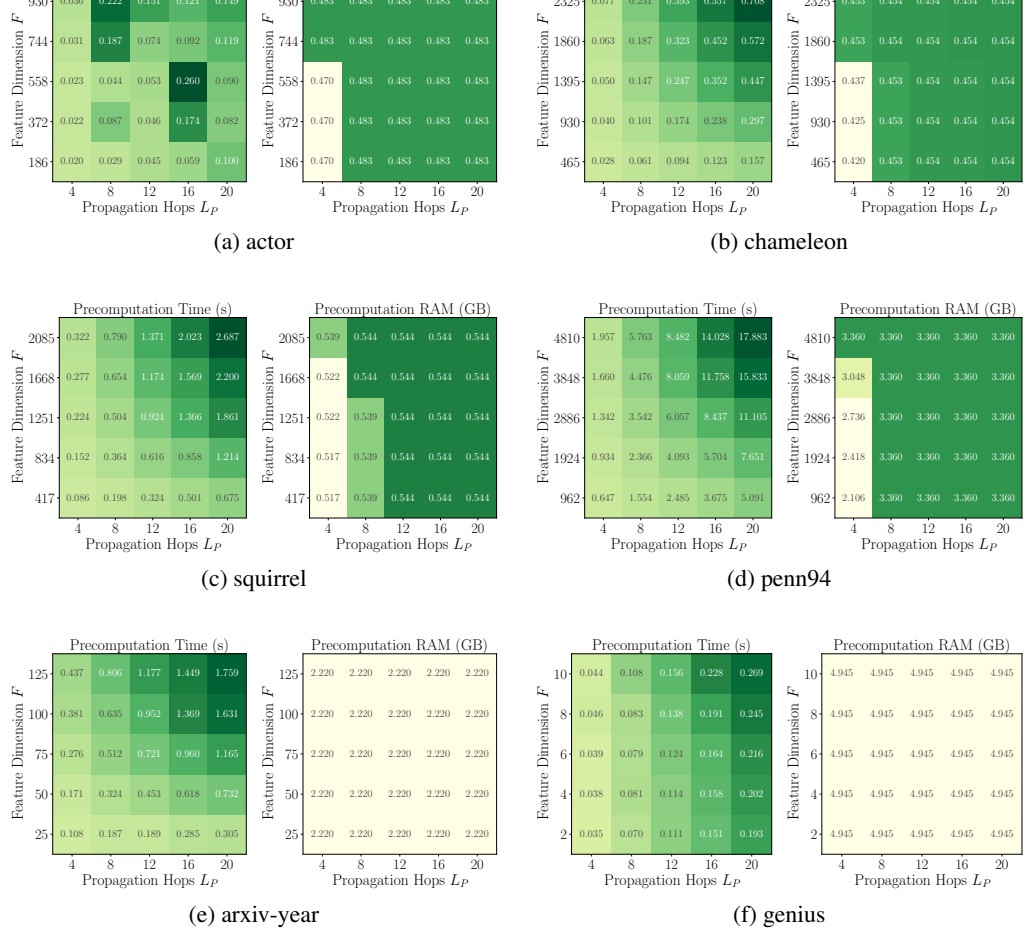

Figure 8: Effect of propagation hops and feature dimensions on A²Prop precomputation time and memory overhead on 6 heterophilous datasets.

## F.5 Directed Edges

While the discussion in the main paper are based on undirected graphs, our LD² design can be easily applied to directed graphs by replacing $A^2$ with $AA^\top$. In our main experiments, we utilize the graphs with directed edges for heterophilous datasets arxiv-year and snap-patents to be aligned with [26]. Here we also evaluate the model performance compared to minibatch baselines on the undirected version. Performance results on snap-patents are shown in Table 17.

Table 17: Average test accuracy (%) and efficiency metrics of minibatch LD² and baselines on snap-patents variants with directed or undirected edges.

| Dataset | snap-patents | | | | | snap-patents | | | | |
|---|---|---|---|---|---|---|---|---|---|---|
| | Acc. | Learn | Infer | RAM | GPU | Acc. | Learn | Infer | RAM | GPU |
| Direction | undirected | | | | | directed | | | | |
| Nodes $n$ | 2,738,035 | | | | | 2,738,035 | | | | |
| Edges $m$ | 30,869,012 | | | | | 16,705,984 | | | | |
| MLP | 23.03 | 27.39 | 0.28 | 9.60 | 9.33 | 23.03 | 27.39 | 0.28 | 9.60 | 9.33 |
| GCNJK-GS | 32.99 | 30.87 | 0.23 | 10.33 | 12.61 | 33.64 | 19.02 | 0.23 | 9.06 | 9.21 |
| MixHop-GS | 33.42 | 74.00 | 0.12 | 10.37 | 21.07 | 34.73 | 45.24 | 0.16 | 9.09 | 19.58 |
| LINKX | 49.78 | 74.54 | 0.60 | 11.3 | 19.96 | 52.69 | 39.80 | 0.22 | 10.87 | 21.53 |
| **LD² (ours)** | 44.28 | 66.58+12.75 | 0.02 | 35.07 | 2.26 | 58.58 | 31.32+6.96 | 0.02 | 31.14 | 3.96 |

For graph statistics, the undirected variant of graph is of the same node size $n$ with the directed one, but about twice the edge size $m$. This is because the edge list representation require to store twice $(u,v)$ and $(v,u)$ for an undirected edge in order to carry out propagation. Consequently, the time and memory overhead of most models also increase. With respect to accuracy, it can be inferred from Table 17 that considering the edge direction is beneficial for GNN accuracy, and $\text{LD}^2$ achieves an increment of nearly 15%. We hence conclude that edge direction is informative for certain heterophilous graph tasks.

## G  Case Study of Approximate Propagation

In order to intuitively illustrate the effect of approximate propagation used in $\text{LD}^2$, here we consider a toy example. Figure 9 depicts a graph with 9 nodes and 10 edges. Its nodes belong to 3 classes, and the connections are mostly heterophilous.

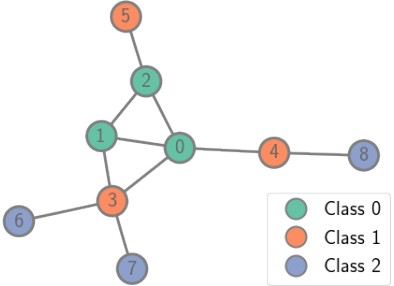

Figure 9: An example heterophilous graph where $\mathcal{H}_{n,1} = 0.204$.

We specifically focus on the inverse Laplacian propagation $\boldsymbol{P}_{X,H} = \frac{1}{L_{P,H}} \sum_{l=1}^{L_{P,H}} (\boldsymbol{I} + \tilde{\boldsymbol{L}})^l \boldsymbol{X}$, as the 2-hop propagation is not suitable for such a small graph.

We first consider the $F = 3$ feature distribution with values in $[-1, 0, 1]$ as shown in the left side of Figure 10. Nodes inside the same class are of the same value in each feature dimension. We then perform $l = 1$ to $8$ times of propagation and illustrate the embedding in the figure. It can be interpreted that the propagation is useful in assigning inverse values to neighboring nodes based on the current ego node embedding. When the number of hops increases, the embedding gradually converges in each feature dimension. It is intuitive in the figure that, with proper steps of propagation, such as $l = 3$ or $4$, it is easy to distinguish nodes in different classes, that their embeddings show different patterns. The Laplacian propagation procedure is hence useful for classifying heterophilous nodes in this case.

In another example in Figure 11, we investigate the one-hot style node feature, where nodes in the same class are assigned with 1 for one feature, and 0 for others. No negative value exists in the raw feature. In this case, the embedding produced by Laplacian propagation quickly converges. When setting $l = 3$ or $4$, it is difficult to distinguish class 0 and 2, since all their nodes exhibit a similar pattern of having negative values in feature dimension $F = 0, 2$ and positive values in $F = 1$.

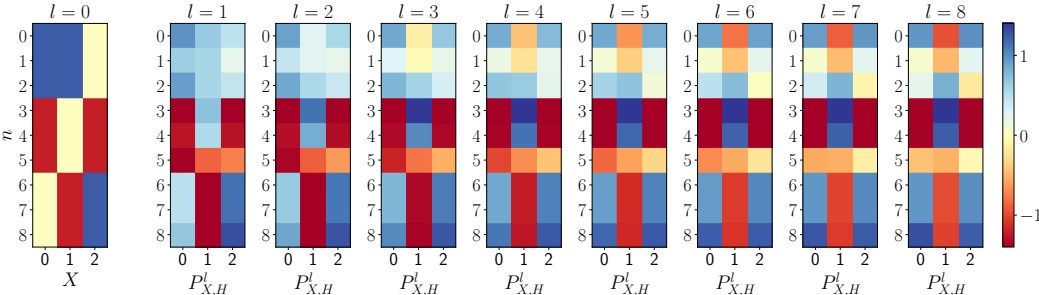

Figure 10: Progression of inverse Laplacian embedding with increasing propagation hops on given positive-negative distribution raw feature.

The example illustrates the propagation procedure of the heterophilous filter. We intend to use the case study to explain the difficulty of $LD^2$ adapting to certain patterns of input features, such as one-hot encoding. We believe this is partially the reason that $LD^2$ with only feature embeddings achieves suboptimal accuracy on graphs with such one-hot features including squirrel and penn94.

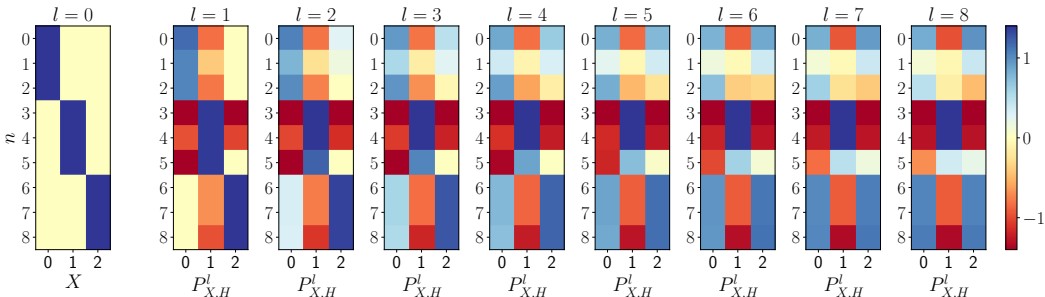

Figure 11: Progression of inverse Laplacian embedding with increasing propagation hops on given one-hot distribution raw feature.

## H Discussion and Limitation

**Limitation.** In the main paper and supplementary experiments, we observe that $LD^2$ exhibits varying performance on graphs with different types of features.In Figure 6 we display the effects of these feature embeddings when changing propagation steps, and in Appendix G we examine a toy model attempting to explain the reason behind the varying propagations. We think that the propagation of $LD^2$ may be less effective for generating expressive embeddings from certain types of features.

As mentioned in Section 3.4, the complexity of our precomputation algorithm $A^2$Prop is $O(L_P mF)$. We also empirically evaluate the effect of these factors on different graphs in Appendix F.4. The evidence indicates that the efficiency bottleneck of the precomputation lies in the linear dependency on the graph and feature size in the algorithm.

**Future Work.** Given these current limitations, we believe that efforts towards more robust precomputation schemes and better adaptability to diverse features could further enhance the non-homophilous model in the future. Although the $A^2$Prop is efficient in implementation, we do recognize that there are graph centrality algorithms and decoupled GNN precomputations reaching sub-linear complexity [50, 53, 25]. $A^2$Prop is potentially configurable for these enhancements. Secondly, various data augmentation approaches are able to transfer the one-hot features to other feature distributions, for instance, by using an embedding model or a simple MLP. Applying $LD^2$ to propagate the augmented embeddings is a promising way to address its limitation on feature distribution.

**Broader Impact.** As our work primarily focuses on theoretical contributions to improve the scalability of graph neural networks, we do not foresee it having a direct negative social impact.