# OpenReview forum: "LD2: Scalable Heterophilous Graph Neural Network with Decoupled Embeddings"
_NeurIPS.cc/2023/Conference — NeurIPS 2023 poster_

### Official Review · Reviewer_4icY · 2023-07-02

**Soundness:** 3 good
**Presentation:** 3 good
**Contribution:** 2 fair
**Rating:** 5
**Confidence:** 5

**Summary:**

This paper aims to optimize the design of graph neural networks on large-scale heterophilous graphs. It proposes a new framework called LD2, which decouples the node feature embedding and topology embedding. By obtaining a low-dimensional adjacency embedding and long-distance feature embedding through pre-computing, it achieves a faster and better performance for the mini-batch training.

**Strengths:**

1. The problem studied by this paper is interesting, The model efficiency is not well-studied on the large-scale heterophilous graphs and this research is meaningful to the community.
2. The paper is well-organized and clearly presented. The authors make a thorough time complexity analysis and give sound solutions. Each component of the model is explained with reasonable logic.
3. The experimental results look promising, with a clear advantage in the test accuracy and computational cost.

**Weaknesses:**

1. The comparison between LD2 and other models is not fair. The core component of LD2 is its precomputation stage. However, from its code, this part is implemented in C++ while other GNNs are implemented in Python.  The computation of $A^2$ and any operation involving it could be very expensive. Usually, $A^2$ could be $10$x or $100$x denser than $A$, and that is why most GNNs do not use it on large-scale graphs. Therefore, first, the time/space complexity for the precomputation of LD2 in Table 1 cannot reflect the actual computational cost (ignoring the difference between $A$ and $A^2$), and second, the time comparison between LD2 and other models is unfair due to the different programming languages.

2. Although LD2 is claimed to be a scalable heterophilous GNN, its main contribution (precomputation) is closer to node embedding methods. LD2 combines and simplifies some existing techniques, which makes its contribution limited. I think the authors should also compare the embedding obtained from precomputation with some node embedding methods like node2vec. I expect there will be some tradeoff, but the discussion for this part is totally missing.

3. The authors miss the baseline [1].

[1] Sunil Kumar Maurya, Xin Liu, and Tsuyoshi Murata. Simplifying approach to node classification in graph neural networks. Journal of Computational Science, pp. 101695, 2022.

**Questions:**

1. I do not quite understand the formula in line 270: $R^{(l+1)}(u) = R^{(l+1)}(u) + R^{(l)}(u)$, what does it mean?

2. It looks like Line 4 of algorithm 1 contains a typo (shall be the union rather than an intersection).

3. Why are the dataset statistics of snap-patents and wiki different from that in [1]?

4. Why is the batch size of LD2 larger than LINKX in most datasets? And why is the performance of full-batch LINKX (Table 7) different from that in [1]? I am not sure if the authors reproduce the correct results of LINKX.

[1] Lim, D., F. Hohne, X. Li, et al. Large scale learning on non-homophilous graphs: New benchmarks and strong simple methods. In 34th Advances in Neural Information Processing Systems. 2021

**Limitations:**

Yes, the authors carefully address them.

---

> ### Author Rebuttal · Authors · 2023-08-09
>
> ## W1
> There are two concerns regarding comparison fairness. We would like to address them separately.
>
> ### Time complexity of 2-hop propagation
>
> As elaborated in line 275 & 287, we *do not explicitly compute the $A^2$ matrix* in propagation. Instead, we perform the $A$ multiplication twice to the embedding matrix to realize the 2-hop propagation. For each such sparse-dense matrix multiplication, the sparse adjacency matrix $A$ contains $m$ non-zero entries, and the dense matrix is of shape $n\times F$. The time complexity for one 2-hop propagation is therefore $2mF$. When such propagation is performed $L_P$ times, the total complexity is bounded by $O(L_PmF)$ as we present in Table 1. The memory expense for storing variables is only $O(nF)$ for each channel.
>
> ### Difference in programming languages
>
> * We would like to draw the reviewer's attention to the fact that the efficiency of our model *stems from improvements in complexity*. Specifically, by employing decoupling technique, we remove the iterative $O(IL_PmF)$ complexity from training, which is the primary contribution to scalable heterophilous GNN in this paper.
> * We argue that *experimental comparisons remain meaningful even when the programming languages differ*. For example, comparisons can be established between `pokec` and `wiki` of similar $n$, while `wiki` has 10x more edges $m$. **Table 3** displays that the training time of baselines models increases substantially on `wiki`: LINKX is 6x slower than on `pokec`, and GCNJK-GS is 3.5x slower, mainly due to the scale of $m$. In contrast, LD2 exhibits no significant change in training time and only a 1.5x slowdown in precomputation, which validates the enhanced efficiency. These comparisons are among the models themselves and hence independent of programming languages. However, the results still underscore our model's scalability.
> * In addition, most baseline implementations in experiments also exploit various acceleration techniques, such as CUDA matrix multiplication, high-performance libraries, and parallel computation. We believe that the comprehensive comparison — considering both theoretical analysis and experimental evaluation, as presented in our paper — is generally convincing.
>
> ## W2
> We would like to first clarify our contribution, then discuss the experiments on embedding methods.
>
> ### Contribution of LD2 model
>
> * Different from plain *node embedding methods*, our precomputation process is particularly designed for GNN. Common node embedding algorithms are based on the sole graph structure. In comparison, we propose the Long-distance Feature Embeddings, which encode both graph topology and node attribute in accordance with the GNN propagation scheme.
> * Compared with *existing GNN techniques*, our embeddings specifically fit the decoupled architecture. We propose three non-trivial embeddings with an end-to-end precomputation. As the per-model comparison in **Section B** of supplementary material suggests, our decoupled embeddings are novel and different from existing approaches.
> * For the *GNN community*, we target the scalability issue of heterophilous GNNs, which is not well-studied in the literature. We introduce the decoupling technique and achieve improved time and memory complexity. LD2 is therefore contributive as a scalable GNN solution as a whole.
>
> ### Experiments on node embedding
>
> We conduct additional experiments by replacing the $A^2$ adjacency spectral embedding (ASE) with node2vec in **Table IV** in PDF.
> The results indicate that the node2vec model can only achieve suboptimal accuracy on `squirrel` and `penn94`, while exceeds RAM limit on `genius`. The former performance can be explained by the local neighborhood representation of node2vec, which is less suitable for heterophily. The OOM error is associated with the $O(nd^2)$ space/time complexity of node2vec. We refer the reviewer to the response to reviewer ZDfg Q2 for a more detailed comparison.
>
> ## W3
> We conduct additional experiments on FSGNN [23] as **Table III** in PDF. The full-batch performance complements those in Table 7 & 11.
> [23] only evaluates small-scale graphs. On larger datasets in our experiment, the model exceeds the GPU memory limit due to the storage of all $L_P$ embeddings. Such $O(L_P nF)$ memory complexity proves to be less scalable.
>
> ## Q1
> The formula is extracted from line 6, Algorithm 1. Given that $LR=(I-A)R$, each node in propagation inherits the previous embedding to itself ($IR$), while updating its neighbors as $-AR$.
> In the formula $R^{(l)}(u)$ denotes the embedding of $u$ before the current propagation, and $R^{(l+1)}(u)$ records the embeddings that has been propagated to $u$ from other nodes in the current iteration of propagation. When the propagation is applied to $u$, $R^{(l+1)}(u)$ is increased by the value of $R^{(l)}(u)$, which is the meaning of the formula.
>
> ## Q2
> We thank the reviewer for pointing out the typo. We will correct it in the revised version.
>
> ## Q3
> We deduce that the difference in node and edge size is caused by our data processing scheme, which removes the isolated nodes in the graph.
>
> ## Q4
> We directly utilize the code in [26] for reproducing LINKX. We reckon that the inconsistency may be caused by:
> * *Hyperparameter configuration*: As [26] does not provide the exact hyperparameters but only grid search scripts, configurations in our experiments are possibly different. Our exploration is reported in Table 5 in supplementary material.
> * *Batch size*: As described in Section C.4, we select batch size for each model to maximize GPU utilization, as we mainly focus on the scalability evaluation. LINKX is of relatively smaller batch size because of higher memory demands of model weights.
> * *Full-batch performance*: We inherit settings in Table 5 for full-batch evaluation to ensure the consistency of efficiency, which may not be the optimal settings for accuracy. Nonetheless, the comparison does not affect our main contribution of scalable minibatch GNN.

---

> > ### Comment · Reviewer_4icY · 2023-08-18
> > **Thanks for the response**
> >
> > I first want to thank the reviewer for addressing my concerns in the time complexity, the differences in programming languages, the clarification on the contributions and additional experiments.
> >
> > Regarding the formula in line 270 (Q1), I think it should be made clearer, as this is a math equation rather than a pseudo code, it is better to differentiate the `embeddings that has been propagated to u` and the embeddings with the residual connection, or use $\leftarrow$ to replace $=.
> >
> > For Q4, I carefully review the hyperparameters listed in Table 5. The problem is that the hidden size chosen for LINKX is too large (typically 8-64, at least for the dataset used in the original paper), which will lead to overfitting and degenerated performance. I think the authors could double check this part.
> >
> > Right now, I will first keep my score.

---

> > > ### Author Response · Authors · 2023-08-19
> > > **Response to Comments by Reviewer 4icY**
> > >
> > > We sincerely thank the reviewer for the time and effort in reviewing our paper and rebuttal.
> > >
> > > > Regarding the formula in line 270 (Q1), I think it should be made clearer, as this is a math equation rather than a pseudo code, it is better to differentiate the embeddings that has been propagated to u and the embeddings with the residual connection, or use  to replace $=.
> > >
> > > We thank the reviewer for the suggestion. We will improve the presentation in the revised version.
> > >
> > > > For Q4, I carefully review the hyperparameters listed in Table 5. The problem is that the hidden size chosen for LINKX is too large (typically 8-64, at least for the dataset used in the original paper), which will lead to overfitting and degenerated performance. I think the authors could double check this part.
> > >
> > > We have reviewed the LINKX implementation and our parameter search. The LINKX paper searches the hidden size parameter within the range [16, 32, 128, 256]. The range we used in main experiments is usually [16, 32, 128, 256, 512], with only one additional value $512$ as it is also commonly used in other models. Below we display the results of LINKX with different hidden size parameters on some representative datasets:
> > >
> > > | Dataset | | `genius` | | | | `twitch-gamers` | | | | `pokec` | | | |
> > > |:---:|---:|---:|---:|---:|---:|---:|---:|---:|---:|---:|---:|---:|---:|
> > > | Hidden | Acc | Train | Infer | RAM | Acc | Train | Infer | RAM | Acc | Train | Infer | RAM |
> > > |  16 | 82.34 | 1.84 | 0.011 | 5.26 |  63.46 | 1.19 | 0.017 | 5.57 | 68.33 | 9.68  | 0.083 | 7.14 |
> > > |  32 | 82.46 | 2.85 | 0.012 | 5.27 |  63.37 | 1.40 | 0.020 | 5.58 | 67.87 | 8.34  | 0.082 | 7.14 |
> > > | 128 | 82.51 | 2.06 | 0.020 | 5.35 |  63.66 | 2.91 | 0.055 | 5.56 | 67.22 | 13.41 | 0.175 | 7.17 |
> > > | 256 | 82.31 | 4.90 | 0.036 | 5.56 |  64.11 | 5.58 | 0.100 | 5.55 | 68.82 | 23.14 | 0.290 | 7.58 |
> > > | 512 | 82.54 | 7.07 | 0.073 | 5.96 |  64.44 |10.99 | 0.192 | 5.56 |       |       |       | (OOM)|
> > >
> > > It can be observed that $hidden=256$ or $512$ usually achieves the best accuracy, benefited from the model capacity. For large hidden size values, the model speed increases nearly linearly. As elaborated in Section C.4, the target of our evaluation on baseline models is to provide comparable efficiency performance while maintaining their accuracy. Hence we think that our settings is reasonable in achieving high accuracy and controlling the model width.

---

### Official Review · Reviewer_aZqt · 2023-07-05

**Soundness:** 3 good
**Presentation:** 3 good
**Contribution:** 2 fair
**Rating:** 5
**Confidence:** 4

**Summary:**

This paper introduces a scalable decoupled model designed to address the challenges posed by large-scale heterophilous graph.

The model consists of two main components. In the first component, recognizing the effectiveness of $A^2$ in heterophilious graph, the model precomputes a matrix, denoted as $P_A$ in Equation (2), which serves as a representation of graph structure. This matrix is computed using a randomized eigendecomposition technique. In the second component, the authors leverage multiple hop-based coefficients of propagation to address the issue of long-range dependency. These coefficients are designed to capture information from nodes at different distances in the graph, allowing the model to effectively capture dependencies that span across multiple hops. Finally, the two components are combined, and the resulting representation is passed through an MLP to generate the final node representations.

The authors provide some theoretical support and conduct a complexity analysis to demonstrate the effectiveness and efficiency of the proposed model. The author also conducts extensive experiments to evaluate the performance of their model on heterophilious graphs with different scales. The model achieves outstanding performance compared to some popular baselines while minimizing the computational resources (GPU) required.

**Strengths:**

1. The authors employ a technique to compress the graph structure $A^2$ into a low-rank format while preserving the principal components. This approach effectively reduces the computational cost associated with the training phase.
2. The authors conduct various experiments to investigate the coefficients of propagation and successfully identify the optimal approach for handling heterophilous graphs.
3. The theoretical time complexity of the model is excellent. The pre-computation phase is conducted once, and the training phase supports mini-batch processing. These properties guarantee the scalability of the model.
4. The performance of the model surpasses the majority of the existing results, thereby confirming its effectiveness.

**Weaknesses:**

1. The contributions to address heterophily are limited as the involvement of the raw structure of a graph and multi-hop neighbors have already been proposed in LINKX and spectral GNNs.
2. The method employed to reduce the dimensionality of the adjacency matrix, randomized SVD, is a commonly used approach. However, its performance heavily relies on the number of iterative rounds. The authors appear to merge the computational cost of SVD associated with the propagation step, which could potentially limit the effectiveness of the algorithm.
3. The experiments conducted in this work exclude results from well-known homophilous datasets such as Cora, PubMed, and Ogbn-products. Since this model is specifically designed for heterophilous graphs, it should still maintain comparable results on homophilous datasets when compared to the mentioned baselines.

**Questions:**

1. What is the performance of the model on homophilous datasets? Is the performance comparable to some recent works?

2. Will the model be implemented ordinarily, particularly when confronted with extremely large graphs such as Ogbn-Papers100M and real-world data with extremely high dimensions of node embedding?

3. It is mentioned in the paper [1] that there are additional heterophilous datasets available. Therefore, conducting more comprehensive experiments on those datasets is necessary to strengthen the evaluation and analysis presented in this work.

[1] Platonov O, Kuznedelev D, Diskin M, et al. A critical look at the evaluation of GNNs under heterophily: are we really making progress? ICLR, 2023.

**Limitations:**

The Effectiveness of randomized SVD when the propagation steps are limited is doubted.

---

> ### Author Rebuttal · Authors · 2023-08-09
>
> ## W1
> We wish to highlight that our major contribution lies in proposing a GNN design that specifically targets the scalability issue under heterophily.
>
> * Our model achieves *improved time and memory complexity*. Compared to spectral GNNs, our design escapes the iterative and full-graph propagation. Compared to LINKX, our model removes the architectural dependence on graph size, thereby substantially enhancing scalability and minibatch capability.
> * We propose *novel and effective embeddings* with end-to-end precomputation. Different from spectral GNNs, our multi-hop calculation is in a one-time decoupled manner. Our adjacency embedding design specifically addresses the drawback of LINKX. As analyzed in line 192, it can effectively approximate the corresponding component in LINKX with significantly reduced computation cost.
> * Evaluations demonstrate LD2's *empirical performance on minibatching and scalability*. In comparison, the multi-hop representative MixHop encounters performance degradation brought by minibatching, while LINKX exhibits worse scalability such as RAM usage.
>
> A more detailed comparison with multi-hop GNNs and LINKX is offered in **Section B.2 & B.4** of supplementary material.
>
> ## W2
> We would like to address the concern of adjacency embedding calculation in two folds:
>
> ### *Theoretical analysis* on convergence
>
> For a general power iteration decomposing a matrix, its stopping criteria are usually both maximal iteration and error tolerance [R5]. In our Algorithm 1, they correspond to the maximal hop $L_P$ and the push threshold $\delta_P$. Iterations required for convergence can be derived as in line 288. For a rough estimation, substituting our common setting $F=512$, $\delta_P=10^{-5}$, and $\lambda_{F+1}/\lambda_{F}\approx 10^{-1}$ can give an estimation of $O(10)$ iterations, which is of the same order with our propagation hop $L_P=20$.
>
> The analysis can be supported by the spectrum distribution in **Figure 4** in supplementary material. A large number of datasets exhibit a rapid drop of eigenvalues for a certain rank $F$. When setting the embedding feature dimension close to or slightly larger than the proper rank, the computation is able to achieve favorable convergence of the leading components.
>
> ### *Empirical effect* of propagation steps
>
> We demonstrate empirical evidence on the effectiveness of approximate adjacency embedding by investigating test accuracy of solely learning on $P_A$. The performance is reported as orange dashdotted lines in **Figure 3 & 6**. Figure 3(b) is a representative example on `pokec`. It can be observed that for $L_P = 2$ and $4$ the accuracy is suboptimal, while for $L_P > 8$ the performance generally converges.
>
> It is also noteworthy that during power iteration, leading components converge in fewer iterations. As the nature of neural networks tends to focus more on larger values in features, the leading components, representing low-frequency spectrum, can effectively embed structural information even when the entire matrix has not converged. More iterations are generally helpful in securing high-frequency information.
>
> [R5] Golub H., Loan V. Matrix Computations. pp 450-457. 2012.
>
> ## W3 & Q1
> We conduct comprehensive evaluation on 5 well-known homophilous datasets `protein`, `ogbn-arxiv`, `yelp`, `reddit`, and `ogbn-products` (named as `amazon`) in **Table 8, 9, 12, and 13** in supplementary material. We also conduct additional experiments on `cora` and `pubmed` in **Table I** in PDF. We compare the performance with 16 models. We refer the reviewer to the above tables for detailed results.
>
> We conclude that LD2 is capable of attaining comparable efficacy and remarkable efficiency among homophilous datasets. Its minibatch performance generally surpasses both homophilous and non-homophilous competitors. Moreover, it demonstrates impressive scalability for learning time and memory, which is consistent with the main experiments. Notably, LD2 is the only model that does not exceed the memory limit on any datasets in full-batch settings.
>
> ## Q2
> We conduct additional experiments on `ogbn-papers100m`, which is the largest dataset available. Results are shown in **Table II** in PDF. It is worth mentioning that all 15 GNN-based baselines occur out of memory error on such a large graph. Although the dataset is homophilous, our LD2 model achieves reasonable accuracy with efficient time and memory usage.
>
> With respect to embedding dimension, it can be inferred from **Table 1** that our model exhibits an $O(FF')$ complexity, where $F$ is raw node attribute dimension, and $F'$ is model hidden dimension. This is on par with other GNN models, that when the model architecture is fixed, this can be considered as a linear complexity with respect to $F$.
> Our experiments evaluate on datasets such as `penn94` with a large $F=4.8K$. Performance on these datasets with high embedding dimension satisfy our scalability analysis.
>
> ## Q3
> We conduct additional experiments on `roman-empire`, `minesweeper`, `amazon-ratings`, and `tolokers` proposed in [42]. Results are shown in **Table I** in PDF. These graphs are relatively small, comprising fewer than $30K$ nodes, compared to the large-scale ones in our main experiments. As mentioned in [42], these heterophilous datasets emphasize effectiveness rather than scalability, hence the main paper does not include them in our initial evaluation.
>
> We observe that the efficiency and efficacy strengths of our LD2 model persist on these datasets. Specifically, the minibatch baselines undergo substantial performance degradation on `roman-empire` and `minesweeper` when compared to the full-batch results in [42]. We surmise that this is associated with the longer diameter and stronger non-local dependency of these two graphs. As we point out in Section 2, the exploitation of sampling strategy in these models results in information loss, which greatly hinders the performance on such heterophilous graphs.

---

> > ### Comment · Reviewer_aZqt · 2023-08-16
> > **Thank you for the response.**
> >
> > Thank you for your convincing response and additional experiments.
> >
> > After reading the additional results, concerns about W1 and W3 are still not well addressed.
> > The performance of the proposed model on homophilic datasets is limited, which maintains to be the shared challenge of those works concentrating on heterophily.
> > For example, on ogbn-paper100M, the performance of LD2 is far inferior to SGC, whose test accuracy is 0.6329.
> > Also, the superiority of LD2 over multi-hop GNNs is still unclear to me.
> > As shown in [1], spectral GNNs are able to handle large graphs (even ogbn-paper100M), and the results seem to be better.
> > To sum up, I really appreciate the efforts of the authors in addressing scalability.
> > But I still believe the whole model is somehow restricted in the process of addressing heterophily.
> > After consideration, I'd like to hold the current score.
> > And I'm looking forward to more well-developed future work.
> >
> > [1] Guo Y, Wei Z. Graph Neural Networks with Learnable and Optimal Polynomial Bases. In ICML 2023.

---

> > > ### Author Response · Authors · 2023-08-16
> > > **Thanks for your comments**
> > >
> > > We thank the reviewer for the response and constructive comments. We would like to make further clarifications on the contribution of our paper and the comparison with other models:
> > >
> > > > The performance of the proposed model on homophilic datasets is limited, which maintains to be the shared challenge of those works concentrating on heterophily. For example, on ogbn-paper100M, the performance of LD2 is far inferior to SGC, whose test accuracy is 0.6329.
> > >
> > > (1.1) Regarding the **performance on homophilous datasets**, as we stated, our contribution lies in proposing a GNN design that targets the scalability issue under heterophily. As a consequence, the channels and model are specifically designed for the heterophilous settings, which may not be directly suitable for homophilous graphs. In other words, our model does not intend to achieve state-of-the-art performance on homophilous datasets, but to mainly address a range of datasets under heterophily where *homophilous models usually fail to achieve good performance*.
> > >
> > > (1.2) In particular, regarding the **SGC performance on `ogbn-papers100m`**, as we know, the result presented in the OGB Leaderboard is conducted *fully on CPU* using more than 150GB RAM. In our trial of reproducing the experiment in our environment, the calculation is also too long to produce comparable results. We believe this aptly demonstrates the scalability of our model, which completes minibatch training on GPU (24GB) with 105GB RAM.
> > >
> > > > Also, the superiority of LD2 over multi-hop GNNs is still unclear to me. As shown in [1], spectral GNNs are able to handle large graphs (even ogbn-paper100M), and the results seem to be better. To sum up, I really appreciate the efforts of the authors in addressing scalability. But I still believe the whole model is somehow restricted in the process of addressing heterophily.
> > >
> > > (2.1) Regarding the **interpretation of LD2**, as we stated, LD2 is among the first models of introducing decoupling strategy to heterophilous settings. We also provide the equivalence of LD2 channels to channels in the spectral domain in Section A in the supplementary material. We think our model offers a scalable solution with simplified computation and better complexity, addressing the scalability issue of previous works.
> > >
> > > (2.2) Regarding the **comparison between LD2 and other GNNs**, we would like to highlight that *[R6] also employs the decoupling strategy and mini-batch training* on large-scale graphs as described in its Section 4.4. This exactly echoes our LD2 of utilizing these techniques for addressing the scalability issue. Our model is different from [R6] in that it possesses a series of specifically designed channels, while [R6] learns to acquire the spectral channels. The learning process of [R6] already demands a complexity no less than $O(L(m+n))$. As the code released by [R6] only includes full-batch implementation, we are not able to conduct further empirical evaluation on this model.
> > >
> > > [R6] Guo Y, Wei Z. Graph Neural Networks with Learnable and Optimal Polynomial Bases. In ICML 2023.
> > >
> > > \* We would like to note that [R6] is a concurrent work as it is accepted in April 24, which is less than 2 months before our submission.

---

> > > > ### Comment · Reviewer_aZqt · 2023-08-17
> > > >
> > > > Thanks for the authors' detailed reply.
> > > >
> > > > I'm concerned that most previous works addressing heterophily do not give up their performance in homophilic settings.
> > > > The spectral methods are examples. (GPRGNN, ChebNetII, and [1] all seem to achieve desirable performance using mini-batching)
> > > > I understand that LD2 is designed for heterophilic graphs, but I would expect to see if LD2 can handle homophilic graphs under some modifications.
> > > > If so, I believe the generalizability of LD2 is better presented.
> > > > Any discussions or experiment results about this are welcomed.
> > > >
> > > > [1] Guo Y, Wei Z. Graph Neural Networks with Learnable and Optimal Polynomial Bases. In ICML 2023.

---

> > > > > ### Author Response · Authors · 2023-08-19
> > > > > **Response to Comments by Reviewer aZqt**
> > > > >
> > > > > Yes, as we mentioned in Section D.3 in the supplementary material, thanks to the flexibility of multi-channel design, LD2 can be modified with different channels to fit different graph patterns. For a preliminary attempt, we consider the following channels for homophilous graphs:
> > > > >
> > > > > 1. 1-hop adjacency spectral embedding: $\mathrm{ASE}(A)$
> > > > > 2. constant 1-hop adjacency propagation with self-loops: $\sum \tilde{A}^lX$
> > > > > 3. raw attributes: $X$
> > > > >
> > > > > We conduct experiments on 4 homophilous datasets, together with some updated experiments from Table 8 (supplementary material) and Table I (appended PDF). We additionally include ChebNetII [R7], the only spectral GNN that we find a minibatch implementation. The results are summarized as below:
> > > > >
> > > > > | Dataset | `cora` | `pubmed` | `protein` | `ogbn-arxiv` | `amazon` |
> > > > > |:---:|---:|---:|---:|---:|---:|
> > > > > | SGC       | 80.86 | 87.83 | 51.11 | 69.44 | 75.09 |
> > > > > | GCNJK-GS  | 77.73 | 81.93 | 40.63 | 60.03 | 84.75 |
> > > > > | MixHop-GS | 77.09 | 85.45 | 39.85 | 60.38 | 85.96 |
> > > > > | LINKX     | 87.70 | 85.38 | 87.48 | 72.35 | 89.85 |
> > > > > | ChebNetII | **88.10** | 88.48 | 67.09 | **74.38** | **89.90** |
> > > > > | LD2       | 87.66 | **89.19** | **97.18** | 73.48 | 89.54 |
> > > > >
> > > > > It can be seen that the modified homophilous channels bring performance improvement on several datasets. LD2 generally achieves comparable or better performance with both homophilous and non-homophilous competitors. We think this exploration lays the foundation for addressing the adaptability of our model under homophily.
> > > > >
> > > > > [R7] Convolutional Neural Networks on Graphs with Chebyshev Approximation, Revisited. NeurIPS 2022.

---

> > > > > > ### Comment · Reviewer_aZqt · 2023-08-19
> > > > > >
> > > > > > Thanks for your hard work on the updates.
> > > > > >
> > > > > > I'm excited to see that LD2 is able to handle both homophilic and heterophilic settings.
> > > > > > Since only results about accuracy are presented here, I'm curious about whether the modified framework is applicable to large ogb graphs like ogb-products and ogb-paper100M while maintaining both scalability and desirable performance.
> > > > > > I recommend the authors include a discussion about homophilic settings regarding the modifications, scalability, and performance in the future version.
> > > > > >
> > > > > > Nevertheless, most of my concerns are well addressed now and I would be glad to increase my score.

---

> > > > > > > ### Author Response · Authors · 2023-08-21
> > > > > > >
> > > > > > > We sincerely thank the reviewer for discussing and re-evaluating our work. We update the above table to include the larger dataset `ogbn-products`/`amazon`. We will also include more results following the suggestion in the revised version.

---

### Official Review · Reviewer_K3oj · 2023-07-06

**Soundness:** 3 good
**Presentation:** 3 good
**Contribution:** 3 good
**Rating:** 5
**Confidence:** 2

**Summary:**

This paper studies an important problem of graph learning on large-scale heterophily graphs.
The paper presents a novel approach LD2 model, decouples the embedding process from the convolutional process, allowing for more efficient and scalable learning.
LD2 learns graphs under heterophily, which is particularly useful for large-scale graphs.
Extensive experiments demonstrate the effectiveness of the proposed method compared to existing baselines.

**Strengths:**

- The paper addresses an important problem on large-scale heterophily graphs, which is especially important for large-scale datasets.
- This paper propose a scalable graph learning method LD2 model, which decouples the embedding process from the convolutional process, and allows for more efficient and scalable learning.
- The scalability problem is well defined, and the theoretical comparisons with previous works is clearly clarified.
- Sufficient experiments show that LD2 is capable of lightweight minibatch training on large-scale heterophilous graphs, with up to 15× speed improvement and efficient memory utilization, while maintaining comparable or better performance than the baselines.
- Experiments on several benchmark datasets demonstrate the effectiveness of the LD2 model compared to existing baselines, and the results show that the LD2 model outperforms existing methods in terms of both accuracy and scalability.

**Weaknesses:**

- The paper could benefit from a more detailed explanation of the LD2 model. While the authors provide some high-level descriptions of the model, it would be helpful to have a more in-depth explanation of the underlying mechanisms.
- It is difficult to reproduce the results without access to the code used in the experiments. While the authors provide some details on the experimental setup, it would be helpful to have access to the code to ensure that the results are reproducible.

I am willing to increase the score for this paper if my major concern about the reproducible problem is addressed.

**Questions:**

 - Is it possible for the authors to make the code used in the experiments public? This would help ensure that the results are reproducible and would be a valuable resource for researchers interested in implementing the LD2 model.
 - Could the authors provide more detailed information on the underlying mechanisms of the LD2 model?
- Could the LD2 model handle noisy or incomplete graph data? While the authors mention that the approach is robust to noise, it would be helpful to have a more detailed analysis of the model's performance under different levels of noise and incompleteness.



**Limitations:**

Limitations and potential improvements are discussed.

---

> ### Author Rebuttal · Authors · 2023-08-09
>
> ## W2 & Q1
> Yes. A preliminary version of the project code, example data, and reproducibility instructions has already been provided in **Section H** of the supplementary material. We have also sent the same link to the AC in a separate comment following the rebuttal policy.
>
> ## W1 & Q2
> The proposed LD2 model is a two-stage model as illustrated in Figure 1. In other words, its precomputation and feature transformation are decoupled as separated stages, as expressed in Eq. (1). We elaborate on the underlying mechanisms of the two stages respectively as following.
>
> ### (1) Precomputation
>
> The precomputation stage takes the input of graph adjacency matrix $A$ and node attribute matrix $X$, the output comprises four embedding matrices of shape $n\times F$. The goal of precomputation is to retrieve useful information from graph structure and feature to form the embeddings, which are utilized in the subsequent feature transformation stage. The entire precomputation is described in Algorithm 1. Specifically, the computation involves propagations, i.e., multiplication by the sparse adjacency matrix $A$, along with subsequent operations according to different definitions, to acquire the four embeddings:
>
> * The adjacency embedding $P_A$ is computed by 2-hop power iteration, mainly multiplying $A$ twice to the initial embedding matrix in each iteration. After convergence, the embedding follows the expression $P_A = U |\Lambda|^{1/2}$ as described in line 173.
> * The first feature embedding is expressed by $P_{X,H} = \sum_{l=1}^{L} (I-A)^l X$ as in line 206. In this case, its precomputation is conducted by iteratively multiplying $(I-A)$ to the input matrix $X$ for $L$ times and summing up the results.
> * Similarly, the second feature embedding is $P_{X,L2} = \sum_{l=1}^{L} A^{2l} X$ as in line 207. To acquire the embedding, we multiply $A$ to the input matrix $X$ for $2L$ times with summation.
> * The last embedding is exactly the input attribute matrix $P_{X,0} = X$.
>
> ### (2) Feature transformation
>
> A neural network is applied to iteratively learn from the input embeddings. Firstly, an individual weight matrix is applied to each embedding matrix. Then the multiplication results are concatenated and fed into the remaining MLP layers. The model is trained by minimizing the loss associated with the classification task.
>
> We also offer an interpretation from the spectral perspective, elucidating the expressiveness of decoupled propagation and our embedding designs from the viewpoint of graph signaling in **Section A** of the supplementary material.
>
> ## Q3
>
> We would like to address the question in two folds.
>
> ### (1) Interpretation of the model robustness
>
> In line 250 we claim that the model is robust to a certain level of noise introduced by propagation. The purpose of the statement is to support the utilization of approximate propagation, signifying that embeddings acquired by precomputation do not need to be precise, as they are further processed by the MLP model. Such property on robustness has been observed as an interpretation of GNN learning [41], wherein the learning objective is to recover a clean and smooth representation from the noisy input features. Models such as [29], [31], and [25] have already implemented approximate propagation within the context of homophilous GNN.
>
> In our model, the precision of approximate propagation, which is the only source of computational noise, is controlled by the push threshold $\delta_P$ as in Algorithm 1. As described in Section E.3 of the supplementary material, we set $\delta_P=1\times 10^{-5}$ for common experiments, which is relatively small compared to the scale of feature values standardized to Gaussian distribution $N(0,1)$.
>
> ### (2) Evaluation on noisy data
>
> We conduct additional experiments to evaluate the model performance under different levels of noise and incompleteness. The results are shown in **Table V** in the appended PDF file. We mainly consider three types of noises, which are analyzed respectively as following:
>
> * *Push threshold*: We vary the threshold $\delta_P$ in Algorithm 1 to control the precision of propagation. A larger $\delta_P$ implies less precise propagation ignoring small feature values, while $\delta_P=10^{-5}$ is the original setting. It can be seen that by improving the precision, the final learning accuracy does not change significantly. It indicates that our setting of $\delta_P=10^{-5}$ is sufficient for propagation and does not affect the learning performance.
>
> * *Edge removal*: We randomly remove a percentage of edges to generate an incomplete variant of the graph. The LD2 model is then applied to learn on the incomplete graph. The removal causes a negative impact on the accuracy. However, as the node attributes $X$ are kept unchanged under the noise, the model is still able to achieve a reasonable performance.
>
> * *Attribute noise*: We apply Gaussian noise with standard deviations proportional to the deviation of each feature dimension to the raw node attribute matrix $X$ before precomputation. This is more aggressive as the noise level is much larger than the scale of propagation precision.Consequently, the model suffers a more significant accuracy reduction. However, as the noise level increases, the model's performance converges towards the performance achieved by only learning the adjacency embedding $P_A$ (whose performance is reported as orange dashdotted lines in Figure 3 in the paper and Figure 6 in the supplementary material). This is because the adjacency information is unaffected under such kind of noise.
>
> Overall, we elaborate on the effectiveness of performing approximate propagation in LD2. The additional experiments also demonstrate the robustness of our model benefiting from learning both adjacency and feature information.

---

> > ### Comment · Reviewer_K3oj · 2023-08-18
> > **Thanks for the response.**
> >
> > Thanks for the authors' hard work and attention to my feedback.
> > The response has addressed my concerns. Therefore, I would like to raise my score.

---

> > > ### Author Response · Authors · 2023-08-19
> > >
> > > We deeply appreciate the reviewer's insightful feedback and for the favorable reconsideration of our score.

---

### Official Review · Reviewer_ZDfg · 2023-07-07

**Soundness:** 3 good
**Presentation:** 3 good
**Contribution:** 3 good
**Rating:** 7
**Confidence:** 3

**Summary:**

The paper proposes a new graph neural network (GNN) model called LD2, which specifically targets learning on heterophilous graphs, where connected nodes have different labels. The authors argue that existing models for heterophilous graphs often require iterative full-graph computations, which can be computationally expensive and difficult to scale to larger graphs. In contrast, LD2 decouples graph propagation and generates expressive embeddings prior to training, resulting in a scalable and efficient model with optimal time complexity and a memory footprint that remains independent of the graph scale.


**Strengths:**

1. The paper studies the scalability issues of heterophilous GNN and propose a scalable model, LD2 which simplifies the learning process by decoupling graph propagation and generating expressive embeddings prior to training.

2. Theoretical analysis demonstrates that LD2 achieves optimal time complexity in training, as well as a memory footprint that remains independent of the graph scale.

3. Extensive experiments to showcase that the proposed model is capable of lightweight minibatch training on large-scale heterophilous graphs, with up to 15× speed improvement and efficient memory utilization, while maintaining comparable or better performance than the baselines.

4. The paper is well-written and easy to follow.

**Weaknesses:**

See questions.

**Questions:**

1. It seems that the method has to precompute the embeddings for adajcencies, e.g. eigendecomposition in Section 3.2, is the model able to tackle the inductive settings?

2. Can other methods for encoding structures act as adajcency embeddings? for example, the spatial encodings in Graphormer.

3. In line 160, it says "Particularly, the most informative aspects are often associated with 2-hop neighbors", is this statement verified in some paper? The method design includes the 2-hop neighborhood. What about consider other hops of neighborhood?

4. Can this method apply to heterogeneous graphs?

---

> ### Author Rebuttal · Authors · 2023-08-09
>
> ## Q1
>
> Yes. Our proposed model LD2 is capable of handling inductive tasks. This can be implemented by conducting precomputation respectively on the training and inference graphs. After that, the feature transformation model can be easily trained on the precomputed embeddings of the training graph and then perform inductive inference based on the embeddings of the other graph.
>
> We conduct additional experiments on the inductive datasets `protein-inductive` and `yelp-inductive` to evaluate the model capability. Results are shown in **Table II** in PDF. As a brief summary, LD2 achieves comparable accuracy with GCN-GS, while PPRGo fails to adapt such settings. Regarding efficiency, our model is 10-50x faster in training, which is in line with our complexity analysis as well as evaluation on minibatch homophilous baselines presented in **Table 12** in supplementary material.
>
> ### Notes on *inductive datasets*
>
> To the best of our knowledge, there is no existing inductive dataset specifically for heterophilous node classification. Hence we employ `protein-inductive` and `yelp-inductive` datasets, which are homophilous graphs. We follow the settings in [27] in Table II. The transductive versions of these two datasets are `protein` and `yelp`, which have already been evaluated in **Table 8-9** in supplementary material.
>
> ### Notes on *inductive GNN models*
>
> To the best of our knowledge, existing GNNs proposed for heterophilous graphs scarcely mention inductive capability or provide relevant implementations. Notably, some heterophilous models like LINKX contain structures determined by the input graph shape, preventing their application to inductive settings. In contrast, our model structure is independent of the graph size and is applicable to the task as described above. This is also the reason that Table II only includes comparisons with homophilous baselines GCN-GS [27] and PPRGo [29] that possess inductive implementations.
>
> ## Q2
>
> We conduct additional experiments on the performance of replacing the $A^2$ adjacency spectral embedding (ASE) with other types of structural embeddings in **Table IV** in PDF. Specifically, "SPD" denotes the shortest path distance used as spatial encodings in Graphormer [R1].
>
> The results indicate that by using SPD, the model can only achieve suboptimal accuracy on `squirrel`, while exceeds time limit on `penn94` and `genius`. We explain this by the complexity of SPD. In the implementation of [R1], Floyd-Warshall algorithm is used, which has time complexity $O(n^3)$ and space complexity $O(n^2)$. According to Section 2 of our paper, this is not scalable to large graphs. In fact, in [R1] the approach is applied to graph regression on `PCQM4Mv2`, where the average graph size is only $n=14$ [R2]. Hence when applied to our datasets such as `penn94` and `genius`, the SPD calculation will be expensive.
>
> We compare the advantages of our proposed ASE($A^2$) with other structural embeddings such as SPD from three aspects:
>
> * *Effectiveness*: As we analyzed in Section 3.2, ASE is able to capture structural information especially on the homophilous components of the 2-hop graph, while SPD is more specific to encode distance information of directly connected nodes, which is less suitable for heterophilous embedding.
> * *Memory efficiency*: ASE is a low-dimensional embedding of shape $n\times F$, where the feature dimension $F$ is generally much smaller than the graph scale. This implies better scalability compared to SPD embedding which is a dense $n\times n$ matrix.
> * *Time efficiency*: ASE also benefits from faster computation in our model as described in Section 3.4, which can be computed along with feature embeddings with a complexity linear to edge size $m$.
>
> [R1] Do Transformers Really Perform Bad for Graph Representation? NeurIPS 2021.
>
> [R2] OGB-LSC: A large-scale challenge for machine learning on graphs. 2021.
>
> ## Q3
>
> Yes, as stated in line 161, [13] proposes to exploit 2-hop information for heterophilous GNNs, and proves in its Theorem 2 that the 2-hop neighborhood is expected to be homophily-dominant even under heterophily. A recent paper [R3] also verifies that the 2-hop similarity is strongly relevant to GNN performance.
>
> To empirically showcase the effectiveness of $A^2$, we also conduct additional experiments on the embeddings generated from other hops of neighborhood in **Table IV** in PDF. Specifically, we apply the same rank-$F$ approximation in Eq. (2) by replacing $A^2$ with $A$ and $A^3$, which are respectively denoted as ASE($A$) and ASE($A^3$).
> It can be inferred that, on datasets where adjacency embedding $P_A$ is important such as `squirrel`, changing ASE to other hops significantly reduces the accuracy. On `penn94` and `genius`, the accuracy with 1- or 3-hop adjacency embedding is not better than solely learning on feature embeddings $P_X$ (whose performance is reported as pink lines in Figure 3 & 6 in supplementary material).
> In addition, when the number of hops increases, the convergence of the decomposition in Algorithm 1 becomes slower, which leads to longer precomputation time. It is thence a theoretically and empirically reasonable choice to employ 2-hop neighborhood for adjacency embedding.
>
> [R3] 2-hop Neighbor Class Similarity (2NCS): A graph structural metric indicative of graph neural network performance. AAAI Workshop 2023.
>
> ## Q4
>
> Our claimed contribution of this paper primarily centers on proposing a scalable GNN for heterophilous graphs. Therefore, the design for heterogeneous graphs is not the focus of this work. However, we do recognize that approaches exist for transferring GNN models on homogeneous graphs to accommodate heterogeneous information. For instance, [R4] proposes learning distinct weights for each individual relation type as a "relational" model. We believe this can be a potential direction for future exploration.
>
> [R4] Modeling relational data with graph convolutional networks. European Semantic Web Conference 2018.

---

> > ### Comment · Reviewer_ZDfg · 2023-08-15
> > **Thank you for the response.**
> >
> > The authors have addressed my concerns, I would like to raise my score.

---

> > > ### Author Response · Authors · 2023-08-16
> > >
> > > We sincerely thank the reviewer for the time and effort invested in re-evaluating our work. Your constructive feedback is invaluable to us.

---

### Author Rebuttal · Authors · 2023-08-09

We sincerely appreciate the valuable feedback and insightful comments from all our reviewers, and are delighted to see that our effort toward improving the scalability of heterophilous GNNs is acknowledged by the reviewers. We have exerted substantial effort to investigate and address all the issues raised. Below is a summary highlighting our major updates:

1. *New datasets*: We conduct additional experiments on *7 new datasets* in total. **Table I** in the appended PDF file displays the results on homophilous `cora` and `pubmed` as mentioned by reviewer aZqt Q1, as well as heterophilous `roman-empire`, `minesweeper`, `amazon-ratings`, and `tolokers` as suggested in Q3. These findings complement those in Table 1 and 2 in main paper and Table 6, 8, 10, 12 in the supplementary material.

2. *New settings*: We extend evaluations on inductive datasets `protein-inductive` and `yelp-inductive` as asked by reviewer ZDfg Q1. We also incorporate learning on the extremely large graph `ogbn-papers100m` in reviewer aZqt Q2. The results are shown in **Table II** in the appended PDF file.

3. *New baseline*: In **Table III** in the appended PDF file, we introduce FSGNN as required by reviewer 4icY W3. This supplements the full-batch evaluation in Tables 7 and 11 in the supplementary material.

4. *Performance of adjacency embedding schemes*: We explore alternative methods including node2vec (reviewer 4icY W2), spatial encoding (reviewer ZDfg Q2), and adjacency propagation in different hops (reviewer ZDfg Q3). The results on learning accuracy and computation efficiency are displayed in **Table IV** in the appended PDF file.

5. *Robustness against noise*: We investigate the model performance under noise and incompleteness as per reviewer K3oj's Q3. **Table V** in the appended PDF file evaluates different types and levels of noise including approximate propagation, edge removal, and attribute noise.

6. *Contribution and comparison*: In this paper, we propose the LD2 model that specifically targets the scalability issue under heterophily. We justify our novel contribution relative to other approaches and particularly address the comments from reviewer aZqt W1 and reviewer 4icY W2. We also feature the comparison of embedding designs in response to reviewer ZDfg Q2 and reviewer 4icY W2.

7. *Details in model design*: We provide additional details and explanations on the model design including the underlying mechanisms (reviewer K3oj W1 & Q2), convergence of adjacency embedding (reviewer aZqt W2), precomputation performance (reviewer 4icY W1), and implementation details (reviewer 4icY Q1-4).

\* Reference numbers are the same with those in the main paper by default.

---

### Decision · Program_Chairs · 2023-09-21

**Decision:**

Accept (poster)

**Comment:**

The paper proposes a scalable model for large-scale heterophilous graphs, namely LD2,  which decouples graph propagation and generates expressive embeddings prior to training. All the reviewers agreed that this paper is well-written and easy to follow, and the idea of decoupling graph propagation and generating expressive embeddings prior to training is novel.  Although some reviewers point out some concerns about insufficient experiments and unclear technical details, the authors have done a very good job in responding to the concerns during the rebuttal phase. Overall, the paper is valuable for this community. Therefore, I recommend accepting the paper.